# Diversity of meso-scale architecture in human and non-human connectomes

Richard F. Betzel[1], John D. Medaglia[2,3] & Danielle S. Bassett [1,3,4]

Brain function is reflected in connectome community structure. The dominant view is that communities are assortative and segregated from one another, supporting specialized information processing. However, this view precludes the possibility of non-assortative communities whose complex inter-community interactions could engender a richer functional repertoire. We use weighted stochastic blockmodels to uncover the meso-scale architecture of *Drosophila*, mouse, rat, macaque, and human connectomes. We find that most communities are assortative, though others form core-periphery and disassortative structures, which better recapitulate observed patterns of functional connectivity and gene co-expression in human and mouse connectomes compared to standard community detection techniques. We define measures for quantifying the diversity of communities in which brain regions participate, showing that this measure is peaked in control and subcortical systems in humans, and that inter-individual differences are correlated with cognitive performance. Our report paints a more diverse portrait of connectome communities and demonstrates their cognitive relevance.

[1] Department of Bioengineering, University of Pennsylvania, Philadelphia, PA 19104, USA. [2] Department of Psychology, Drexel University, Philadelphia, PA 19104, USA. [3] Department of Neurology, Perelman School of Medicine, University of Pennsylvania, Philadelphia, PA 19104, USA. [4] Department of Electrical and Systems Engineering, University of Pennsylvania, Philadelphia, PA 19104, USA. Correspondence and requests for materials should be addressed to D.S.B. (email: dsb@seas.upenn.edu)

Cognitive processes are thought to emerge from the coordinated activity of distributed networks of neural elements, from small-scale neuronal populations to large-scale brain areas[1,2]. This coordination is facilitated by the brain's network of physical, hard-wired connections—its connectome[3]. Accordingly, the range of cognitive processes in which a neural element participates as well as its computational capacity depends critically on its connectivity profile, i.e., its set of outgoing and incoming connections along which it transmits information to and receives information from other brain areas[4].

While individual neural elements are thought to perform local operations, their organization into motifs, circuits, and clusters engenders a richer, more diverse functional repertoire[5]. In particular, the connectome's communities—which collectively comprise its so-called meso-scale structure—have attracted a great deal of attention (see ref. [6] for a recent review). Here, we define the meso-scale as the level between that of individual nodes and the network as a whole. At that scale, a network's nodes can be grouped into clusters called "communities", which are usually assumed to be assortative, meaning that nodes preferentially connect to nodes with similar attributes, namely membership to the same community[7]. The resulting communities are internally dense and externally sparse, and are oftentimes described as "nearly decomposable", segregated, and autonomous[8].

The assortative community model has informed our current understanding of brain network function, perpetuating a stylized view of the brain in which segregated (i.e., assortative) communities engage in specialized information processing while a small number of highly connected hubs integrate information across communities[9]. This view is supported by cross-species analyses uncovering analogous structure in both human and non-human connectome data[10–12], suggesting that assortative communities may be an evolutionarily conserved architectural feature.

While this perspective has proven useful, it has a number of drawbacks, of which we focus on two. First, it makes the strong assumption that connectome meso-scale architecture is strictly assortative (Fig. 1a). This assumption stems in part from the algorithms used to detect communities, the most popular of which seek internally dense and externally sparse sub-networks[7]. As a result, these algorithms are incapable of detecting non-assortative structure, such as core-periphery (Fig. 1b) and disassortative (Fig. 1c) communities or mixtures of different community types (Fig. 1d), all of which are evident in real-world socio-technical and biological networks[13]. Moreover, modularity maximization and related techniques may overlook important and functionally relevant characteristics of neural circuits, which exhibit non-assortative, cell type-specific wiring diagrams[14,15]. It is unclear, then, whether the assortative communities uncovered using these algorithms represent an accurate picture of connectome meso-scale structure or whether they reflect the assumptions and limitations of the algorithms themselves.

Second, this view implies that the connectome's meso-scale structure is rigidly uni-functional. That is, networks with assortative communities are well-poised for specialized, segregated information processing, but are not suited for integrative function. Higher order cognitive processes, for example, are thought to emerge through integration of information originating in different brain systems[16], an integration that is thought to occur via the interaction of communities with one another. We hypothesize, then, that in order to produce complex thought and adaptive

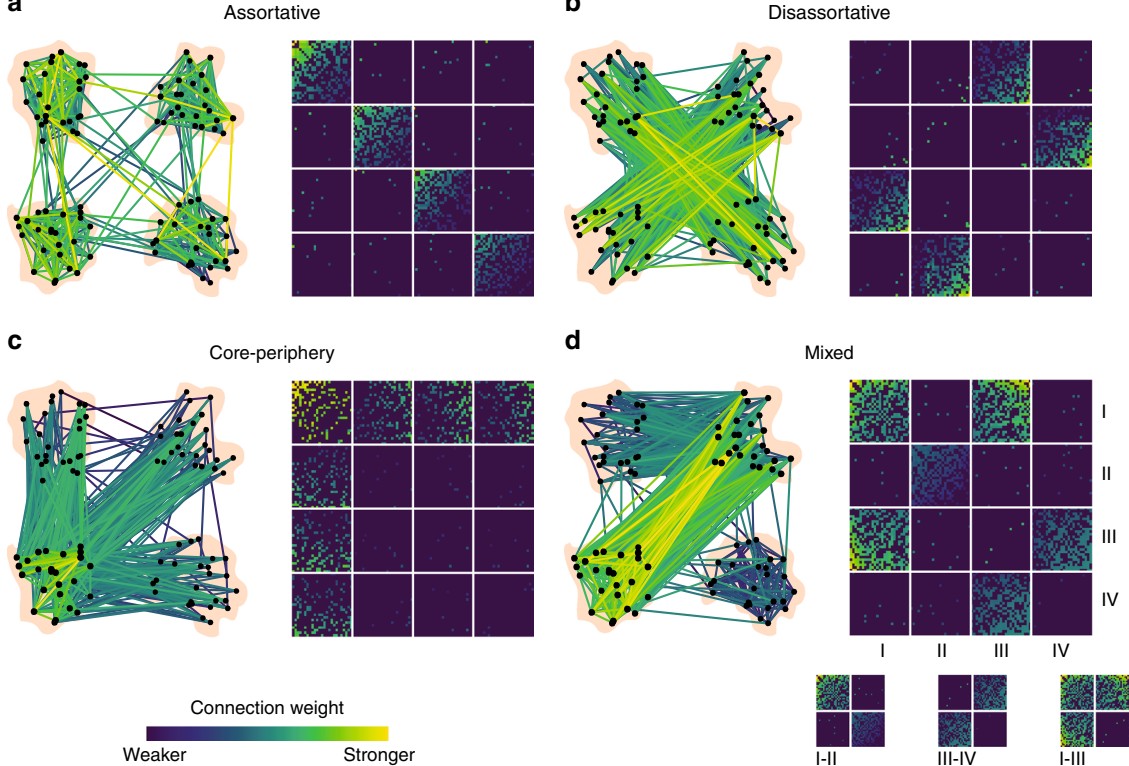

**Fig. 1** Community structure types. Networks can exhibit different types of meso-scale structure. **a** Assortative communities are sub-networks whose internal density of connections exceeds their external density. **b** Disassortative (multi-partite) communities are sub-networks where connections are made preferentially between communities so that communities' external densities exceed their internal densities. **c** Core-periphery organization consists of a central core that is connected to the rest of the network and then peripheral nodes that connect to the core but not to one another. **d** These meso-scale structures can be present simultaneously in the same network. For example, communities I–II interact assortatively, III–IV interact disassortatively, while I–III interact as a core and periphery

behavior, the brain's underlying meso-scale architecture must deviate (even if only slightly) from the strictly assortative model.

In this report, we address these hypotheses by using a flexible community detection method, the weighted stochastic blockmodel (WSBM), to uncover the meso-scale architecture of both human and non-human connectome data[17,18]. We show that, in addition to assortative communities described in previous reports, connectomes show evidence of non-assortative structure, including subsets of communities that interact disassortatively and others that form cores and peripheries. Next, we define a node-level diversity index that quantifies the extent to which individual neural elements participate in communities of all classes. We show that in humans, this index is peaked in regions associated with cognitive control and in sub-cortical areas, suggesting that traditionally defined cortico-subcortical circuits that support cognitive control are positioned to participate in a wide range of meso-scale processes. Finally, we show that diversity of connections in these same systems predicts individual differences in performance on two cognitive control tasks: namely, the Stroop[19] and Navon[20] tasks.

## Results

### The weighted stochastic blockmodel and connectome data sets.
We fit the WSBM to group-representative human connectome data. The WSBM assumes that a network's nodes can be partitioned into $K$ communities and that the weight and probability of a connection forming between two nodes are governed by parameterized generative processes. Critically, these processes depend only on the communities to which those nodes are assigned. Using the WSBM to uncover a network's community structure involves inferring both the parameters of these processes and the nodes' community assignments that maximize the log-evidence that the WSBM generated the observed network (see Methods for details on connectome reconstruction and the WSBM). The resulting communities, therefore, reflect similarities in nodes' connectivity profiles and are not constrained to be assortative, meaning the WSBM is capable of detecting disassortative and core-periphery structure.

As a point of comparison, we also obtained partitions using modularity maximization ($Q_{max}$)[7], a widely used community detection technique popular in network neuroscience[6]. Unlike the WSBM, $Q_{max}$ is designed to detect only assortative community structure. As both $Q_{max}$ and the WSBM algorithms are non-deterministic—i.e., repeated runs of the algorithm usually result in slightly different solutions—we varied the number of communities from $K=2$ to $K=10$ and repeated both algorithms 250 times for each $K$ (throughout this section we show that our results are robust over this range but focus, at times, on an intermediate number of communities, namely $K=5$).

In this section we report results using empirical human connectome data. To ensure that our results are not biased by a specific network reconstruction technique, we repeated all analyses using non-human data sets including mouse, rat, macaque, and *Drosophila* inter-areal connectomes (Supplementary Note 1; Supplementary Figs. 1–8). In addition, we repeated a subset of analyses using the network of neuron-to-neuron chemical synapses in the *C. elegans* connectome (Supplementary Note 2; Supplementary Figs. 9 and 10). We also present several confirmatory analyses that include demonstrating the convergence of the WSBM algorithm (in Supplementary Note 3 and Supplementary Figs. 11 and 12), a comparison of empirical networks to randomly rewired human connectome data (Supplementary Note 4; Supplementary Figs. 13–15), and evidence that reported results do not trivially depend on node definition (Supplementary Note 4; Supplementary Figs. 16 and 17).

### Connectomes support diverse meso-scale architecture.
The human connectome's ground truth meso-scale structure is unknown. This motivates studying alternative methods for uncovering communities and characterizing their similarities and differences. In this section, we compare communities obtained using two community detection methods: the WSBM and modularity maximization ($Q_{max}$) (Fig. 2).

We assessed the global dissimilarity of detected partitions using pairwise variation of information (VI)[21] (see Methods for details). Specifically, we computed pairwise VI among all 250 partitions detected using $Q_{max}$ and separately for all partitions detected using the WSBM. We also computed between-technique VI among all pairs of $Q_{max}$ and WSBM partitions. This process was repeated separately for different values of $K$, which helped make the comparison as fair as possible.

This procedure resulted in a sequence of within-technique and between-technique VI scores as a function of $K$ (Fig. 3a). At each $K$ we computed one-tailed $t$ tests of whether the mean within-technique dissimilarity of partitions detected with either the WSBM or $Q_{max}$ was smaller than the between-technique dissimilarity. We observed that from $K=2, \ldots, 9$, both the WBSM and $Q_{max}$ uncovered partitions that were self-consistent yet distinct from one another (maximum $p < 10^{-15}$). This observation was consistent across the non-human connectome data as well, confirming that the WSBM and $Q_{max}$ generate statistically dissimilar estimates of connectome community structure (Supplementary Fig. 2).

Next, we wished to confirm that the WSBM uncovered non-assortative communities, specifically. To test this hypothesis, we computed for each community $r$, its size, $N_r$, and assortativity score, $\mathcal{A}_r$ (Methods). We aggregated all detected communities and computed the mean assortativity score as a function of community size, $\overline{\mathcal{A}}(N)$ (Fig. 3b). These procedures were performed separately for the WSBM and $Q_{max}$. We compared these curves using functional data analysis, which is a set of statistical tools for comparing continuous curves[22]. Specifically, we computed the summed pointwise difference in both curves, which we treated as a test statistic. We found that the observed statistic was smaller than those obtained under a permutation-based null model ($p < 10^{-3}$), confirming that WSBM communities tend to be less assortative than $Q_{max}$ (Fig. 3c). Again, these findings are consistent across connectome data obtained from all species (Supplementary Fig. 3).

Finally, because the functional roles of brain regions also depend on local connections to their own and other communities, we tested whether our understanding of these roles changed when we considered WSBM communities rather than those uncovered using $Q_{max}$. Specifically, we examined how regional assortativity scores, a node-level metric analogous to community assortativity, differed given the WSBM versus $Q_{max}$ partitions (Methods; Fig. 3d, e). We found that regional assortativity decreased for most nodes. Aggregating differences by functional systems[23] (Methods), we found that the greatest decrements were concentrated within visual and somatomotor systems (Fig. 3f). Interestingly, decreased regional assortativity was also correlated with node degree (the number of connections a node makes), with low-degree nodes exhibiting greater decreases compared to high-degree nodes ($r = 0.37$, $p < 10^{-4}$; Fig. 3g). These findings were, overall, consistent in non-human connectome data sets. There were, nonetheless, some differences, which are discussed in Supplementary Note 1 (Supplementary Fig. 4).

In summary, these findings confirm that the weighted stochastic block model and modularity maximization uncover communities of fundamentally different nature. Among the most profound differences is the assortativity of detected communities, with the WSBM consistently detecting less assortative

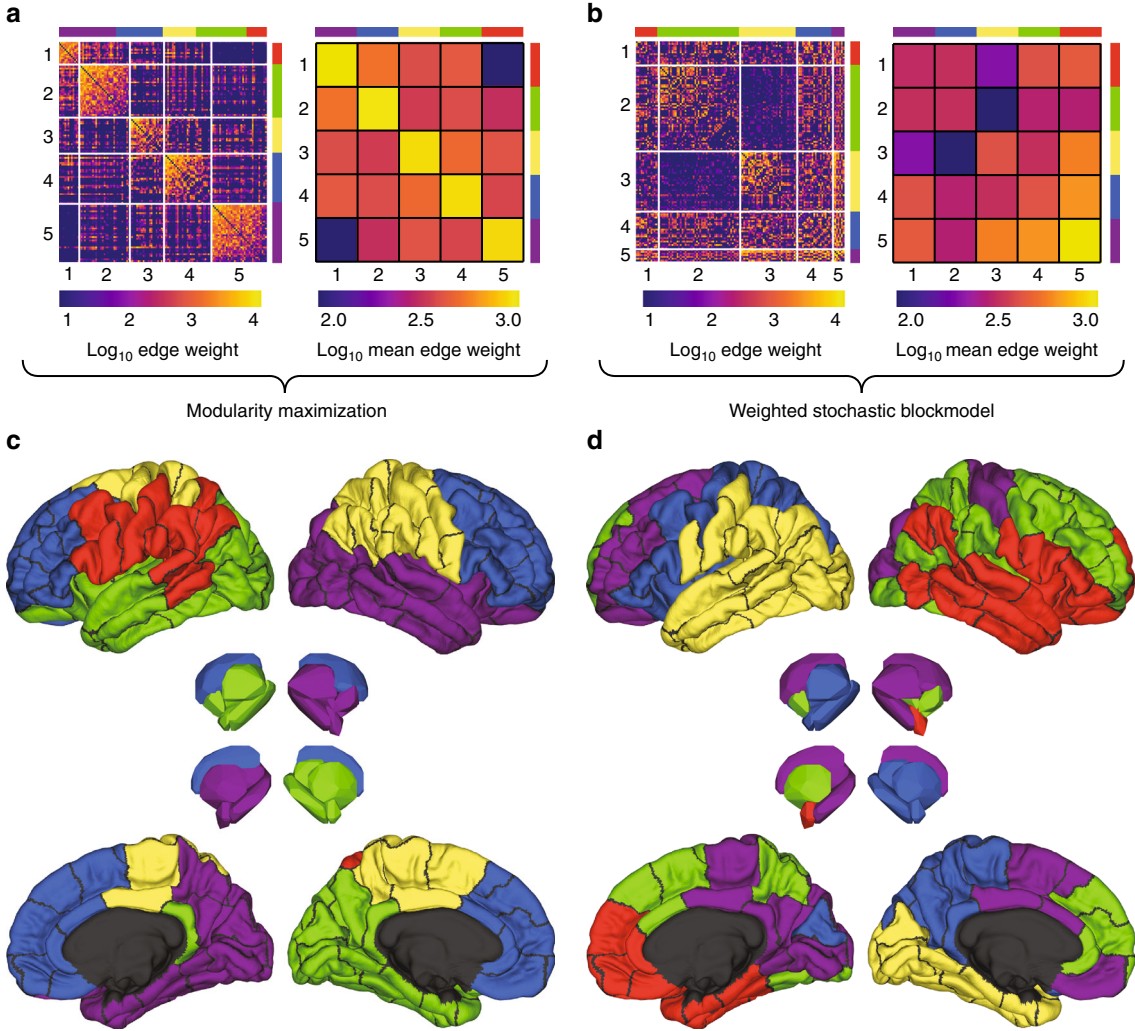

**Fig. 2** Example WSBM and $Q_{max}$ communities. Human connectome network ordered by community partitions detected using **a** $Q_{max}$ and **b** the WSBM. Both examples are shown with the number of communities fixed at $K = 5$. The color of matrix elements for the left sub-panels represents log-transformed edge weights while the color of matrix elements for the right sub-panels represents the log-transformed mean within-community and between-community edge weights. Panels **c** and **d** depict the spatial distributions of those same partitions

communities (i.e., less segregated and more integrated) than $Q_{max}$. The brain's meso-scale structure is generally assumed to be uniformly assortative (i.e., communities are segregated from one another), a feature thought to support specialized information processing[6]. The WSBM challenges this view, detecting less assortative (and hence increasingly integrated) communities, suggesting that communities might play a more diverse range of functional roles. Demonstrating this point empirically, however, remains a challenge.

**Many but not all communities are assortative**. In the previous section, we provided evidence that connectomes exhibit diverse, non-assortative communities. This finding, however, runs counter to the dominant narrative surrounding brain network function, namely that information processing is carried out by specialized, assortative communities. In addition, assortative communities are thought to confer many positive attributes to a network, including separation of dynamic timescales, efficient spatial embedding, and evolutionary robustness[6]. An important question, then, is whether the reduction in assortativity and increase in community diversity described in the previous section

are driven by a small subset of non-assortative communities (so that most communities are still assortative) or whether all communities uniformly decrease in assortativity.

To address this question, we uncovered the maximally assortative set of communities for each WSBM partition, which comprises the largest set of communities (in terms of the number of nodes included in those communities) whose minimum within-community density of connections exceeds its maximum between-community density (Methods). We then estimated how frequently, on average, each brain region participated in this set. We found that as we varied the number of communities from $K = 2$ to $K = 10$, the maximally assortative set comprised $75 \pm 13$ percent of all nodes (Fig. 4a). In general, the maximally assortative set was also comprised of low strength, non-rich club nodes (Fig. 4b, c; see Supplementary Fig. 18 for more details on nodes' assignments to the rich club). Breaking down inclusion in this set by cognitive system, we found that control and subcortical systems were the least likely to be included in the maximally assortative set compared to the other systems (Fig. 4d). As in the previous section, while we find similar results in non-human connectome data sets, we also note some differences (Supplementary Fig. 5). For instance, the *Drosophila* data set is unique in

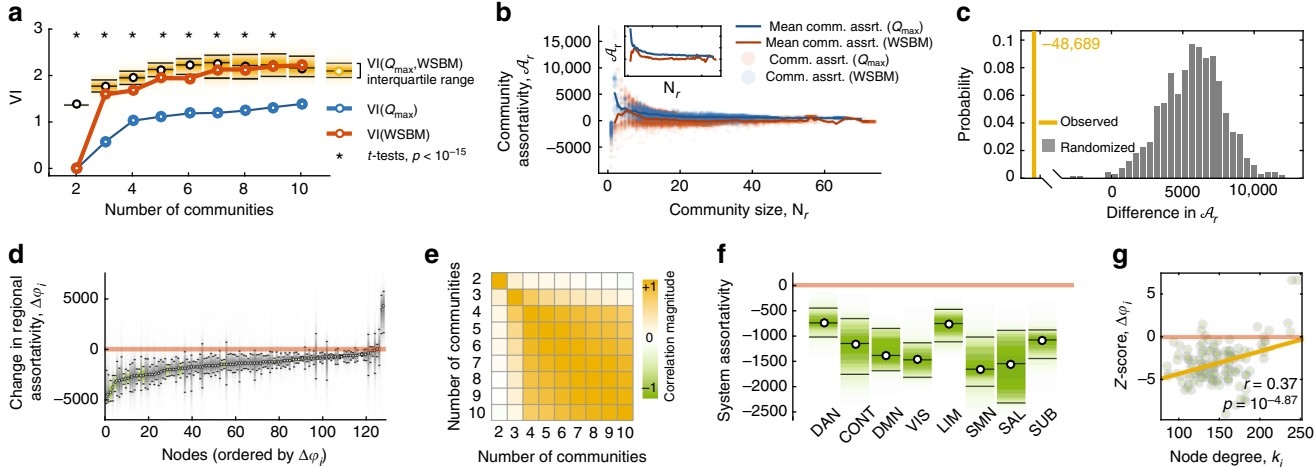

**Fig. 3** Modularity maximization and the weighted stochastic blockmodel uncover fundamentally different architectural signatures. **a** Variation of information within and across community detection techniques ($Q_{max}$ and WSBM) demonstrating greater dissimilarity of partitions between techniques than within techniques. **b** Community assortativity, $\mathcal{A}$, as a function of community size, $N$, demonstrating that $Q_{max}$ communities, on average, are more assortative (the inset shows the mean community assortativity curves as a function of distance). Note: asterisks indicate that both $t$ tests were statistically significant. **c** Comparison of statistic derived using functional data analysis (yellow line) with that expected in a null distribution. Specifically, we generated a statistic by performing a pointwise subtraction and summation of the curves $\overline{\mathcal{A}}(N)$ obtained for the WSBM and $Q_{max}$. The value of this statistic quantifies the difference between mean community assortativity across communities of all sizes and is negative when communities detected using $Q_{max}$ are more assortative than WSBMs. We compared this statistic against a null distribution obtained from a null model wherein we preserved the number and size of communities in a given partition but permute nodes' assignments uniformly and randomly (1000 repetitions). **d** Changes in regional assortativity, $\Delta a_i$, when considering WSBM versus $Q_{max}$ partitions, ordered by greatest to least decrease. Note that the majority of regions decrease assortativity in the partitions estimated from WSBM compared to those estimated from $Q_{max}$ (i.e., $\Delta\phi_i < 0$). **e** Correlation of regional assortativity while varying the number of communities from $K = 2,...,10$. Note the high consistency for $K \geq 4$. **f** Regional assortativity scores grouped by cognitive systems: DAN, dorsal attention; CONT, cognitive control; DMN, default mode; VIS, visual; LIM, limbic; SMN, somatomotor; SAL, salience; SUB, subcortical. The limits of each box represent the interquartile range (25th and 75th percentiles). **g** Regional assortativity (corrected for degree through standardization procedure) as a function of node degree, $k_i$.

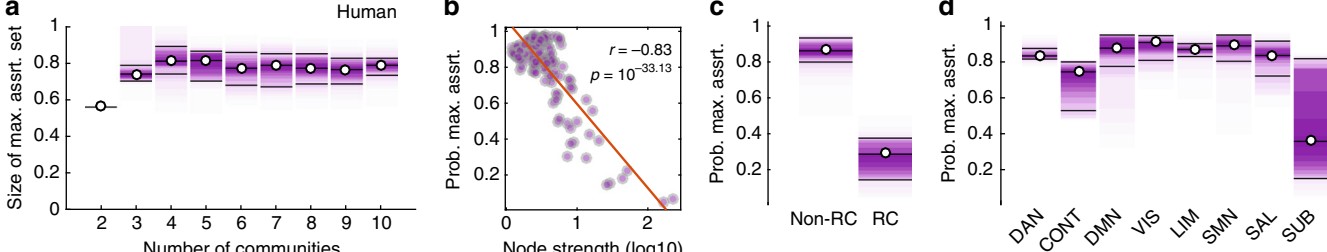

**Fig. 4** Maximally assortative set. **a** Fraction of brain regions comprising the maximally assortative set as a function of the number of communities. **b** High strength nodes are less likely to participate in the maximally assortative set. **c** As a consequence, the maximally assortative set is comprised mostly of non-rich club brain regions; RC, rich club. **d** At the system level, control and subcortical networks are the least likely to participate in the maximally assortative set. DAN, dorsal attention; CONT, cognitive control; DMN, default mode; VIS, visual; LIM, limbic; SMN, somatomotor; SAL, salience; SUB, subcortical

that the relationship of node strength and the probability of being assigned to the maximally assortative set exhibits a u-shaped curve (Supplementary Fig. 5f). The macaque data set exhibits a similarly shaped curve, and possibly as a consequence of where we drew the cutoff for rich club assignment or the incompleteness of the macaque connectome, rich club nodes are actually more likely to be assigned to the maximally assortative set than non-rich club nodes (Supplementary Fig. 5o).

In summary, these findings confirm that while the WSBM tends to detect less assortative communities than $Q_{max}$, there nonetheless exists a backbone of highly assortative communities that, as a group, exhibit the ubiquitous internally dense, externally sparse connection density. This collection of communities, which largely excludes the brain's highly connected regions, therefore has the capacity to perform segregated information processing.

**Functional relevance of the WSBM**. It is generally agreed upon that structural connectivity in the brain determines the partners that any given region can "talk to", and therefore constrains communication patterns among brain regions, shaping the correlation pattern of ongoing neural activity, i.e., the functional network organization. We reasoned that if two brain regions receive input from the same set of brain regions and deliver output to the same set of regions, then their activity over time should be correlated, i.e., those regions would appear functionally connected to one another. This set of assumptions has a long tradition in the network neuroscience community. In the past, before it was common to empirically estimate FC as the correlation of neural activity, measures of similarity between brain regions' connectivity profiles (e.g., matching index) have been used as a stand-in ref.[24]

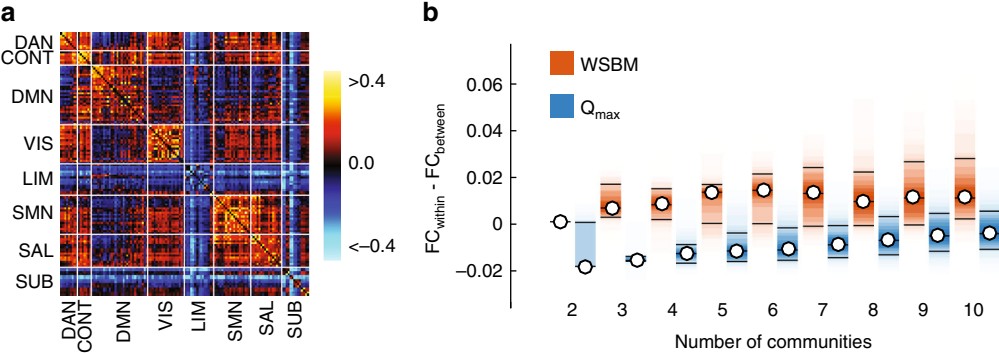

**Fig. 5** Communities estimated from the weighted stochastic block model are more functionally segregated than communities estimated from modularity maximization. **a** Functional connectivity (FC) matrix ordered by functional system: DAN, dorsal attention; CONT, cognitive control; DMN, default mode; VIS, visual; LIM, limbic; SMN, somatomotor; SAL, salience; SUB, subcortical. Note that the order of nodes shown in this panel does not correspond to partitions generated by either the WSBM or $Q_{max}$. **b** Difference between within-community and between-community FC for the WSBM (orange) and $Q_{max}$ (blue). Each box plot depicts the variance over partitions detected using either the WSBM or $Q_{max}$

Though via different mechanisms, both the WSBM and $Q_{max}$ produce communities composed of brain regions with similar patterns of incoming and outgoing connections and so we would expect the resulting communities to be internally dense in terms of functional connectivity. In the case of $Q_{max}$, this similarity is entirely incidental—nodes get grouped into internally dense, mutually connected clusters, inflating their similarity. The WSBM, on the other hand, explicitly defines communities as clusters of nodes whose connections were generated by the same statistical process; by definition, pairs of nodes in the same community will have similar connectivity patterns even if they, themselves, are not directly connected.

Because the similarity of regions' structural connectivity is associated with strong functional connectivity, we expect that two nodes in the same community should be more strongly functionally connected to one another than two nodes in different communities, irrespective of which technique was used to define the communities. However, the WSBM and $Q_{max}$ represent vastly different hypotheses about how brain networks function. An assortative brain is aligned with the hypothesis that communities function and process information relatively independently from one another, while a brain that allows for some non-assortative communities implies that function arises not solely from contributions of independent communities, but from the interactions between communities. Whereas past work has emphasized the assortative model of brain function, in which integration is performed by a few outlying nodes whose connections span community boundaries, the non-assortative model holds that integration is fundamentally a community-level action performed by clusters of brain areas with similar (non-assortative) connectivity profiles.

We can compare these two hypotheses of brain function with cross-validation methods using empirical functional connectivity as metadata[25,26]. We reasoned that if functional connectivity emerges from interactions among brain regions in independent, autonomous clusters, then its organization will be closely aligned to the communities detected using $Q_{max}$. On the other hand, if functional connectivity is the result of non-assortative, integrated clusters, then the WSBM communities will more closely resemble the brain's functional connectivity. To compare communities with functional connectivity, we classified every functional connection as "within-community" or "between-community". We calculated the mean weight of all connections assigned to each class and finally the difference between those values. This measure—the difference between mean within-community and

between-community functional connections—serves as a measure with which we can evaluate the performance of the two algorithms.

We found that over a range $K = 2, ..., 10$, the WSBM consistently uncovered communities whose internal FC density exceeded their between-community density (Fig. 5a). Moreover, the difference of within-community and between-community FC density was greater using the WSBM communities than using $Q_{max}$ communities ($t$ tests, $p < 0.01$; Fig. 5b), suggesting that the WSBM communities capture functional relationships among brain regions. We also report consistent findings when we apply the same methodology to correlated gene expression patterns for the mouse connectome (Supplementary Fig. 8). These findings show that WSBM communities are more closely aligned with human FC than $Q_{max}$ communities, informing our understanding of the role played by non-assortative communities in shaping the correlation structure of correlated neural activity.

Although the results of this section suggest that the WSBM is closely aligned with human FC, we report several caveats. First, our analysis assumes a close relationship between FC and the underlying structure. While structure constrains FC, the mapping between the two is imperfect and fluctuates over shorter timescales[27] and can vary when different measures of FC are used. The use of a Pearson correlation, for example, induces transitive functional connections by placing statistical bounds on correlations among triplets of nodes[28]. This implies that the correlation values are not independent, which may influence our estimates of mean within-community and between-community FC magnitudes.

**Community morphospace reveals rules for between-community interactions.** To this point we have shown that the WSBM uncovers rich, non-assortative meso-scale structure and that, compared to $Q_{max}$, these communities are more functionally and genetically segregated in human and mouse connectomes, respectively. In this section, we seek a fundamental understanding of the exact nature of that mesoscale architecture, and therefore ask the question: "How do interactions among pairs of communities combine to generate assortative and non-assortative meso-scale architecture?" To address this question, we focus our analysis onto the interactions among pairs (dyads) of communities. Community dyads represent the building blocks of a network's meso-scale structure, and can be combined in different configurations and proportions to engender larger, more complex

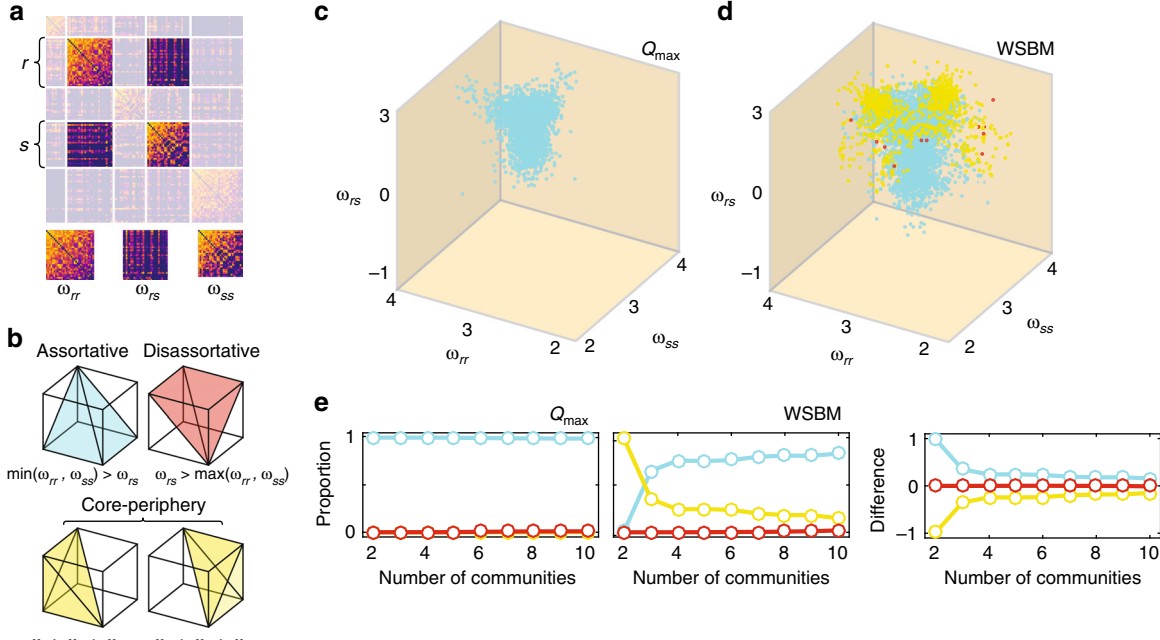

**Fig. 6** A rich community morphospace. **a** A community motif is constructed on the average connection weight over blocks of the connectivity matrix. Here, we show blocks within and between two communities, labeled r and s. **b** Given within-community and between-community connection densities, it is possible to classify each pair of communities into one of three motifs: assortative, disassortative, or core-periphery. **c, d** All pairs of communities placed in a network morphospace and colored by their motif type (note: axes are log-scaled). **e** The relative proportion of each motif type as a function of the number of detected communities, $K$, for $Q_{max}$ (left), the WSBM (middle), and their difference (right). Note: The WSBM does, in fact, generate a small fraction of disassortative communities and so points on the red curves in D and E are not equal to zero

functional circuits. We investigate community interactions using a theoretical morphospace analysis[29], a technique recently adapted to the study of complex networks[30].

A morphospace is a hyperspace whose axes are features of a particular class of organism or system. In the case of complex networks, axes usually are defined to be topological properties of a network, e.g., efficiency, wiring cost, modularity, or the parameters of generative network models. Once the axes are defined, any observed network can be represented in the morphospace as a point whose location is defined by that network's particular combination of features. In general, morphospaces are not uniformly populated. Evolutionary and functional constraints render some regions more favorable (and hence more densely populated) than others, and by studying the density of points throughout the morphospace one can better understand how those constraints influence the structure of a network.

Here, rather than constructing a morphospace of networks, we construct a three-dimensional community morphospace, allowing us to investigate how interactions among pairs of communities combine to generate assortative and non-assortative meso-scale architecture. In this morphospace, each point represents a pair of communities, $\{r, s\}$, and the axes are defined to be their respective within-community and between-community connection densities, $\omega_{rr}$, $\omega_{ss}$, and $\omega_{rs}$ (Fig. 6a). These features can be used to classify the interaction of r and s into one of three canonical community interaction motifs: assortative, core-periphery, and disassortative (Methods for details) (Fig. 6b).

We compared morphospaces constructed based on communities detected using the WSBM and $Q_{max}$, and computed the relative proportion of each motif type (Fig. 6c, d). Across $K = 2, \ldots, 10$, we found that $Q_{max}$ partitions resulted in, almost exclusively, assortative interactions among communities (Fig. 6e).

The WSBM also favored assortative interactions, but included a significant number of core-periphery and disassortative interactions. Again using functional data analysis, we compared the relative proportion of each motif by performing a pointwise subtraction and then summation of each motif's relative proportion as a function of $K$ and aggregated the motif-specific scores to generate a statistic (Fig. 6e). This statistic measured the absolute difference in relative motif proportion as the number of communities varied from $K = 2$ to $K = 10$. We compared this statistic against a null distribution generated by randomly and uniformly permuting nodes' community assignments and recalculating motif proportions. We found that the observed difference exceeded what would be expected by chance ($p = 0.029$; 1000 permutations), indicating that the relative proportion of community motifs discovered using the WSBM was different than $Q_{max}$ and could be largely explained by increased diversity of motif types using the WSBM. Again, these findings were largely replicated in non-human connectome data, though the relative proportions of motif types was variable (Supplementary Fig. 6). While the incompleteness of the non-human connectome data sets make cross-species comparisons difficult, these differences raise the prospect that the meso-scale structure of different organisms features nuanced, organism-specific motifs.

**Community motifs identify a class of diversely connected nodes**. Community motifs represent interactions among pairs of communities. We mapped motifs back to the level of individual brain regions by computing a motif participation index for each brain region, which measured the fraction of times the community that region was assigned to participated in each motif. Importantly, for the core-periphery motif, we distinguished between the "core" community and the "periphery" community. In addition, we also computed a diversity index: an entropy over

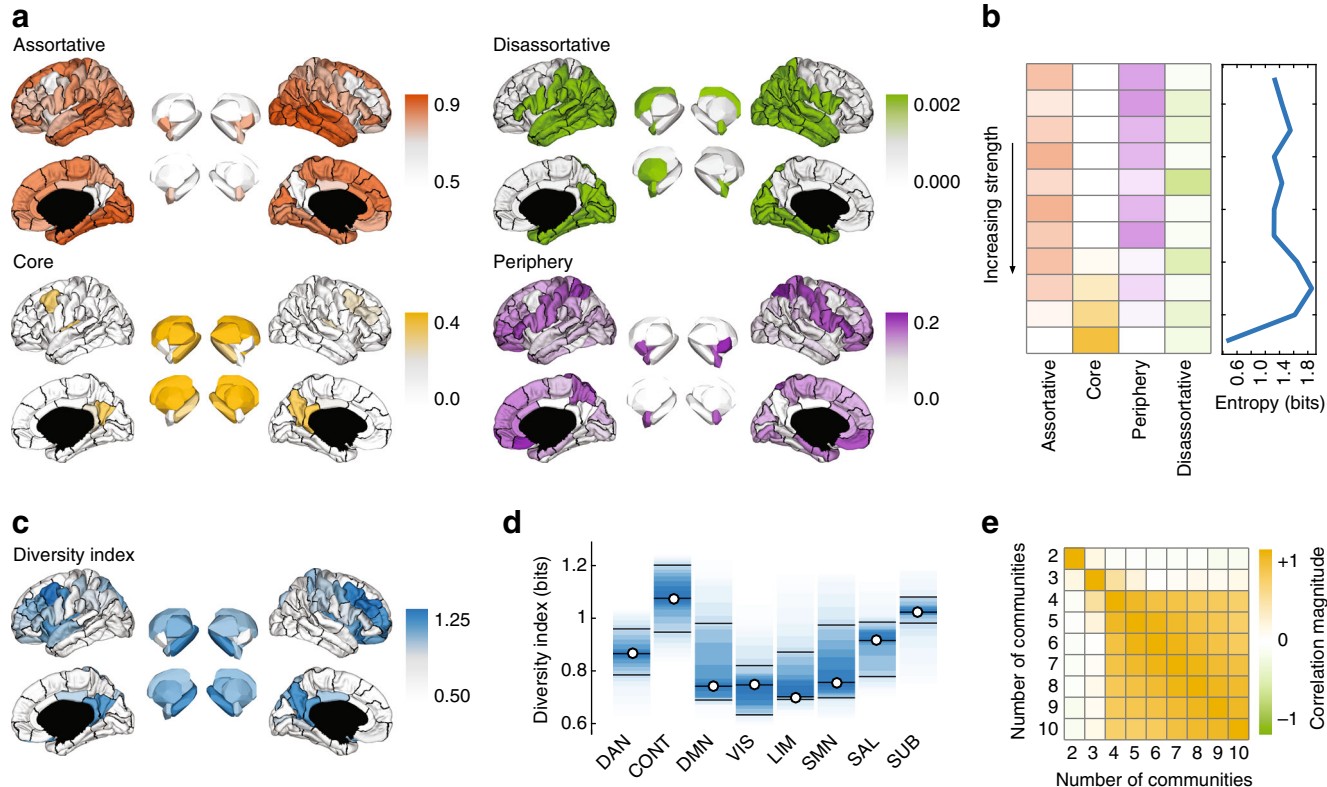

**Fig. 7** Regional variation in motif participation highlights diversely connected nodes. **a** Regional participation in the four community motif types. Note that the scales vary from panel to panel. **b** Dominance of each motif type as a function of node strength (weighted degree). Note that motif dominance varies with strength; high-strength nodes are predominantly located in the core while low-strength are assortative and predominantly located in the periphery. **c** Diversity index measuring the entropy across each node's motif participation. **d** Diversity index grouped by functional system. **e** Correlation of regional diversity indices as a function of the number of detected communities, $K$, demonstrating robustness of results across choice of $K$

the motif participation distribution (Methods: "Diversity index"). A region that participated largely in one motif type had lower diversity than a region that participated equally in all motif types.

As expected, motif participation was heterogeneous across brain regions. Core participation, for example, was dominated by highly connected sub-cortical regions as well as the precuneus, insula, and frontal cortices. This agrees with our understanding of those regions as being highly central in the network, with the capacity to exert influence and regulate information flow across the brain[9]. Participation in assortative motifs, on the other hand, was dominated by middle temporal, lateral occipital, and fusiform cortices (Fig. 7a). Interestingly, participation was stratified by node strength (Fig. 7b). Binning brain regions by their strengths, we found that low-strength bins were composed of nodes that participated predominantly in assortative and periphery motifs while high-strength bins included nodes that participated almost exclusively in core motifs. By quantifying the diversity within each bin as an entropy, we found that both the low-strength and high-strength regions were among the least diverse—they participated in a narrow, well-defined set of motifs. The set of regions with the most diverse motif participation were those with above average but never the greatest strength (Fig. 7b). This finding was also observed in the non-human connectome data sets (Supplementary Fig. 7).

These results suggest an important functional role for middle-strength brain regions. While both high-strength and low-strength regions are highly stereotyped in terms of the range of motifs in which they participate, middle-strength brain regions are among the most diverse, participating in all motif types nearly equally, and hinting at the capacity for enhanced functionality.

This is not to diminish the putative functional roles of high-strength and low-strength regions, which have the capacity to readily exert influence and be influenced, respectively, but only to suggest that middle-strength nodes might have the ability to do both. Based on these findings, we hypothesized that the diversity of communities in which a region participates is related to its functional repertoire, with increased diversity corresponding to a broader range of functions. We further hypothesized that polymodal association areas, because they participate in a range of cognitive processes and require the synthesis of sensory information, attentional resources, and control mechanisms, would be among the most diverse.

To test this hypothesis, we computed a region-level diversity index (Fig. 7c), which we aggregated by functional system. As expected, the most diverse regions were concentrated within control and subcortical networks (permutation tests; $p = 0.001$) (Fig. 7d), and these results were robust across a range of values for the number of communities, $K$ (Fig. 7e). The cognitive control network includes some of the brain's more recently evolved cortical structures[31], which are thought to play critical roles across a multitude of executive functions[32], while the sub-cortex contains many nuclei responsible for performing distinct functional and regulatory roles[33,34]. Note that because $Q_{max}$ uncovers only assortative community motifs, each brain region's diversity score is effectively zero. Accordingly, we never assessed the distribution of diversity scores for the $Q_{max}$ partitions over functional systems.

**Behavioral relevance of motif diversity**. In the previous section, we demonstrated that the most diverse brain regions, in terms of

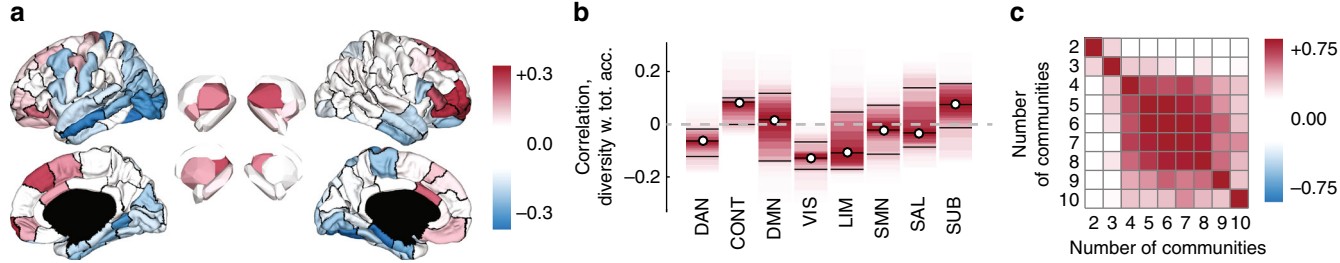

**Fig. 8** Diversity index correlates with individual differences in performance on tasks demanding cognitive control. **a** Regional correlation coefficients of total accuracy with diversity index on the brain surface (shown with $K = 5$). Areas in white did not show a significant correlation after FDR correction for multiple comparisons. **b** Regional correlation coefficients grouped according to functional system: DAN, dorsal attention; CONT, cognitive control; DMN, default mode; VIS, visual; LIM, limbic; SMN, somatomotor; SAL, salience, and SUB, subcortical. **c** Similarity of regional correlation coefficients as a function of the number of communities, indicating robustness to the choice of $K$ when $4 \leq K \leq 10$

their community motif participation, include control and subcortical regions. We speculated that this diversity might represent a neuroanatomical, network-level substrate for flexible cognitive behavior. In this section, we test whether inter-individual differences in regional diversity can account for behavioral variability.

Specifically, we asked 30 subjects to perform canonical cognitive control tasks (the Stroop and Navon), which require the rapid interactions of visual, attentional, and executive control systems (see Methods for task details). We combined total accuracy from both tasks to generate a composite accuracy score for each subject, which measured their performance. We hypothesized that individuals with greater diversity would perform better on both tasks than individuals with little diversity, suggesting that brain networks configured to facilitate integration across many types of meso-scale organization can more effectively exert control over processes requiring many complex representations and discriminative judgments.

To this end, we fit the WSBM to subjects' connectome data while varying the number of communities over the range $K = 2, \ldots, 10$, classified community motifs, and computed motif participation scores and diversity indices for each brain region. As an additional step, we partialed out the effect of subjects' total connection weight over the whole brain from the regional diversity indices. We then computed the Spearman correlation of total accuracy with the residuals, resulting in a correlation coefficient for each brain region. In Supplementary Note 5 we show that we get similar results using different parcellation schemes (Supplementary Figs. 16 and 17) and without partialing out total connectome weight (Supplementary Fig. 19).

Interestingly, we found that the strongest correlations (both positive and negative) were distributed heterogeneously across the brain (Fig. 8a), but also tended to cluster within a few cognitive systems. The strongest positive correlations belonged to regions that were associated with cognitive control (permutation test; $p = 0.031$) and sub-cortical systems ($p = 0.015$), while the visual system was more anti-correlated than expected ($p < 10^{-3}$; all tests FDR corrected for multiple comparisons) (Fig. 8b). Moreover, when the number of detected communities was greater than $K = 4$, this pattern remained largely unchanged (Fig. 8c). Generally, these findings posit a link between individual differences in behavior and regional variation in motif diversity. More specifically, they also implicate control and subortical systems, a finding reminiscent of the corticostriatal loops thought to play an important role in control-oriented behavior[35].

## Discussion

In this report, we hypothesized that the view of the connectome as being comprised of segregated communities is one that

overlooks, for methodological convenience and ease of interpretation, competing and equally plausible accounts of its meso-scale architecture. These alternatives allow for the possibility of heterogeneous community configurations, including cores and peripheries and disassortative motifs. Using the WSBM, which belongs to a class of community detection algorithms sensitive to both assortative and non-assortative communities, we presented evidence supporting the existence of such motifs. Moreover, by cross-validating communities using metadata, we showed that meso-scale structure uncovered by the WSBM was more closely aligned with functional connectivity compared to $Q_{max}$. We then observed that the extent to which brain regions participated in particular classes of motifs varied across the brain, with the greatest degree of diversity concentrated in control and subcortical systems. This prompted us to formulate the hypothesis that participating in a diverse set of communities engenders a broader functional repertoire and flexible cognitive behavior. We tested this hypothesis using single-subject connectome data and behavioral scores recorded during cognitive control tasks. We found that regional diversity within both control and subcortical systems was predictive of subjects' task performance, supporting our hypothesis that structural diversity is relevant for human behavior.

A central aim of biology is to understand how a system's form and function are related. In neuroscience in particular, evidence across spatial scales points to the critical importance of structural links between neural elements in predicting the function of a circuit or motif[5]. For example, synapse formation dynamics shape distributed synchronization patterns[36], offering a mechanistic explanation of sequential activity underlying motor gestures and memory[37]; evidence from theoretical and computational studies suggests that neural sequences may be shaped by synaptic constraints and network circuitry rather than time constants[38]. At the small-scale of neuronal circuits, these studies collectively highlight topological rules and structural motifs that explain observed function: from feedforward loops to repressor lattices[39]. Yet obtaining similar insights at the large-scale of whole-brain circuits has proven challenging, in part due to incomplete understanding of which motifs matter at this scale. Our work directly addresses this gap by offering novel concepts to define structural motifs in the meso-scale architecture of connectomes, tools for estimating these motifs from available data, and a proof-of-concept demonstration that the diversity of such motifs can be used to understand individual differences in a cognitive process that requires computations in large-scale distributed circuits (cognitive control[40]).

In seeking large-scale structural motifs, we shifted focus away from regional and whole-brain organization, and onto the brain's meso-scale structure. Meso-scale analysis is a coarse-graining of a

network, analogous to dimension reduction. While the modest-sized networks studied here benefit from such an analysis, this approach may see broader applicability in the future, where advances in connectome imaging and reconstruction techniques have resulted in high-dimensional data sets[41,42]. Making sense of such data while still respecting its underlying relational structure—parsimoniously encoded as a network or graph—is a major computational challenge. To comprehend the organization of connectome data, especially at the cellular scale, may require dimension reduction techniques like community detection that can distill the important organizational principles from those that are less useful. Modularity maximization and related techniques may miss out on functionally relevant characteristics of complicated neural circuits, which exhibit non-assortative, cell type-specific wiring[14,15]. Our study represents one of the first to explore the utility of blockmodels in conjunction with structural connectome data (though others have investigated blockmodels in the context of functional connectivity[43–46]). We demonstrate the potential benefits of this approach, linking blockmodels to behavior, functional connectivity (in the human), and gene co-expression (in the mouse). Future studies may extend these approaches to the study of neurodevelopment or psychiatric disease and disorders.

Our work refines the widely held view of the brain as being composed of segregated communities interlinked by integrative hubs[9]. While most communities detected using the WSBM engaged in assortative interactions, others engaged in core-periphery and disassortative interactions. This diversity of inter-connection types suggests that the integrative units of the brain are not necessarily highly connected hubs, but groups of brain regions with similar connectivity profiles, weight distributions, and shared functional capacity[5]. Importantly, these findings were replicated in non-human connectome data sets, suggesting that non-assortative meso-scale architecture is not unique to *Homo sapiens* and is, likely, not an artifact attributable to a specific connectome reconstruction technique.

These findings build upon and extend other recent studies reporting non-assortative structure in connectome data. The mammalian visual system for instance, exhibits feed-forward like structure at both the inter-areal[47] and cellular levels[48]. A previous analysis of *C. elegans*' meso-scale structure using mixture models (a relative of the WSBM) revealed a core-like community composed of highly connected inter-neurons that play critical roles in mechanosensation and locomotion, highlighting its apparent role in the control of behavior[49]. Similarly, mouse, rat, and macaque connectomes exhibit core-periphery organization, where the core is composed of associative brain areas and proposed to act as a "pacemaker"[50]. Moreover, this type of architecture is consistent with wiring-cost reduction models, suggesting that core-periphery structure, like assortative communities, can be efficiently embedded in three-dimensional space[51].

The richness of assortative, disassortative, and core-periphery interactions alters the picture of possible computations that the brain can support. Indeed, that picture morphs from a simple egalitarian description in which each cognitive system acts equally and independently, into a more varied landscape that could possibly support top-down[52] and bottom-up influence[53], hierarchies of processes[47], and repertoire diversity, all canonical features of neural dynamics observed in empirical studies across spatial scales and species.

Our analyses uncovered an unexpectedly critical role of the topological "middle class"—brain regions with above average but never the greatest strength displayed the most diverse motif participation. This finding joins a series of recent examples highlighting the importance of non-hub and non-rich-club brain regions. Several recent functional studies have demonstrated the utility of low-strength nodes in explaining individual differences in fluid intelligence[54] and alterations in functional brain dynamics in psychiatric disease[55]. Complementary studies of human and non-human primate structural connectivity suggest that low-strength nodes have the capacity to drive neural dynamics into distant target states[56]. Yet, the role of middle-strength nodes has not been broadly studied, and their architectural characteristics are not well understood. Our finding that middle strength nodes display diverse participation in motifs suggests enhanced functionality: the capacity to both readily exert influence on and be influenced by others[56]. While we provide an empirical observation of this diversity, and initial evidence supporting its role in cognitive function (see next section), our understanding could benefit from future work exploring mathematical models of neural dynamics that explicitly account for these observations.

We demonstrated that individual differences among brain regions' diversity was correlated with subjects' performances on Stroop and Navon tasks. Moreover, the strongest correlations were associated with the systems whose diversity was, on average, the greatest. This finding suggests that diverse motif participation, which we speculate allows regions to engage in a wider range of function, represents a neuroanatomical substrate for flexible cognitive behavior[32]. This finding corroborates past studies showing that the brain's control systems reorganize the flexible reconfiguration of brain FC in adapting to complex tasks[57] and agrees closely with what is already known about their circuitry; cortico-subcortical loops are believed to play important roles in supporting cognitive control processes[58]. In our work, these systems are also highlighted as the most diverse in terms of their motif participation, suggesting that in addition to their traditional roles in cognitive control, they may also have the capacity to perform manifold functional roles, including the support of polysensory integration and association as well as exerting control over those processes. Future studies can explicitly examine whether this meso-scale anatomical organization guides functional network organization such as hierarchies from sensory to association to control regions[59].

With the rapid acquisition of large connectomic data sets, it is becoming increasingly urgent to develop and share computational tools for studying complex systems. Here, we made several methodological innovations in the form of novel network-based metrics, including assortativity scores, community morphospace analysis and motif classification, and node-level motif participation and diversity indices. With the exception of the diversity index, our exploration of these measures was largely theoretical. Future studies can capitalize on these theoretical advances by comparing their values in healthy and diseased conditions[60], across the human lifespan[61], or as a function of cognitive or behavioral state[62]. Moreover, the tools themselves are agnostic to the exact nature of the network under study, and may therefore also prove useful in understanding meso-scale organization in non-neural networked systems.

It is important to note a few methodological limitations of this study. First, the WSBM requires that the user specify the number of communities, $K$, for which there is no agreed upon method[63]. Accordingly, we never focused on a particular value of $K$, but showed that our results are robust over a reasonable choice of $K$. Another limitation, especially with the human connectome data, is the verisimilitude of the reconstructed network. Diffusion imaging and tractography algorithms are prone to inaccurate reconstructions that limit their utility for connectome inference[64]. Despite these shortcomings, tractographic reconstructions of fiber bundles have been incorporated into neurosurgical planning[65], suggesting that the accuracy of tractography should be evaluated in specific contexts. Moreover, hardware advances and a new

generation of ensemble[66] and global reconstruction[67] techniques offer the possibility of improved estimates. In the context of this discussion, however, it is also important to point out that our results in the human were all confirmed in the non-human connectomes as well, which are constructed from inherently different sorts of empirical data. The reliability of our findings across *Drosophila*, mouse, rat, macaque, and human suggest that they cannot be accounted for by deficiencies in any one data modality.

In this work, we sought to understand the structural basis for cognitive computations. We hypothesized that diverse meso-scale structure allows a network to engage in a wider functional repertoire, and that inter-subject variability in diversity is predictive of variation in cognitive performance. To address these hypotheses, we applied a WSBM to the connectome data acquired from five different species (*Drosophila*, mouse, rat, macaque, and human). We showed that the communities it detects are different from those commonly discussed in the literature, and that they provide statistically better explanations of resting state functional connectivity in the human and gene co-expression in the mouse. Finally, we showed that a diversity metric derived from those communities predicts behavioral outcomes in cognitive control tasks. Collectively, this body of work provides an alternative view of the structural substrate for computations in large-scale distributed circuits, and opens up new avenues of inquiry into the development and evolution of this architecture.

## Methods

**Connectome data sets**. A connectome refers to the complete set of neural elements and the physical connections that link those elements to one another[3]. In the main text we analyze human connectome data. In the supplement, we repeat those analyses using previously published connectome data representative of five different species: *Drosophila*, mouse, rat, macaque, and human. In this section, we offer brief descriptions of the methodologies used to reconstruct human connectome data. More details on the nun-human data sets are provided in Supplementary Note 1.

**Human connectome data set**. We analyzed both individual and group-representative, whole-brain networks generated by combining single-subject data from a cohort of 30 healthy adult participants. Each participant's network was reconstructed from diffusion spectrum images (DSI) in conjunction with state-of-the-art tractography algorithms to estimate the location and strength of large-scale interregional white-matter pathways. Details of the acquisition and reconstruction have been described elsewhere[68] but are repeated here for the sake of completeness.

DSI were acquired for a total of 30 subjects along with T1-weighted anatomical scans. We followed a parallel strategy for data acquisition and construction of streamline adjacency matrices as in previous work. DSI scans sampled 257 directions using a Q5 half-shell acquisition scheme with a maximum *b*-value of 5000 and an isotropic voxel size of 2.4 mm and an axial acquisition with the following parameters: repetition time = 5 s, echo time = 138 ms, 52 slices, field of view (231, 231, 125 mm). All procedures were approved in a convened review by the University of Pennsylvania's Institutional Review Board and were carried out in accordance with the guidelines of the Institutional Review Board/Human Subjects Committee, University of Pennsylvania. All participants volunteered with informed consent in writing prior to data collection.

DSI data were reconstructed in DSI Studio (www.dsi-studio.labsolver.org) using *q*-space diffeomorphic reconstruction (QSDR)[69]. QSDR first reconstructs diffusion-weighted images in native space and computes the quantitative anisotropy (QA) in each voxel; the image is then warped to a template QA volume in Montreal Neurological Institute (MNI) space using the statistical parametric mapping nonlinear registration algorithm. Once in MNI space, spin density functions were reconstructed with a mean diffusion distance of 1.25 mm using three fiber orientations per voxel. Fiber tracking was performed in DSI studio with an angular cutoff of 55°, step size of 1.0 mm, minimum length of 10 mm, spin density function smoothing of 0.0, maximum length of 400 mm and a QA threshold determined by DWI signal in the colony-stimulating factor. Deterministic fiber tracking using a modified FACT algorithm was performed until 1,000,000 streamlines were reconstructed for each individual.

Anatomical scans were segmented using FreeSurfer59 and parcellated using the connectome mapping toolkit[70]. A parcellation scheme including *n* = 128 regions was registered to the B0 volume from each subject's DSI data. The B0 to MNI voxel mapping was used to map region labels from native space to MNI coordinates. To extend region labels through the gray-white matter interface, the atlas was dilated by 4 mm[71]. Dilation was accomplished by filling non-labeled voxels with the

statistical mode of their neighbors' labels. In the event of a tie, one of the modes was arbitrarily selected.

Based on the division of the brain into regions, we constructed for each individual an undirected and weighted connectivity matrix, $A \in \mathbb{R}^{N \times N}$, whose edge weights were equal to the number of streamlines detected between regions $i$ and $j$ normalized by the geometric mean of their volumes: $A_{ij} = \frac{S_{ij}}{\sqrt{(V_i V_j)}}$.

Each individual's resulting network was undirected (i.e., $A_{ij} \left(\frac{V_i V_j}{V_j}\right) A_{ji}$) with density and mean node strength of $d = 0.58 \pm 0.04$ and $\langle s \rangle = 85.49 \pm 11.82$, respectively. These individual-level networks were then aggregated to form a group-representative network. This procedure can be viewed as a distance-dependent consistency thresholding of connectome data and the details have been described elsewhere[68]. The resulting group-representative network has the same number of binary connections as the average individual and the same edge length distribution. This type of non-uniform consistency thresholding has been shown to be superior to other, more commonly used forms[72].

**Behavioral tasks**. All participants completed a modified local-global perception task based on classical Navon figures[20] and a Stroop task with color-word pairings that were eligible and ineligible to elicit interference effects[19].

For the Navon task, local-global stimuli were comprised of four shapes—a circle, X, triangle, or square—that were used to build the global and local aspects of the stimuli. On all trials, the local feature did not match the global feature, ensuring that subjects could not use information about one scale to infer information about another scale. Stimuli were presented on a black background in a block design with three blocks. In the first block type, subjects viewed white local-global stimuli. In the second block type, subjects viewed green local-global stimuli. In the third block type, stimuli switched between white and green across trials uniformly at random with the constraint that 70% of trials included a switch in each block. In all blocks, subjects were instructed to report only the local features of the stimuli if the stimulus was white and to report only the global feature of the stimuli if the stimulus was green. Blocks were administered in a random order. Subjects responded using their right hand with a four-button box. All subjects were trained on the task outside the scanner until proficient at reporting responses using a fixed mapping between the shape and button presses (i.e., index finger = "circle", middle finger = "X", ring finger = "triangle", pinky finger = "square"). In the scanner, blocks were administered with 20 trials apiece separated by 20 s fixation periods with a white crosshair at the center of the screen. Each trial was presented for a fixed duration of 1900 ms separated by an interstimulus interval of 100 ms during which a black screen was presented.

For the Stroop task, trials were comprised of words presented one at a time at the center of the screen printed in one of four colors—red, green, yellow, or blue—on a gray background. For all trials, subjects responded using their right hand with a four-button box. All subjects were trained on the task outside the scanner until proficient at reporting responses using a fixed mapping between the color and button presses (i.e., index finger = "red", middle finger = "green", ring finger = "yellow", pinky finger = "blue"). Trials were presented in randomly intermixed blocks containing trials that were either eligible or ineligible to produce color-word interference effects. In the scanner, blocks were administered with 20 trials apiece separated by 20 s fixation periods with a black crosshair at the center of the screen. Each trial was presented for a fixed duration of 1900 ms separated by an interstimulus interval of 100 ms during which a gray screen was presented. In the trials ineligible for interference, the words were selected to not conflict with printed colors ("far," "horse," "deal," and "plenty"). In the trials eligible for interference (i.e., those designed to elicit the classic Stroop effect), the words were selected to introduce conflict (i.e., printed words were "red," "green," "yellow," and "blue" and always printed in an incongruent color).

**Additional data**. The connectome data was accompanied by (1) annotated system labels, which assigned each node to a single functional system, and (2) a group-representative functional connectivity (FC) matrix constructed from resting state scans that were collected concurrently with the behavioral data. See[73] for details. The system labels were taken from[23] and included seven cortical systems (dorsal attention, control, default mode, visual, limbic, somatomotor, and salience networks) along with an eighth sub-cortical label. The group-representative resting state FC network was generated by averaging subject-level resting state FC and by partialling out the effect of distance. The elements of the resulting matrix quantified the strength of functional connection between brain regions beyond what would be expected given their Euclidean distance from one another.

**Stochastic blockmodel**. The SBM seeks to partition a network's nodes into $K$ communities. Let $z_i \in \{1, \ldots, K\}$ indicate the community label of node $i$. Under the standard blockmodel, the probability that any two nodes, $i$ and $j$, are connected to one another depends only on their community labels: $p_{ij} = \theta_{z_i z_j}$.

To fit the blockmodel to observed data, one needs to estimate the parameters $\theta_{rs}$ for all pairs of communities $\{r, s\} \in \{1, \ldots, K\}$ and the community labels $z_i$. Assuming that the placement of edges are independent of one another, the likelihood of a blockmodel having generated a network, $A$, can be written as:

$$P(A|\{\theta_{rs}\}, \{z_i\}) = \prod_{i,j>i} \theta_{z_i z_j}^{A_{ij}} \left(1 - \theta_{z_i z_j}\right)^{1 - A_{ij}}. \quad (1)$$

Fitting the SBM to an observed network involves selecting the parameters $\{\theta_{rs}\}$ and $\{z_i\}$ so as to maximize this function.

**Weighted stochastic blockmodel.** The classical SBM is most often applied to binary networks where edges carry no weights. In order to maximize their utility to the network neuroscience community (where most networks are weighted), the SBM needs to be able to efficiently deal with weighted edges. Recently, the binary SBM was extended to weighted networks as the WSBM[17,18].

Equation (1) can be rewritten in the form of an exponential family of distributions[17]:

$$P(A|\{\theta_{rs}\}, \{z_i\}) \propto \exp\left(\sum_{ij} T(A_{ij}) \cdot \eta(\theta_{z_i z_j})\right). \qquad (2)$$

For the classical (unweighted) SBM, $T$ is the sufficient statistic of the Bernoulli distribution and $\eta$ is its function of natural parameters. Different choices of $T$ and $\eta$, however, can allow edges and their weights to be drawn from other distributions. The WSBM, like the classical SBM, is parameterized by the set of community assignments, $\{z_i\}$, and the parameters $\theta_{z_i z_j}$. The only difference is that $\theta_{z_i z_j}$ now specifies the parameters governing the weight distribution of the edge, $z_i z_j$.

Here, we follow[17], and model edge weights under a normal distribution, whose sufficient statistics are $T = (x, x^2, 1)$ and natural parameters are $\eta = (\eta/\sigma^2, -1/(2\sigma^2), -\mu^2/(2\sigma^2))$. Under this distribution, the edge $z_i z_j$ is parameterized by its mean and variance, $\theta_{z_i z_j} = \left(\mu_{z_i z_j}, \sigma^2_{z_i z_j}\right)$, and the likelihood is given by:

$$P(A|\{z_i\}, \{\mu_{rs}\}, \{\sigma^2_{rs}\}) = \prod_{ij} \exp\left(A_{ij} \cdot \frac{\mu_{z_i z_j}}{\sigma^2_{z_i z_j}} \right. \\ \left. -A_{ij}^2 \cdot \frac{1}{2\sigma^2_{z_i z_j}} - 1 \cdot \frac{\mu^2_{z_i z_j}}{\sigma^2_{z_i z_j}}\right). \qquad (3)$$

The above form assumes that all possible edges falling between communities are drawn from a normal distribution. However, most connectomes are sparse, i.e., edges where $A_{ij} = 0$ indicate the absence of a connection. One solution for dealing with this problem is to model edge weights with an exponential family distribution and to model the presence or absence of edges by a Bernoulli distribution (akin to the unweighted SBM)[17]. Letting $T_e$ and $\eta_e$ represent the edge-existence distribution and $T_w$ and $\eta_w$ represent the normal distribution governing edge weights, we can rewrite the likelihood function for the sparse WSBM as:

$$\log(P(A|z, \theta)) = \alpha \sum_{ij \in E} T_e(A_{ij}) \cdot \eta_e\left(\theta^e_{z_i z_j}\right) + (1-\alpha) \sum_{ij \in W} T_w(A_{ij}) \cdot \eta_w\left(\theta^w_{z_i z_j}\right), \qquad (4)$$

where $E$ is the set of all possible edges, $W$ is the set of weighted edges ($W \subset E$), and $\alpha \in [0, 1]$ is a tuning parameter governing the relative importance of either edge weight or edge presence (or absence) for inference. Here, we fix $\alpha = 0.5$, which balances their relative importance.

For each of the five data sets (connectomes from *Drosophila*, mouse, rat, macaque, and human), we maximize the likelihood of this sparse WSBM using a Variational Bayes technique described in[17] and implemented in MATLAB using code made available at the author's personal website (http://tuvalu.santafe.edu/aaronc/wsbm/). We varied the number of communities from $K = 2, \ldots, 10$ and repeated the optimization procedure 250 times, each time initializing the algorithm with a different set of parameters. We explore the convergence of the WSBM across multiple repetitions and the similarity of detected partitions in Supplementary Note 3 (Supplementary Figs. 11 and 12).

Blockmodels are flexible and can accommodate various classes of community structure. In network neuroscience, however, the majority of studies examining the brain's community structure have focused on its division into assortative communities by maximizing a modularity quality function:

$$Q(\{z_i\}, \gamma) = \sum_{ij} [A_{ij} - \gamma \cdot P_{ij}] \delta(z_i z_j). \qquad (5)$$

Here, $P_{ij}$ is the expected number of connections between nodes $i$ and $j$ under a null connectivity model and $\delta(\cdot \cdot)$ is the Kronecker delta function and is equal to 1 when its arguments are the same and 0 otherwise. The variable $Q(\{z_i\}, \gamma)$ is maximized by choosing community assignments $z_i$ that result in modules whose observed internal density maximally exceeds what would be expected under the null model. The free parameter, $\gamma$, is the structural resolution parameter and can be tuned to uncover communities of different size. The partition $\mathcal{P} = \{z_i\}$ that maximizes $Q(\{z_i\}, \gamma)$ is usually treated as a reasonable estimate of the network's community structure. While recent studies have investigated alternative definitions of $P_{ij}$, we use the common configuration model: $P_{ij} = \frac{k_i k_j}{2m}$, where $k_i = \sum_j A_{ij}$ and $2m = \sum_i k_i$.

Unlike WSBMs, most modularity maximization algorithms (henceforth referred to as $Q_{max}$) do not allow the user to specify the number of detected communities. In order to extract partitions of the network into exactly $K$ communities, we proposed a greedy algorithm in which nodes are initialized with random $K$-community partition, their assignments switched one at a time, and the new assignment accepted if the switch results in an increased $Q$. We repeated this algorithm 250 times for each $K$ and during each repetition we considered 10,000 random community switches. We fixed $\gamma = 1$ throughout.

**Statistics for comparing the WSBM with $Q_{max}$.** Variation of information: Modularity maximization is designed to uncover assortative communities while blockmodels are capable, at least in principle, of detecting more general types of community structure. It is unclear, however, when applied to brain network data whether the detected communities using either technique will actually differ from one another. We develop a set of statistics for comparing community structure at different topological scales ranging from global (whole partition), to mesoscale (community), to local (individual node).

At the global scale, we compare two partitions, $\mathcal{P}_1 = \{z_i^1\}$ and $\mathcal{P}_2 = \{z_i^2\}$, using the dissimilarity measure variation of information, VI, which yields an information theoretic distance between two partitions[21]:

$$\mathrm{VI}(\mathcal{P}_1, \mathcal{P}_2) = H(\mathcal{P}_1) + H(\mathcal{P}_2) - 2I(\mathcal{P}_1, \mathcal{P}_2), \qquad (6)$$

where $H(\mathcal{P})$ and $I(\mathcal{P}, \mathcal{Q})$ are the entropy and mutual information. The more similar two partitions are to one another, the closer their variation of information is to zero. Two partitions may differ from one another, trivially, if they feature a different number of communities. Throughout this section and the next and in order to avoid this issue, we only compare partitions if they feature the same total number of communities.

Community and regional assortativity: While variation of information makes it possible to assess the similarity of partitions as a whole, we also wanted to assess which brain regions, systems, and communities differ between techniques. One dimension along which we expect the techniques to differ is the extent to which the detected communities are assortative. To quantify this property, we propose community and regional assortativity scores.

For a community $r$, we define its assortativity as:

$$\mathcal{A}_r = \left[\omega_{rr} - \max_{s \neq r}(\omega_{rs})\right], \qquad (7)$$

where $\omega_{rs} = \frac{1}{n_r \cdot n_s} \sum_{i \in r, j \in s} A_{ij}$ is the weighted density of connections between communities $r$ and $s$. For directed networks, we consider both incoming and outgoing connections, and we replace $\max_{s \neq r}(\omega_{rs})$ with the greater of $\max_{s \neq r}(\omega_{rs}^{\mathrm{In}})$ or $\max_{s \neq r}(\omega_{rs}^{\mathrm{Out}})$.

We also calculated an analogous score for individual brain regions. Given region $i$'s community assignment $z_i$, we calculated its connection density to community $r$ as $a_{ir} = \frac{1}{n_r} \sum_{j \in r} A_{ij}$. Then, its regional assortativity score was given by:

$$\phi_i = a_{iz_i} - \max_{r \neq z_i} a_{ir}. \qquad (8)$$

Again, we modified this equation slightly for directed networks to take into account both incoming and outgoing connections. We replaced $a_{iz_i}$ with the lower of either $a_{iz_i}^{\mathrm{In}}$ or $a_{iz_i}^{\mathrm{Out}}$, and we replaced $\max_{r \neq z_i} a_{ir}$ with the greater of either $\max_{r \neq z_i} a_{ir}^{\mathrm{In}}$ or $\max_{r \neq z_i} a_{ir}^{\mathrm{Out}}$.

Under this definition, the assortativity score measures the minimum difference between the density of connections made between a region and its own community, and the density of connections made between a region and any other community. In computing both regional and community assortativity scores, we excluded singleton communities.

**Maximally assortative set.** In addition to the metrics described above, we also sought to identify the largest set of communities uncovered by the WSBM that exhibited assortative community structure. We termed this set the maximally assortative set and defined it as the set of $k \leq K$ communities, $\{c_1, \ldots, c_k\}$ such that $\min_i(\omega_{c_i, c_i}) > \max_{i \neq j}(\omega_{c_i, c_j})$ and the total number of nodes in those communities was maximized.

**Rich club estimation.** We identified putative rich club nodes by maximizing a weighted rich club coefficient, $\phi^w(k)$, where $k$ is node degree[74]. Intuitively, a weighted rich club is composed of highly connected nodes linked to one another by connections with strong weights. To calculate $\phi^w(k)$, we first identify the sub-network composed only of nodes whose degree is $k$ or greater, the number of connections among those nodes, $E_k$, and the total weight of those connections $W_k$. We also calculate $W_{k>}^{\max} = \sum_{l=1}^{E>k} w_l^{\mathrm{rank}}$, which measures the maximum possible value that $E_k$ connections could have given the edge weights present in the network.

$$\phi^w(k) = \frac{W_{>k}}{W_{k>}^{\max}}. \qquad (9)$$

We compared $\phi^w(k)$ for the observed network against the same measure made over an ensemble of 100 randomized networks with the same degree sequence. For every possible $k$, we calculated the fraction of all randomized networks whose rich club coefficient was in excess of that in the observed network's. This fraction served

as a *p*-value for performing statistical tests and made it possible to identify statistically significant rich clubs ($p < 0.05$).

This procedure results in a range of *k* over which rich clubs are considered statistically significant. Rather than characterize this entire range, we focused on a 20–80 split of network nodes into rich and non-rich groups. We justify this split on the grounds that (1) all of the networks we studied exhibited a statistically significant rich club in this range, making it unnecessary to develop separate criteria for studying rich clubs across species, and (2) a rich club composed of 20% of a network's nodes is exclusive enough to be of interest but not so large as to be trivial (Supplementary Fig. 18).

**Community interaction motifs and morphospace analysis.** Uncovering a network's community structure makes it possible to shift focus away from individual nodes and edges and onto communities and their aggregate interactions with one another. Taking such a coarse view of a network can make it possible to more easily infer the functions of communities and the roles of individual nodes within those communities.

Here, we study those interactions using a theoretical morphospace analysis[29], a technique recently adapted to the study of complex networks[30]. A morphospace is a hyperspace whose axes represent the features of an organism or system. Take, for example, foraminiferal tests—the shells that form the outer layers of certain aquatic protists—that can be modeled and fully parameterized using a small number of morphological traits[75]. A simple morphospace can be constructed whose axes are represented by these traits, and any observed test can then be situated within this space. Oftentimes, there will exist certain regions of space (i.e., particular sets of traits) that are densely populated and other regions that, by comparison, are not populated at all. By studying which sets of traits are more common, it becomes possible to deduce the evolutionary constraints and pressures that drove their emergence.

It is in this same spirit that network morphospaces can be constructed[30]. Instead of axes representing an organism's morphological or physiological traits, the axes of a network morphospace represent topological properties of a network, e.g., its efficiency, wiring cost, complexity, etc., or the parameters of network models.

In this case, we construct a community morphospace. Each point in the morphospace represents a pair of communities, *r* and *s*, and the point's location is given by the within-community and between-community connection densities: $\omega_{rr}$, $\omega_{ss}$, and $\omega_{rs}$. Given these values, we can also classify community interactions into one of three distinct motifs (interaction types):

$$M_{rs} = \begin{cases} M_{assortative}, & \text{if } \min(\omega_{rr}, \omega_{ss}) > \omega_{rs} \\ M_{core-periphery}, & \text{if } \omega_{rr} > \omega_{rs} > \omega_{ss} \\ M_{core-periphery}, & \text{if } \omega_{ss} > \omega_{rs} > \omega_{rr} \\ M_{disassortative}, & \text{if } \omega_{rs} > \max(\omega_{rr}, \omega_{ss}). \end{cases}$$

From these classifications we were able to associate motifs to individual nodes. Node *i*'s participation in motif *M* was calculated as the number of times that the community to which node *i* was assigned interacted with any other community to form a motif of type *M*. We then normalized these counts by the total number of motifs (for a *K*-community partition there are in total $K(K-1)/2$ or $K(K-1)$ total motifs depending upon whether the network is undirected or directed, respectively). Importantly, when computing participation in core-periphery motifs, we distinguished between the core and the periphery, and computed separate participation scores for each. Finally, from participation types we computed each node's diversity index, which measured the entropy of its normalized participation in each motif type.

**Diversity index.** A partition of a network into communities induces a set of two-community motifs based on connection densities. In the previous section we presented rules for classifying those motifs into one of three classes. For a *K*-community partition, community *r* participates in $K-1$ interactions. We can calculate for each motif class (now differentiating between cores and peripheries, resulting in four distinct classes), how frequently it appears among community *r*'s $K-1$ interactions. If we express these frequencies as probabilities, $P_a$, $P_c$, $P_p$, and $P_d$ (subscripts indicate "assortative", "core", "periphery", and "disassortative" motif frequencies, respectively), we can then calculate an entropy:

$$H_r = -\left[P_a \log_2 P_a + P_c \log_2 P_c + P_p \log_2 P_p + P_d \log_2 P_d\right]. \quad (10)$$

This entropy is zero if community *r* participates in only one motif class and is maximized when *r* participates in all classes equally. We then assign this score to all nodes $i \in r$. The resulting vector of length $[N \times 1]$ specifies the single-partition diversity index for each node. We can calculate this vector for all *K*-community partitions and estimate mean diversity indices for each node by averaging across partitions.

Note that while we define the diversity index at the level of individual brain regions (network nodes), it would be straightforward to average node-level diversity scores to compute a global diversity score that could serve to characterize the diversity of meso-scale structure in the network as whole. Alternatively, a global diversity index could be computed straightforwardly as an entropy based on the complete set of community motif frequencies.

**Code Availability**. All analysis code is available from the authors upon reasonable request. MATLAB code for implementing WSBM is available at http://tuvalu.santafe.edu/~aaronc/wsbm/.

**Data availability**. All data are available from the authors upon reasonable request.

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

## Acknowledgements

R.F.B. and D.S.B. would like to acknowledge support from the John D. and Catherine T. MacArthur Foundation, the Alfred P. Sloan Foundation, the Army Research Laboratory and the Army Research Office through contract numbers W911NF-10-2-0022 and W911NF-14-1-0679, the National Institute of Health (2-R01-DC-009209-11, 1R01HD086888-01, R01-MH107235, R01-MH107703, R01MH109520, 1R01NS099348 and R21-M MH-106799), the Office of Naval Research, and the National Science Foundation (BCS-1441502, CAREER PHY-1554488, BCS-1631550, and CNS-1626008). J.D.M. acknowledges support from the Office of the Director at the National Institutes of Health through grant number 1-DP5-OD-021352-01. The content is solely the responsibility of the authors and does not necessarily represent the official views of any of the funding agencies. We are also grateful to Mikail Rubinov for sharing mouse connectome and gene expression data, and to Olaf Sporns for sharing rat connectome data.

## Author contributions

R.F.B. and D.S.B. designed study and carried out analyses. J.D.M. collected the human data set. All authors wrote the manuscript.

## Additional information

**Competing interests:** The authors declare no competing financial interests.

