## [Peer Review File · Nature Communications]

Reviewers' comments:

Reviewer #1 (Remarks to the Author):

Most network analysis of the human brain uses the modularity maximization algorithm to detect meso-scale network architecture, which assumes the brain is composed of assortative communities. This assumption may not be valid, however, so this manuscript relates the modularity maximization algorithm to another algorithm, WBSM (weighted stochastic blockmodel), that does not assume assortative community structure. This is a compelling argument and an important exploration to help the field most accurately estimate community structure. This manuscript is well-organized and well-written. As will be elaborated below, the reasons for proposing that all assortative communities is not a valid description of the brain's network structure need to be fleshed out to increase the impact of this manuscript.

More detailed comments:

Your argument is compelling and the introduction sets the problem up well. I found myself wondering, however, about concrete examples, where nodes and edges are known, of these different types of meso-scale architectures. To elaborate, many of the early modularity articles talk about small world/modular organization that has mainly assortative communities, and how those are observed in biology, nature and society when we can measure all connections, as well as in central nervous systems where we can measure all neurons and synapses (i.e., *c. elegans*). Those examples provide a strong argument that the brain may work in a similar efficient manner to all these other network examples. Are there concrete examples of these other types of meso-scale architectures, or a reason why fully assortative structure may work for some categories of networks but fall apart in the brain?

On a somewhat related note, the addition of other species, where we have better measurements of edges, is a strength of this manuscript. However, the main text is limited to general "and results are similar in other species" statements. This is 1) not fully accurate, since in reading the supplementary materials there are nuanced differences that may be important; and 2) reduces the strength of your argument. As of now, that data is a supplemental side note, but mentioned in the title, abstract and discussion as if it were a main part of the paper as well. If possible, I would add results from the other datasets into the main text. If you do not have space, I would make it more clear that the non-human datasets are not equal to the human dataset in terms of explanation in the paper, which would mean taking non-human out of the title and other parts of the manuscript that imply that you considered all equally.

Your last paragraph of the "Connectomes support diverse meso-scale architecture" Results section seems to be a bit strong of a conclusion given the analyses you conducted. You compared assortativity between Qmax and WSBM methods, and should expect assortativity to be lower in the WSBM method based on its algorithm versus that of Qmax, as you wrote earlier in that section. So this is more of a validation that the different algorithms are doing what you think they are than that the differences are functionally meaningful. Adding other network metrics that are thought to be important for cognition and comparing their values across algorithms would make that conclusion more appropriate...or not making that claim until you relate these two community structures to cognition or other metrics below.

Similarly, your first paragraph of the "Functional relevance of the WSBM" section "To this point, we have used the WSBM to demonstrate that connectomes exhibit diverse, non-assortative meso-scale structure...suggesting that the richer, non-assortative communities are closer to the brains canonical architecture." seems like a straw man, since the algorithm that allows for non-assortative meso-scale

structure finds some non-assortative communities, while the algorithm that limits its results to maximizing assortativity finds all assortative communities. This argument (that is scattered throughout the manuscript) needs an explanation regarding why maximizing assortativity to the exclusion of other types of communities is not as biologically plausible as allowing multiple types of community structure. There have been many papers arguing why algorithms maximizing modularity are biologically plausible. This concept that we are limiting our investigations by doing so is very compelling, but it needs more biological motivation (as well as functional relevance...see other comments about that below) to have a strong impact on the field.

You state later in that section that "Intuitively, functionally-related brain regions are linked by strong functional connections. If a community does a good job identifying sets of such regions, then the within-community density of functional connections should be greater than the between-community density." But the point of the manuscript is that assuming communities have this strong-within and weak-between connectivity is not necessarily correct, and we should allow for other types of communities (Figure 1). That sentence seems to contradict the main point of the manuscript. It is very possible that I am missing something, but regardless that concept should be clarified so as to not seem contradictory if it isn't. Further, the result that follows appears to show that even with WSBM, when averaging across the whole brain, communities are assortative (strong within, weaker between). It is also unclear to me how both within and between-community connections can be more dense if the nodes and edges are matched across algorithms, so more explanation about that is needed as well.

"Community morphospace reveals rules for between-community interactions": how do the previously-described results offer "a better explanation for human brain function and mouse genetic expression than that provided by assortative communities alone"? Thus far, I am convinced that it is "different", but have seen no evidence that it is "better".

Also in that section, your description of the results is that the WSBM algorithm identifies a significant number of core-periphery interactions (which I see in Figure 6) and disassortative interactions (which I do not see). It looks to me that the proportion of disassortative interactions using WSBM, and the difference between WSBM and Q_{max} , are both 0. Please clarify in the text and the figure what you mean by this.

Your brain figures exclude subcortical brain regions, yet they keep popping up in the article as diverse and one of the groups of regions that would be better defined using WSBM. Please add images of the subcortical structures as well (this goes for all figures that show where in the brain different node types are).

Behavioral relevance section: it would be interesting to do this analysis using Q_{max} to define community structure as well, to determine which of the two algorithms best explains behavior. If WSBM is more related to behavior on these cognitive control tasks, that is an argument for the functional relevance (and potentially higher accuracy) of community structure derived from that algorithm.

In your first sentence of the discussion, while I don't disagree that having different types of communities is plausible, I have not seen a strong argument for that other than that an algorithm that attempts to find these different types of communities can. This to me is no less biased than modularity maximization being biased to find an assortative community structure. This argument should be made more explicit and linked to biology throughout the manuscript to support your conclusions.

Your sentence: "Moreover, communities detected using the weighted stochastic blockmodel better

recapitulated observed intrinsic functional relationships among brain regions in the human, and relationships between gene co-expression patterns in the mouse, compared to more commonly-used techniques such as modularity maximization." See my point above about this results section. This argument does not logically follow to me.

The last full sentence on page 12 is the first I have seen that gives a biological reason for looking for non-assortative community structure. If this and other related arguments are highlighted more in both the introduction and discussion, this paper will be more convincing.

Reviewer #2 (Remarks to the Author):

"Diversity of meso-scale architecture" by Betzel et al. compares two forms of community decompositions applied to structural connectomic data – a classic decomposition that maximises the assortativity (Q_{max}) of the decomposition with a more recently developed weighted stochastic block algorithm (WSBM). The WSBM appears to group nodes into modules such that the likelihood of inter-module connections is approximately even for all nodes within each module (i.e. does not depend upon heavily upon each individual module). There is substantial interesting material in this manuscript which is clearly written and nicely illustrated. I do however hold a number of concerns:

1. One of the major findings is not really that surprising and I think more work is required to establish the significance of the finding. Namely, it is not surprising that any decomposition that does not maximise the assortativity of a module decomposition is less assortative than one that does! i.e. there is an element of circularity in the central finding of the paper. The question that seems not to be answered is not whether the WSBM decomposition reveals a less assortative community structure than Q_{max} , but whether the degree of core-periphery and disassortativity in the human connectome is greater or lesser than expected by chance: Since we already know that Q_{max} -sorted connectomes are more assortative than degree and strength preserving random surrogates, we should now check whether WSBM ones are likewise more core-periphery or more disassortative than strength-sequence matched surrogates. Are they likewise more or less arranged when such surrogates also account for the geometric embedding of real connectomes into three dimensions? Only after this has been established, do I think the more fine-grained analyses of figure 3-5 make sense.

2. I doubt as presented, many readers will understand what the WSBM actually does do, other than being "not maximally pro-assortative". There is a brief heuristic for the standard block model that seeks to minimise a cost function that penalises the heterogeneity of inter-module connectivity amongst nodes within the same module (that is my reading of equation (1)). The WSBM is more complex, but I assume it is a generalisation of the same principle to accommodate edge weights. I think the authors should provide a better heuristic explanation of the WSBM minimization and put it at the beginning of the Results so that the general reader can understand what is being optimized.

3. I also think the paper would be improved with some groundtruth validations, using growth models, to show that the WSBM algorithm does what the authors assume it does – namely that Q_{max} and WSBM should converge if applied to networks that are generated by suitable preferential attachment models, that add new edges to maximize assortativity (the authors are leading experts in such generative modelling); Also that they should strongly diverge when such networks are grown to maximise core-periphery arrangements and for maximally disassortative networks.

4. I found the choice of the statistic for comparing functional and structural connectivity somewhat counter-intuitive: Namely of seeing if the corresponding functional networks (when grouped into the

corresponding modules) were maximally assortative. Its interesting, although counter-intuitive that WSBM outperforms Qmax on this metric. The authors should likely also be cautious that network measures based on linear correlations induce artificial assortativity through the nature of the measure [1].

5. Novelty: As the authors cite, there is already substantial work using SBMs in human connectomes. A brief summary of what is new here would help. Also, there are elements of the current findings that could be unpacked from prior work: for example, [2] previously showed that rich club nodes preferentially existed as apex nodes in open motifs of 3 nodes (that is maximally disassortative motifs [3]). Also, Figure 10a of [4] shows that very high degree nodes are actually less often inter-connected than by chance – i.e. have a “cartel-like” disassortative property. The present finding, by very different methods, seems reassuringly convergent.

6. I am somewhat (pleasantly) surprised, given the very noisy nature of individual tractography data that I have seen (expect perhaps for carefully reconstructed connectomes from the highly curated human connectome project) that individual correlations with performance were discovered. Brief details of acquisition and reconstruction must be provided in the Methods here, since this remains a contentious area. What was the connection density? Also, a very brief summary of these data could be given at the beginning of the Results.

7. Section on “Behavioural relevance...”: Why/how were differences in total connection weight partialled out? Why not normalize the matrices to have uniform weights before the analysis? Were do the differences arise (e.g. do they correlate with white matter volume)? Also, what was the nature of the FDR correction? How many tests were performed/corrected for?

Minor:

1. Most of the first paragraph of the section “Connectomes support diverse ...” simply repeats the Intro and could be deleted.
2. I found it impossible to see any differences between Qmax and WSBM in Figure 3B.
3. Why are the WSBM networks more assortative than the null distribution in Fig 3C?
4. Suggest delete the interpretive phrase “suggesting the capacity for an equally ...” from p6 of the Results.
5. It is the authors’ own preference, but I found aspects of the Discussion highly speculative for an original research article.
6. p13: The cite regarding repertoire diversity might also consider [3].
7. p15: How did the authors go from a whole group consistency-based connectome back to individual subjects?
8. p16: were the structural and functional connectomic data and the behavioural data all from the same subjects? Why 30 for structural connectivity and 70 for functional connectivity?

References:

1. Zalesky, A., Fornito, A., & Bullmore, E. (2012). On the use of correlation as a measure of network connectivity. *Neuroimage*, 60(4), 2096-2106.
2. Harriger L, Van Den Heuvel MP, Sporns O. 2012 Rich club organization of macaque cerebral cortex and its role in network communication. *PLoS ONE* 7,
3. Sporns O, Kotter R (2004) Motifs in brain networks. *PLoS Biology* 2: e369.
4. Roberts JA, Perry A, Lord AR, Roberts G, Mitchell PB, Smith RE, Breakspear M (2016) The contribution of geometry to the human connectome. *Neuroimage* 124: 379-393.

Reviewer #3 (Remarks to the Author):

I quite enjoyed reading this report, which challenges the classical view of the view of the connectome being composed of segregated communities and introduces the alternative view on the existence of more heterogeneous community configurations.

Unlike to the standard methods used to define communities of the structural connectome, the authors utilize a different technique, weighted stochastic block model (WSBM), one that does not explicitly impose the assumption of the modularity maximization and hence segregated communities. Using the WSBM, this work “reveals” other kinds of communities and community interactions, where the newly found communities show a better overlap with the functional networks of the brain. It is also quite interesting to see that the intersubject variability in diversity of the community profiles of certain brain regions shows some correlation with the behavioral performance.

I think the paper introduces a novel and quite interesting perspective on the diversity of community organization in the connectome. My major comment is about the lack of true ground truth for the studied problem. As a logical decision, the authors chose to compare to the most commonly used state-of-the art method, modularity maximization, referred as Q_{max} here. However, this algorithm, as also stated by the authors is designed to maximize modularity and hence assortativeness. Hence, removing this particular constraint – modularity maximization – as in the case of using WSBM, naturally leads to less assortative community structures compared to Q_{max} , which by design extracts segregated communities. My main question is how do we know that the partitions returned by WSBM are more “correct” compared to those detected by Q_{max} ? Having said that, I would acknowledge that this is not a drawback of the method presented here but a general concern about the lack of a true ground truth for the problem at hand. It may be important to at least mention or discuss this point and maybe point out that the results drastically change when the modularity maximization constraint is removed, although a true ground truth for neither algorithm exists.

Please find below some questions and comments about the specifics of the method and statistics used.

1) How does the approach followed by the authors differ from the hierarchical clustering method, cited as ref [35] in the manuscript, as to my knowledge this method also falls outside of the modularity maximization framework. Also, what was the motivation behind the choice of WSBM instead of for instance the hierarchical clustering approach as in [35]?

2) Page 2, first paragraph: “Next, we define a node-level diversity index that quantifies the extent to which individual neural elements participate in communities of all classes.” Can a node (neural element) belong to multiple clusters; i.e. the communities can be overlapping and are not disjoint?

3) Fig. 3a: Are the within-technique variation of information (VI) scores based on the comparison of partitions with the same number of communities on two different subjects’ datasets? As the comparisons are performed on partitions with the same number of communities, I assume two different partitions using the same technique can come from the use of different datasets. However, that is not clear at that point of the manuscript, as any information on different subjects’ data etc. has been provided yet. The authors may want to explain what leads to different partitions with the same number of communities, which are used for comparison.

4) Fig. 3A: Also, the authors mention that both techniques, WSBM and Q_{max} lead to self-similar partitions that are statistically different between techniques. In Fig. 3A, the variation of information (VI) scores of WSBM are much higher than those of Q_{max} and for $K=10$, the within-technique VI for

WSBM is even higher than between-technique VI. What is the cause of such difference in within-technique VI observed between the two techniques? Are the WSBM partitions reliable, if they show such high within technique VI scores?

5) The authors mention: "We compared these curves using functional data analysis, which is a set of statistical tools for comparing continuous curves [47,48]. We found that the observed scores were smaller than those obtained under the null model ($p < 10^{-3}$), confirming that WSBM communities tend to be less assortative than Qmax (Fig. 3C)". Aren't these curves discrete set of measurements, hence allowing for a comparison for instance by Monte-Carlo approach; simply by shuffling the labels of assortativity scores between two methods over multiple comparisons?

6) I fail to understand Fig. 3C and the stats performed with functional data analysis. What does the y-axis labeled as "Probability" represent? The authors mention "Specifically, we generated a statistic by performing a pointwise subtraction and summation of the curves $A(N)$ obtained for the WSBM and Qmax. The value of this statistic quantifies the difference between mean community assortativity across communities of all sizes and is negative when communities detected using Qmax are more assortative than WSBMs. We compared this statistic against a null distribution obtained from a null model wherein we preserved the number and size of communities in a given partition but permute nodes' assignments uniformly and randomly (1000 repetitions)." What does the pointwise subtraction and summation of the curves $A(N)$ yield, is it average difference between the two curves? For the null distribution, doesn't the permutation of nodes result in non-continuous clusters, similar to a random assignment? Would that be a fair comparison to use?

7) The communities detected by WSBM more closely reflect the functional networks. However, I believe it is important to point out that functional networks emerge from the dynamics and interactions between neural elements that is constrained by the structural connections but not purely determined by them. Hence, although some degree of overlap between structure and function is expected, it is not expected that they will be the same or very similar. The effect of the dynamics would play a crucial role in the emergence of the functional networks.

8) The authors state "To test whether this was the case, we imposed partitions obtained from the WSBM and Qmax applied to the structural connectome onto the FC matrix and computed the difference of within and between community FC density. We found that over a range $K = 2, \dots, 10$, the WSBM consistently uncovered communities whose internal FC density exceeded their between-community density (Fig. 5A)." The functional networks are defined on the FC matrix, hence I would have thought that they would superimpose FC matrix parcellation onto the structural connectome (SC) partitions. For instance Fig. 5A caption states: "Functional connectivity (FC) matrix ordered by functional system". As both, FC connectivity and the labeling of the functional systems come from the functional connectivity, how does this figure capture the partitions of the structural connectome? Is it a misunderstanding on my side or is there a confusion between SC and FC in the wording here?

9) I think the correlations between the diversity index and performance categorized according to functional networks is very interesting. From what I can see in Fig. 8B, one can conclude that some networks require certain type of motifs (interactions) such as the visual network and the DAN, whereas others such as the control network, subcortical areas and maybe also the default mode network contain interactions of various kinds for a good cognitive performance. What about the whole brain diversity of connections? Would that make any inference on the cognitive performance?

Minor points:

- Page1: What is the difference between clusters and communities? I found the illustration of different connectivity profiles in Fig. 1 very useful. It may be very helpful to illustrate the concepts of region, community, partition in a similar manner for the naive reader, if possible, at least as supplementary material.
- Page 1: what do individual network nodes represent? Brain regions? It would be useful to specify here.
- Is assortative architecture the same as small-world, as used in some reports on connectome's architecture?
- Fig. 2: Community labeled with purple seems to consist of only one small brain region in the right hemisphere, which does not seem to have any correspondence in the left hemisphere, whether labeled as the same community or not. Where does this asymmetry stem from? Is it an algorithmic artifact?
- Fig. 3. Caption: Q_{\max} should be $Q_{\{max\}}$ in latex notation, "perserved" should be "preserved".
- Fig. 3F, what do the upper and lower limits of the box plot represent?
- Fig. 4C, how were the rich club nodes labeled/estimated?
- What is the difference between a core community and a hub?
- Page 13, first paragraph: I would say "functional connectivity" (FC) instead of "functional dynamics", as the comparison was done to FC and with the emergence of new methods such as dynamic functional connectivity etc, "functional dynamics" is now understood as changing functional connectivity.
- What was the motivation behind using a 128 parcellation and can the authors comment on if/how the change of parcellation may change the observed effects?
- Page 18: In section "Community and regional assortativity" the authors provide the equations for both directed and undirected graphs. Is that done so for the generalization of the provided methods for directed graphs? As far as I understand the results in the actual manuscript are based on undirected graphs. I believe the directed graph use may be necessary for the mouse data? If so, it may be worth mentioning this in the manuscript.
- Eq. (8): it could be easier for the reader if a different notation instead of double indexing was used to refer a_i and $a_{\{iz_i\}}$. It is not clear to me what $a_{\{iz_i\}}$ represents.
- It may also be useful to express the diversity index mathematically.
- The macaque connectome results seem to show the opposite trade in terms of being assigned to maximally assortative set as rich club and non-rich club members (Fig. S4O). Do the authors have any speculative idea on what may be the reason for this opposite trade?

Reviewer #1

Comment 1

Most network analysis of the human brain uses the modularity maximization algorithm to detect meso-scale network architecture, which assumes the brain is composed of assortative communities. This assumption may not be valid, however, so this manuscript relates the modularity maximization algorithm to another algorithm, WBSM (weighted stochastic blockmodel), that does not assume assortative community structure. This is a compelling argument and an important exploration to help the field most accurately estimate community structure. This manuscript is well-organized and well-written. As will be elaborated below, the reasons for proposing that all assortative communities is not a valid description of the brains network structure need to be fleshed out to increase the impact of this manuscript.

Your argument is compelling and the introduction sets the problem up well. I found myself wondering, however, about concrete examples, where nodes and edges are known, of these different types of meso-scale architectures. To elaborate, many of the early modularity articles talk about small world/modular organization that has mainly assortative communities, and how those are observed in biology, nature and society when we can measure all connections, as well as in central nervous systems where we can measure all neurons and synapses (i.e., *C. elegans*). Those examples provide a strong argument that the brain may work in a similar efficient manner to all these other network examples. Are there concrete examples of these other types of meso-scale architectures, or a reason why fully assortative structure may work for some categories of networks but fall apart in the brain?

This is a good point. Most work in modern network science has focused on assortative communities (oftentimes detected by maximizing a modularity quality function). Consequently, there is no shortage when it comes to finding examples in the literature of this type of community. However, there is also a parallel literature on blockmodeling that originated in the social sciences and statistics [1, 2] that has only recently been widely appreciated in other fields, like physics and computer science [3]. In any case, we agree that it would be good to note examples of non-assortative communities (core-periphery and disassortative) in complex networks.

We have now added the following sentences to the **Introduction** section:

- “While this perspective has proven useful, it has a number of drawbacks, of which we focus on two. First, it makes the strong assumption that connectome meso-scale architecture is strictly assortative (Fig. 1A). This assumption stems in part from the algorithms used to detect communities, the most popular of which seek internally dense and externally sparse sub-networks [4, 5]. As a result, these algorithms are incapable of detecting non-assortative structure, such as core-periphery (Fig. 1B) and disassortative (Fig. 1C) communities or mixtures of different community types (Fig. 1D), all of which are evident in real-world socio-technical and biological networks [6, 7, 8, 9, 10, 11, 12]. Moreover, modularity maximization and related techniques may overlook important and functionally-relevant characteristics of neural circuits, which exhibit non-assortative, cell type-specific wiring diagrams [13, 14, 15]. It is unclear, then, whether the assortative communities uncovered using these algorithms represent an accurate picture of connectome meso-scale structure or whether they reflect the assumptions and limitations of the algorithms themselves.”

The reviewer also suggests that it might be useful to test the WSBM framework on a network for which we have complete knowledge of its connectivity. To address this point, we used both Q_{max} and WSBM to detect communities in the *C. elegans* network of chemical synapses (we ignored electrical synapses, as it was unclear how to combine weight information about electrical and chemical synapses into the same network model and still retain interpretable and neurobiologically meaningful edge weights) [16].

Our analysis consists of two components. First, from communities detected using the WSBM and Q_{max} and with identical methods as in the main text, we constructed a morphospace of community interactions (Fig. 1). This figure demonstrates that when we use Q_{max} to uncover communities (and varying the number of communities from $K = 2$ to $K = 10$), they are *always* arranged in assortative motifs. Using the WSBM, on the other hand, we consistently identify both core-periphery and disassortative motifs. This finding indicates that the WSBM, indeed, detects non-assortative community structure. Moreover, because the *C. elegans* connectome has been painstakingly mapped out at the cellular level, we can rule out the possibility that the

80 non-assortative communities we reported in the original submission are a product of scale (inter-areal *versus*
 81 inter-cellular connectomes).

Figure 1: **Community morphospace for *C. elegans* connectome.** The top panels depict the community morphospace (in log scale) for community dyads recovered from partitions detected using Q_{max} (left) and the WSBM (right). The color of each point represents each dyad’s classification: cyan = assortative, yellow = core-periphery, and red = disassortative. The top plots are depicted with the number of communities fixed at $K = 5$. The bottom panels depict the proportion of dyad classes as we varied the number of communities from $K = 2$ to $K = 10$. In general, Q_{max} only detected assortative community dyads while the WSBM detected all three types.

As a second comparison of Q_{max} and the WSBM applied to *C. elegans* connectome data, we identified
 representative partitions for each technique as we varied the number of communities from $K = 2$ to $K = 10$.
 We then reordered and blocked the *C. elegans* connectivity matrix according to the communities uncovered
 by the WSBM (Fig. 2A). In the margins of each plot, we color-coded each node’s corresponding community
 label as detected using Q_{max} . Had the two techniques generated similar community partitions, then we
 would expect the Q_{max} labels to be homogeneous within each block. However, Q_{max} community labels are
 heterogeneously distributed within WSBM communities, demonstrating qualitatively that the two techniques
 uncover communities of different character.

In addition, for the representative WSBM communities, we also show the density (average weight of all
 possible connections) of each block (Fig. 2B). While certain pairs of communities are configured in assortative
 relationships, many are not. These results suggest that the WSBM identifies non-assortative communities in
 the *C. elegans* connectome. Seeing this structure at the cellular scale in a fully-mapped connectome further
 supports the conclusions of our manuscript.

We now include these additional analyses in the **Supplementary Materials** under the section **WSBMs**
 **at the cellular level:**

- • “The human connectome data analyzed in the main text and the non-human connectomes analyzed in
 this supplement are examples of inter-regional networks. Individual cells and populations have been
 aggregated into spatially-contiguous, macroscopic parcels or regions. While this approach is common

and serves to reduce the dimensionality of a network (making it more manageable for analysis), it also
averages over the properties of those cells and populations. If regions are homogeneous in terms of the
cells that they contain, then region-level analysis loses very little information. However, if a region's
constituent cells exhibit heterogeneity in terms of their connectivity patterns, then we lose access to
this information. It is unclear, then, how the WSBM would behave when applied to a cellular-level
network.

In this section, we apply the WSBM to the network of the nematode *C. elegans*. We analyze the
directed and weighted network of chemical synapses described in [16]. This network consists of 279
neurons, 2194 connections, and 6394 synapses (neurons can synapse onto one another more than once).
Our aim is to show that, even at this cellular scale, the WSBM identifies high levels of non-assortativity
while Q_{max} does not. As we note in the main text, this does not demonstrate conclusively that the
“true” meso-scale structure of *C. elegans* is composed of non-assortative communities. Instead, it
complements other recent papers [9] highlighting the apparent utility of blockmodels for identifying
non-trivial communities in cellular-level data.

Our analysis consisted of two components. First, using identical methods as in the main text, we
constructed a morphospace of community interactions (Fig. 1). This figure demonstrates that when
we use Q_{max} to uncover communities and vary the number of communities from $K = 2$ to $K = 10$,
communities are *always* arranged in assortative motifs. Using the WSBM, on the other hand, we
consistently identify both core-periphery and disassortative motifs.

As a second comparison of Q_{max} and the WSBM applied to *C. elegans* connectome data, we identified
representative partitions for each technique as we varied the number of communities from $K = 2$
to $K = 10$. We then reordered and blocked the *C. elegans* connectivity matrix according to the
communities uncovered by the WSBM (Fig. 2A). In the margins of each plot, we color-coded each
node's corresponding community label as detected using Q_{max} . Had the two techniques generated
similar community partitions, then we would expect the Q_{max} labels to be homogeneous within each
block. However, Q_{max} community labels are heterogeneously distributed within WSBM communities,
demonstrating qualitatively that the two techniques uncover communities of different character.

In addition, for the representative WSBM communities, we also show the density (average weight of
all possible connections) of each block (Fig. 2B). While certain pairs of communities are configured in
assortative relationships, many are not. These results suggest that the WSBM identifies non-assortative
communities in the *C. elegans* connectome. Seeing this structure at the cellular scale in a fully-mapped
connectome further supports the conclusions of our manuscript.”

Comment 2

*On a somewhat related note, the addition of other species, where we have better measurements of edges, is*
*a strength of this manuscript. However, the main text is limited to general “and results are similar in other*
*species” statements. This is 1) not fully accurate, since in reading the supplementary materials there are*
*nuanced differences that may be important; and 2) reduces the strength of your argument. As of now, that*
*data is a supplemental side note, but mentioned in the title, abstract and discussion as if it were a main part*
*of the paper as well. If possible, I would add results from the other datasets into the main text. If you do*
*not have space, I would make it more clear that the non-human datasets are not equal to the human dataset*
*in terms of explanation in the paper, which would mean taking non-human out of the title and other parts*
*of the manuscript that imply that you considered all equally.*

We agree and thank the reviewer for noting this. As mentioned earlier, the main text now focuses
more clearly on the human dataset. Our rationale for not acknowledging the nuanced differences between
human and non-human datasets was that while the non-human datasets added breadth to our submission by
representing alternative and arguably higher fidelity connectome reconstruction techniques, they also suffered
from certain peculiarities, e.g. the macaque connectome is incomplete (full connection information on 29 of
91 regions) while the mouse and rat data represent single hemispheres. In any case, we now explicitly note
the differences between non-human and human connectome datasets.

In the section, **Connectomes support diverse meso-scale architecture**, we now write:

Figure 2: *C. elegans* connectivity matrix reordered by community assignments. (A) Each panel in the top row depicts the same connectivity matrix of chemical synapses among $N = 279$ neurons of the nematode *C. elegans*. Edge weights represent the number of synapses and are indicated by both the color and the size of each edge. The rows and columns of each matrix are ordered according to WSBM community assignments. Along both the x- and y-axes are colored plots. The color of each row and column represents the Q_{max} community assignment of the corresponding neuron. Note: in general, the Q_{max} assignments are heterogeneously distributed across WSBM communities, suggesting an inexact correspondence. (B) Panels in the bottom row depict the connection weight density of the blocks defined by the WSBM community assignments. Note that in general, connection density is not strongest along the diagonal, which would indicate assortative communities. Instead, the density of off-diagonal blocks is sometimes greater than that of the diagonal blocks, which indicates the presence of non-assortative communities.

- • “While these results were, overall, consistent in the non-human connectome datasets, there were
 nonetheless some differences (Fig. S5). For example, in the mouse dataset the relationship between
 node degree and change in regional assortativity was practically non-existent. The source of this vari-
 ation is unclear, though it is important to note that, while the non-human datasets are reconstructed
 using what are arguably higher-fidelity techniques, e.g. tract tracing, they nonetheless suffer from
 peculiarities, notably incompleteness. The macaque connectome includes connection data on only 29
 of 91 brain areas while the mouse and rat data include only a single hemisphere. For this reason, it is
 difficult to ascertain whether differences in connectome meso-scale structure across species arises due
 to genuine architectural differences or whether complete connectivity information would improve the
 consistency of results.”

We also address this issue in the section **Many (but not all) communities are assortative:**

- • “As in the previous section, while we find similar results in non-human connectome datasets, we also
 note some differences (Fig. S4). For instance, the *Drosophila* dataset is unique in that the relationship
 between node strength and the probability of being assigned to the maximally assortative set exhibits

a u-shaped curve (Fig. S4F). The macaque dataset exhibits a similarly-shaped curve, and possibly as a
consequence of where we drew the cutoff for rich club assignment or the incompleteness of the macaque
connectome, rich club nodes are actually more likely to be assigned to the maximally assortative set
than to non-rich club nodes (Fig. S4O).”

Finally, in the section **Community morphospace reveals rules for between-community interactions:**

- • “Again, these findings were largely replicated in non-human connectome data, though the relative
proportions of motif types was variable (Fig. S6). While the incompleteness of the non-human connec-
tome datasets make cross-species comparisons difficult, these differences raise the prospect that the
meso-scale structure of different organisms features nuanced, organism-specific motifs.”

**Comment 3**

*Your last paragraph of the “Connectomes support diverse meso-scale architecture” Results section seems to*
*be a bit strong of a conclusion given the analyses you conducted. You compared assortativity between Q_{max}*
*and WSBM methods, and should expect assortativity to be lower in the WSBM method based on its algorithm*
*versus that of Q_{max} , as you wrote earlier in that section. So this is more of a validation that the different*
*algorithms are doing what you think they are than that the differences are functionally meaningful. Adding*
*other network metrics that are thought to be important for cognition and comparing their values across*
*algorithms would make that conclusion more appropriate . . . or not making that claim until you relate these*
*two community structures to cognition or other metrics below.*

We agree with the reviewer that the results presented in the section **Connectomes support diverse**
**meso-scale architecture** offer no evidence that one method is better or worse than the other. The aim of
that section was to show that the WSBM detects communities whose character was fundamentally *different*
than those detected using Q_{max} (not necessarily better or worse). Specifically, whereas Q_{max} communities
are highly assortative and segregated, the WSBM detects communities that are more integrated thanks to
many cross-community links. We have softened the tone of this paragraph to more accurately reflect the
content of this section.

We note, however, that elsewhere in the **Results** section we leverage well-established methods and present
objective, quantifiable evidence documenting that the WSBM outperforms Q_{max} . Specifically, we show that
WSBM communities in the human and mouse connectomes partition functional connectivity and gene co-
expression networks into segregated modules (we discuss this later in our response to the reviewer). We
also derive a region-level diversity index and show that this index is predictive of subjects’ performances on
cognitive tasks. Importantly, when communities are assortative (as they are when detected using Q_{max}),
the diversity index of every brain region has a value of zero and the index is no longer informative. So while
we agree that the current section does not constitute evidence of superiority, the main findings of the paper
support the hypothesis that non-assortative community structure out-performs assortative communities along
neuroscientifically relevant dimensions.

We have revised the opening paragraph of this section accordingly:

- • “The brain’s meso-scale structure is generally assumed to be uniformly assortative (i.e., communities are
segregated from one another), a feature thought to support specialized information processing [17]. The
WSBM challenges this view, detecting less assortative (and hence increasingly integrated) communities,
suggesting that communities might play a more diverse range of functional roles. Demonstrating this
empirically, however, remains a challenge.”

We are also intrigued by the reviewer’s suggestion to test whether other network measures that have been
shown to correlate with behavioral measures also vary with community detection technique. Of course, it
is important to note that irrespective of the technique we use, the underlying network is fundamentally the
same, so any metrics we compare must be sensitive to changes in detected communities. One possibility is
the participation coefficient [18], which measures the extent to which a node’s links are concentrated within
its own community *versus* distributed evenly over other communities. Accordingly, we compared partitions
detected using both methods, Q_{max} and WSBM, in terms of their participation coefficient.

First, we separated partitions by number of communities, $k = 2, \dots, 10$. Then, for each k we calculated
the mean regional participation coefficient. We repeated this analysis for each method separately. To compare

the two techniques, we computed the Pearson correlation coefficient between the two $N \times 1$ vectors of mean
 regional participation coefficients. When k was small ($k = 2$ and $k = 3$), we found that the correlation was
 weak (Fig. 3). As k increased, however, the correlation also increased in magnitude. To some extent, this is to
 be expected. In the limiting case when $k = N$ (each node is in its own community), the algorithms converge
 to the same result. However, over the range $k = 2, \dots, 10$, the correlation between regional participation
 coefficient values is similar.

Figure 3: **Regional participation coefficient for WSBM and Q_{max} averaged across partitions.** Each panel depicts the mean regional participation coefficients for brain regions estimated from partitions detected using either the WSBM (x -axis) or Q_{max} (y -axis).

The previous analysis would seem to suggest that both techniques result in similar intuitions about
 nodes' roles in the network. However, this is not true. A participation coefficient close to 1 means that a
 node's connections are distributed almost evenly across communities, while a value close to 0 means that
 its connections are concentrated within own community. Therefore, we can calculate the mean participation
 coefficient over all brain regions to assess whether connections tend to fall within or between communities.
 When we perform this analysis, we see that the WSBM results in much greater average participation than
 Q_{max} (maximum $p < 10^{-97}$ over all k) indicating that while both techniques identify similar high and low
 participation brain regions, those values are consistently greater with the WSBM (Fig. 4). This means that
 connections tend to cross the boundaries of communities, which aligns exactly with the results we reported
 in the main text.

Figure 4: **Boxplots of participation coefficient for WSBM and Q_{max} averaged across brain regions.** For a given number of communities, k , the WSBM consistently results in higher participation coefficients than Q_{max} .

**Comment 4**

*Similarly, your first paragraph of the “Functional relevance of the WSBM” section “To this point, we*
 *have used the WSBM to demonstrate that connectomes exhibit diverse, non-assortative meso-scale struc-*
 *ture...suggesting that the richer, non-assortative communities are closer to the brains canonical architecture.”*
 *seems like a straw man, since the algorithm that allows for non-assortative meso-scale structure finds some*
 *non-assortative communities, while the algorithm that limits its results to maximizing assortativity finds all*
 *assortative communities. This argument (that is scattered throughout the manuscript) needs an explanation*
 *regarding why maximizing assortativity to the exclusion of other types of communities is not as biologically*
 *plausible as allowing multiple types of community structure. There have been many papers arguing why algo-*
 *rithms maximizing modularity are biologically plausible. This concept that we are limiting our investigations*
 *by doing so is very compelling, but it needs more biological motivation (as well as functional relevance ... see*
 *other comments about that below) to have a strong impact on the field.*

We agree with the reviewer that the opening paragraph of this section is a bit strong. We have revised
 it to read:

- • “To this point, we have used the WSBM to characterize the meso-scale structure of human and non-
 human connectomes. Using this method, we find a diverse meso-scale structure that includes non-
 assortative communities.”

The reviewer’s second point – why might we expect a neural system to have non-assortative communities?
 – is a critical one. The hypothesis that we posit in the introduction can be summarized as follows. Assortative
 meso-scale structure is beneficial for networks whose sub-systems are intended to function in isolation and
 autonomously from one another. If the brain is organized into strictly assortative communities it suggests
 that all communities are used in this same way, namely to carry out specialized information processing. On
 the other hand, we think that the integration of information from many systems is a critical component of
 higher-order cognitive processes, mentalizing, and planning [19]. In order to accomplish this, communities
 need to interact with one another. That is, the brain’s meso-scale must deviate (even if only slightly) from
 the strictly assortative model that Q_{max} imposes upon it. Following this reasoning, we hypothesized that
 non-assortative interactions among communities help the brain to support complex cognitive processes.

The reviewer is also correct that the network neuroscience literature lacks a balanced discussion of assort-
 ative *versus* non-assortative communities. Though we can only speculate on why this is the case, one likely
 contributor is the fact that modularity maximization and infomap algorithms are fast, easily implemented,
 and already pervasive throughout network neuroscience research. These factors could effectively reinforce
 the assumption that brain communities “should” or “must” be assortative, spurring further empirical and
 theoretical research on that type of meso-scale structure.

Nonetheless, there are many compelling theoretical and empirical findings suggesting that non-assortative

communities confer advantages to neural systems and represent more accurate models of their network
organization. The mammalian visual system for instance, exhibits feed-forward like structure (a chain-like,
non-assortative topology) as it encodes progressively more abstract details of visual information. This type
of organization is evident at the inter-areal level [20] but also at the microscale, where retinal neurons are
wired according to cell-specific, distance-dependent, and function-driven rules [21, 22]. Elsewhere, analysis of
*C. elegans*' meso-scale structure using mixture models (a relative of the stochastic blockmodel) revealed non-
assortative communities, including a "rich" community composed of highly-connected inter-neurons known to
play critical roles in mechanosensation and locomotion, highlighting the community's apparent role in control
of behavior [9]. Similarly, the inter-areal neurochemical rat connectome exhibits core-periphery organization,
where the core is composed of serotonergic structures associated with sleep-wake cycles, arousal, and stress,
suggesting a "pacemaker"-like role for the core [23]. This same kind of organization has been observed in
mouse and macaque connectomes, where the cores are dominated by associative areas, again suggesting that
cores and non-assortative structures play pivotal roles in integrative neural processes [24].

These same studies [23, 24] also demonstrated that the core-periphery organization is compatible with
an exponential distance rule, in which the connection strength decreases with length [25]. Concurrently, a
recent modeling study of human connectome data suggested the existence of a non-assortative "geometric
core" composed of brain regions that emerges naturally under wiring cost constraints [26]. Collectively
these findings suggests that, like networks with assortative communities, core-periphery organization can be
embedded efficiently in three-dimensional space so as to reduce the network's total cost of wiring.

Lastly, while we claim that modularity maximization precludes the possibility of detecting non-assortative
communities, we make no claim that the brain exhibits *strictly* non-assortative communities. We devote a
full subsection to this topic: **Many (but not all) communities are assortative**. In the human connec-
tome, in fact, we find both assortative and core-periphery community motifs, but very few disassortative
(bipartite) interactions. So while modularity maximization might miss out on some of the richness of the
brain's community structure, it is possible that communities detected with the WSBM might retain many
of the functional and evolutionary advantages associated with assortative communities (e.g. efficient spa-
tial embedding, robustness to perturbations, etc.), while non-assortative communities increase the network's
diversity and confer additional functionality to the network.

We have revised the manuscript to reflect the above statements. In the introduction, we restate our
hypotheses more clearly and in the discussion we include a paraphrased version of the above paragraphs.

- • "Secondly, this view implies that the connectome's meso-scale structure is rigidly uni-functional. That
is, networks with assortative communities are well-poised for specialized, segregated information pro-
cessing, but are not suited for integrative function. Higher order cognitive processes, for example, are
thought to emerge through integration of information originating in different brain systems [19], which
can only occur *via* the interaction of communities with one another. We hypothesize, then, that in order
to produce complex thought and adaptive behavior, the brain's underlying meso-scale architecture
must deviate (even if only slightly) from the strictly assortative model."
- • "These findings build upon and extend other recent studies reporting non-assortative structure in
connectome data. The mammalian visual system for instance, exhibits feed-forward like structure (a
chain-like, non-assortative topology) at both the inter-areal level [20] and at the microscale [21, 22]. A
previous analysis of *C. elegans*' meso-scale structure using mixture models (a relative of the stochas-
tic blockmodel) revealed a core-like community composed of highly-connected inter-neurons known to
play critical roles in mechanosensation and locomotion, highlighting its apparent role in the control of
behavior [9]. Similarly, the inter-areal mouse, rat, and macaque connectomes exhibit core-periphery
organization, where the core is composed of associative brain areas and proposed to act as a "pace-
maker" [24, 23]. Moreover, this type of architecture is consistent with wiring-cost reduction models,
suggesting that core-periphery structure, like assortative communities, can be efficiently embedded in
three-dimensional space [25]."

Comment 5

*You state later in that section that "Intuitively, functionally-related brain regions are linked by strong func-*
*tional connections. If a community does a good job identifying sets of such regions, then the within-community*

*density of functional connections should be greater than the between-community density.” But the point of*
 *the manuscript is that assuming communities have this strong-within and weak-between connectivity is not*
 *necessarily correct, and we should allow for other types of communities (Figure 1). That sentence seems to*
 *contradict the main point of the manuscript. It is very possible that I am missing something, but regardless*
 *that concept should be clarified so as to not seem contradictory if it isnt. Further, the result that follows*
 *appears to show that even with WSBM, when averaging across the whole brain, communities are assortative*
 *(strong within, weaker between). It is also unclear to me how both within and between-community connections*
 *can be more dense if the nodes and edges are matched across algorithms, so more explanation about that is*
 *needed as well.*

We agree with the reviewer that, as written, our previous explanation for why we expect the within-
 community density of *functional connections* to be greater than between-community density (even if com-
 munities are non-assortative) was only weakly justified. We take this opportunity to detail our rationale.

Figure 5: **Matching index.** We show two example adjacency matrices: one for a bipartite network and another with assortative communities. We compute the matching index for all pairs of nodes to show that both networks, despite their vastly different connectivity patterns, result in similar patterns of matching index.

In past work when empirical estimates of FC could not be easily obtained, the similarity of brain region’s
 structural connectivity profiles (as measured by the “matching index”) was treated as a measure of their
 functional connectedness [27, 28, 29]. Importantly, the matching index can be strong between disconnected
 regions, so long as their inputs and outputs are similar. This implies that even bipartite communities with
 no internal structural connections will have strong within-community matching. We demonstrate this using
 two toy networks: one with bipartite communities and the other with assortative communities (Fig. 10). The
 point of this demonstration is to show that there is both an historical and structural rationale for expecting
 high levels of within-community FC in both assortative as well as non-assortative communities.

[revised manuscript text omitted]

Comment 6

“Community morphospace reveals rules for between-community interactions”: how do the previously-described results offer “a better explanation for human brain function and mouse genetic expression than that provided by assortative communities alone”? Thus far, I am convinced that it is “different”, but have seen no evidence that it is “better”.

This is an important point. In general, it is difficult to demonstrate conclusively that one community detection method is objectively better than another and, of course such a demonstration depends on how one defines “better”. However, cross-validation through meta-data represents a powerful technique for objectively and quantitatively comparing different methods [30, 31]. By drawing on domain-specific knowledge of how the structure and function of neural systems are related to one another, we formed hypotheses and designed objective functions that, when evaluated for both the WSBM and Q_{max} , clearly favored the WSBM over Q_{max} . In the text, we have also explicitly defined what we mean when we use the term “better”.

We have now edited the main text to better emphasize these points.

- “We can compare these two hypotheses of brain function with cross-validation methods using empirical functional connectivity as metadata [30, 31]. We reasoned that if functional connectivity emerges from interactions among brain regions in independent, autonomous clusters, then its organization will be closely aligned to the communities detected using Q_{max} . On the other hand, if functional connectivity is the result of non-assortative, integrated clusters, then the WSBM communities will more closely resemble the brain’s functional connectivity. To compare communities with functional connectivity, we classified every functional connection as “within-community” or “between-community”. We calculated the mean weight of all connections assigned to each class and finally the difference between those values. This measure – the difference between mean within- and between-community functional connections – serves as a measure with which we can evaluate the performance of the two algorithms.”

Comment 7

Also in that section, your description of the results is that the WSBM algorithm identifies a significant number of core-periphery interactions (which I see in Figure 6) and disassortative interactions (which I do not see). It looks to me that the proportion of disassortative interactions using WSBM, and the difference between WSBM and Q_{max} , are both 0. Please clarify in the text and the figure what you mean by this.

We thank the reviewer for bringing this to our attention. Overall, the WSBM uncovers many more assortative and core-periphery motifs than disassortative motifs. In fact, from $K = 2$ to $K = 4$ it uncovers exclusively assortative and core-periphery motifs. From $K = 5$ to $K = 10$ the relative proportion of disassortative motifs is always less than 2%, and in Figure 6, amounts to exactly 0.2%. In that figure, there are a small number of red points (representing the disassortative motifs), but they are difficult to see because of the large number of yellow and cyan points and because of the beige background. In the revised manuscript, we have changed the opacity of the red points to make them more visible (Fig. 6).

Comment 8

Your brain figures exclude subcortical brain regions, yet they keep popping up in the article as diverse and one of the groups of regions that would be better defined using WSBM. Please add images of the subcortical structures as well (this goes for all figures that show where in the brain different node types are).

We agree with the reviewer that this is a useful visualization. We now depict sub-cortical structures alongside the surface plots.

Figure 6: **A rich community morphospace.** (A) A community motif is constructed as the average over blocks of the connectivity matrix. Here, we show blocks within and between two communities, labeled r and s . (B) Given within- and between-community connection densities, it is possible to classify each pair of communities into one of three motifs: assortative, disassortative, or core-periphery. (C,D) All pairs of communities placed in a network morphospace and colored by their motif type. Note: axes are log-scaled. (E) The relative proportion of each motif type as a function of the number of detected communities, K , for Q_{max} (left), the WSBM (middle), and their difference (right).

Comment 9

Behavioral relevance section: it would be interesting to do this analysis using Q_{max} to define community
 structure as well, to determine which of the two algorithms best explains behavior. If WSBM is more related
 to behavior on these cognitive control tasks, that is an argument for the functional relevance (and potentially
 higher accuracy) of community structure derived from that algorithm.

We agree that, in principle, this would be interesting and make for a more compelling comparison.
 However, because the Q_{max} algorithm uncovers only assortative community interaction motifs, the diversity
 of every region (measured as an entropy) is zero. We now note this in the main text at the end of the section
 **Community motifs identify a class of diversely connected nodes:**

- • “Note that because Q_{max} uncovers only assortative community motifs, each brain region’s diversity
 score is effectively zero. Accordingly, we never assessed the distribution of diversity scores for the Q_{max}
 partitions over functional systems.”

Comment 10

In your first sentence of the discussion, while I dont disagree that having different types of communities
 is plausible, I have not seen a strong argument for that other than that an algorithm that attempts to find
 these different types of communities can. This to me is no less biased than modularity maximization being
 biased to find an assortative community structure. This argument should be made more explicit and linked
 to biology throughout the manuscript to support your conclusions.

The last full sentence on page 12 is the first I have seen that gives a biological reason for looking for
 non-assortative community structure. If this and other related arguments are highlighted more in both the
 introduction and discussion, this paper will be more convincing.

These two comments deal with the same concern (which was also raised in the reviewer’s **Comment 4**)
 and so we respond to them with a single, cohesive reply. In addition to more fully fleshing out the sentence

on page 12 that the reviewer refers to, we now include a lengthier explanation of other studies that have
found some evidence of non-assortative structure in connectome datasets.

• “These findings build upon and extend other recent studies reporting non-assortative structure in
connectome data. The mammalian visual system for instance, exhibits feed-forward like structure
(a chain-like, non-assortative topology) at both the inter-areal level [20] but also at the microscale
[21, 22]. A previous analysis of *C. elegans*’ meso-scale structure using mixture models (a relative
of the stochastic blockmodel) revealed a core-like community composed of highly-connected inter-
neurons known to play critical roles in mechanosensation and locomotion, highlighting its apparent
role in the control of behavior [9]. Similarly, the inter-areal mouse, rat, and macaque connectomes
exhibit core-periphery organization, where the core is composed of associative brain areas and proposed
to act as a “pacemaker” [24, 23]. Moreover, this type of architecture is consistent with wiring-cost
reduction models, implying that core-periphery structure, like assortative communities, can be efficiently
embedded in three-dimensional space [25].”

Comment 11

*Your sentence: “Moreover, communities detected using the weighted stochastic blockmodel better recapitulated*
*observed intrinsic functional relationships among brain regions in the human, and relationships between*
*gene co-expression patterns in the mouse, compared to more commonly-used techniques such as modularity*
*maximization.” See my point above about this results section. This argument does not logically follow to me.*

This sentence was intended to be a summary of the metadata cross-validation of the communities. In
that analysis we showed that compared to Q_{max} , WSBM communities were enriched for strong functional
connections and correlated patterns of gene expression in human and mouse, respectively. In line with our
reply to the reviewer’s **Comment 6**, we have revised this sentence to better clarify its intended meaning.

• “Moreover, by cross-validating communities using metadata (a technique that has been employed
elsewhere [30, 31]), we showed that meso-scale structure uncovered by the WSBM was more closely
aligned with functional connectivity compared to Q_{max} .”

As we noted in our reply to **Comment 4**, we also include a longer description explaining why the
cross-validation is an appropriate method for objectively comparing the WSBM with Q_{max} :

• “It is generally agreed upon that brain structural connectivity determines the partners that any given
region can “talk to”, and therefore constrains communication patterns among brain regions, shaping
the correlation pattern of ongoing neural activity, i.e. functional network organization. We reasoned
that if two brain regions receive input from the same set of brain regions and deliver output to the
same set of regions, then their activity over time should be correlated, i.e. those regions would appear
functionally connected to one another. This set of assumptions has a long tradition in the network
neuroscience community. In the past when empirical estimates of FC could not be easily obtained,
measures of similarity between brain regions’ connectivity profiles (e.g., matching index) have been
used as a stand-in [27, 28, 29].

Though through different mechanisms, both the WSBM and Q_{max} produce communities of brain
regions with similar patterns of connections. However, these methods differ in that communities
are defined according to two vastly different topological principles. Q_{max} assumes that the brain’s
meso-scale organization is based on assortative and segregated sub-systems, while the WSBM allows
communities to be both assortative and non-assortative. These differences in meso-scale structure
imply differences in brain function. A strictly assortative brain is aligned with the hypothesis that
the brain is composed of communities operating nearly autonomously, while a brain composed of some
non-assortative communities implies that brain function arises not from independent communities, but
from the interactions between communities.

Here, we test these two hypotheses by cross-validating and comparing WSBM and Q_{max} partitions
using empirical FC as metadata (See **Materials and Methods** for more details on FC reconstruction
from BOLD signals). This approach – cross-validation through metadata – is well-established and has

been used extensively in past studies [31, 30]. In essence, it assumes that metadata better represents
some aspect of a network’s ground truth organization than its structural topology alone. Community
detection methods that are more closely aligned with the metadata may be more sensitive to the
network’s ground truth organization and are considered, in this quantitative and objective sense,
superior to those that do not. We reasoned that if the brain’s correlated activity pattern is better
described by assortative communities behaving autonomously, then the FC network will be more closely
aligned with Q_{max} communities. On the other hand, if the correlation pattern is better described by
interacting, non-assortative communities, the alignment of FC to WSBM communities will be greater.
Here, we quantify this alignment as the mean weight of within-community functional connections minus
the mean weight of between-community functional connections.”

- • “We note that the use of a Pearson correlation as a measure of FC results in increased transitivity (if
a strong correlation exists between nodes A and B as well as B and C, then A and C will tend to be
strongly correlated), which can reinforce block structure in correlation matrices [32].”

Reviewer #2

*“Diversity of meso-scale architecture” by Betzel et al. compares two forms of community decompositions*
*applied to structural connectomic data a classic decomposition that maximises the assortativity (Q_{max}) of*
*the decomposition with a more recently developed weighted stochastic block algorithm (WSBM). The WSBM*
*appears to group nodes into modules such that the likelihood of inter-module connections is approximately*
*even for all nodes within each module (i.e. does not depend heavily upon each individual module). There*
*is substantial interesting material in this manuscript which is clearly written and nicely illustrated.*

We thank the reviewer for the kind comments.

Comment 1

*One of the major findings is not really that surprising and I think more work is required to establish the*
*significance of the finding. Namely, it is not surprising that any decomposition that does not maximise the*
*assortativity of a module decomposition is less assortative than one that does! i.e. there is an element of*
*circularity in the central finding of the paper. The question that seems not to be answered is not whether*
*the WSBM decomposition reveals a less assortative community structure than Q_{max} , but whether the degree*
*of core-periphery and disassortativity in the human connectome is greater or lesser than expected by chance:*
*Since we already know that Q_{max} -sorted connectomes are more assortative than degree and strength preserv-*
*ing random surrogates, we should now check whether WSBM ones are likewise more core-periphery or more*
*disassortative than strength-sequence matched surrogates. Are they likewise more or less arranged when such*
*surrogates also account for the geometric embedding of real connectomes into three dimensions? Only after*
*this has been established, do I think the more fine-grained analyses of figure 3-5 make sense.*

We thank the reviewer for these comments and for the opportunity to clarify our aims. The reviewer
brings up two points – one related to the central question of the paper “does the brain have strictly assortative
communities and can we find them?” and a second question related to structural null models. We address
these two separately, starting with the question on null models.

We agree that applying the WSBMs to some variation of a randomly rewired network would serve as an
important control, and that a strength-preserving null model is a good place to start. In general, randomly
rewiring a network will decrease the frequency of triangle motifs – nodes a , b , and c that are mutually
connected. These sorts of triangles inflate a network’s clustering coefficient, “fill out” modules, and lead to
the formation of assortative communities. Accordingly, if we rewire a network and destroy its triangles, we
would actually expect a decrease in assortative communities. Accordingly, we expect an increase in non-
assortative communities as a result of rewiring. We confirm this hypothesis using two separate tests (and
in the process, we show that observed networks have different community statistics than randomly-rewired
networks).

First, we generated 100 rewired networks (for the human network) in which we preserved degree sequence
exactly but allowed strength sequence to vary [33]. Then, using a simulated annealing algorithm, we shuffled
edge weights until the node strength sequence was almost exactly preserved (in general, it is not possible

to preserve a precise set of edge weights and the strength sequence exactly). Over the 100 realizations, the
 average correlation of the empirical network’s strength sequence with those of the rewired networks was
 $r = 0.999994 \pm 0.000002$, suggesting excellent correspondence. We then used the WSBM to cluster each of
 the 100 networks, repeating the optimization 10 times for each K of the range $K = 2$ to $K = 10$ (we used
 10 optimizations in place of the 250 in the main text to reduce total runtime).

Figure 7: **Summary of strength-preserving null models.** (A) Scatterplot showing strong correlation of observed and randomized strength sequences. (B) Mean connectivity matrix obtained by averaging over all 100 realizations of the null model.

As a first comparison, we calculated community assortativity as a function of community size (Fig. 8).
 The assortativity scores are computed using partitions detected based on rewired networks. The construction
 of this figure is identical to Figure 2B in the main text. For ease of interpretation, we only plotted the mean
 community assortativity curves. As expected, applying the WSBM to randomly rewired networks resulted
 in communities that were less assortative than when applied to the observed network, and far less assortative
 than those detected when Q_{max} was applied to the observed network. This demonstrates that communities
 detected using both the WSBM and Q_{max} are distinct from one another and also differ from randomly
 rewired networks.

As a second comparison, we computed the morphospace of community interaction motifs based on com-
 munities detected in the rewired networks. As in the main text, this involved generating for each network
 and partition a set of community interaction motifs and classifying them as either “core-periphery”, “assor-
 tative”, or “disassortative” (Fig. 9A). We then calculate the proportion of each motif type as a function of
 the number of communities. We show these proportions for the rewired networks (Fig. 9B), the observed
 network (Fig. 9C), and the difference between the two (Fig. 9D). As expected, the rewired networks exhib-
 ited far fewer assortative motifs than the observed network and far more core-periphery and disassortative
 motifs.

These additional analyses demonstrate that observed brain networks exhibit different community statis-
 tics compared to rewired brain networks. These findings inform the results in the main text. The application
 of the WSBM to brain networks results in less assortative communities than if Q_{max} had been used to detect
 communities. This level of disassortativity, however, is not as severe as that observed in random networks,
 suggesting that the observed brain networks, in fact, maintain an unexpected level of assortative commu-
 nities. This is an important point, as the functional and evolutionary advantages of assortative community
 structure have been well-documented [17], indicating that brains may balance these advantages with other
 additional advantages conferred by possessing a small proportion of non-assortative communities.

The results of these analyses have now been summarized and added to the **Supplementary Materials**
 under the subsection entitled **Application of the WSBM to rewired networks:**

- • “The WSBM is a flexible tool for detecting communities in networks using statistical inference. To

Figure 8: **Community assortativity comparisons.** Mean community assortativity curves as a function of community size for Q_{max} and the WSBM applied to the observed network and then for the WSBM applied to the strength-preserved null randomized networks.

properly contextualize the results presented in the main text, we applied the blockmodel to randomly rewired networks. Specifically, we compared the observed brain network to networks with precisely the same degree sequence and approximately the same strength sequence. This process entailed first using a standard edge rewiring algorithm to rewire the observed network while preserving its exact degree sequence [33]. However, this procedure does not preserve nodes' strengths. To approximate the observed strength sequence, we randomly swapped the weights of existing edges and, using a simulated annealing algorithm, gradually found configurations of edge weights such that nodes' strengths were minimally different from that of the observed network (Fig. 7A). We repeated the algorithm 100 times, generating 100 realizations of the rewired network (Fig. 7B).

We then used the WSBM to uncover the mesoscale structure of each rewired network. We varied the number of communities from $k = 2$ to $k = 10$ and repeated the algorithm 10 times. Next, we calculated the assortativity of each detected community based on the connection pattern of the rewired network. We found that in randomly rewired networks, the assortativity of communities detected using the WSBM was far less than that of the observed network. This is because the rewiring procedure tends to reduce the number of triangles and cliques in the network. Because these structures reinforce assortative communities, a reduction in their prevalence corresponds to a reduction in the overall

Figure 9: **Community morphospace for strength preserved randomized network.** (A) Each point represents a two-community motif classification. (B) and (C) Proportion of community interaction types at different numbers of communities, $k = 2$ to $k = 10$, for the randomized and observed networks. (D) The difference in community interaction type proportions.

assortativity of communities (Fig. 8). We traced out the average assortativity of communities as a function of community size and compared the resulting curves using functional data analysis. We found that the assortativity of communities in the observed network was significantly greater than that of the communities detected in the rewired networks ($p < 10^{-4}$).

Next, we submit the rewired networks to a morphospace analysis. As in the main text, this process entailed enumerating and classifying all two-community interaction motifs as “assortative”, “core-periphery”, or “disassortative” (Fig. 9A). This process was repeated as we varied the number of communities from $k = 2$ to $k = 10$. For each k , we calculated the proportion of motifs within each class (Fig. 9B-D). From this analysis we found that the rewired networks resulted in a decrease in the fraction of assortative motifs. In parallel, this reduction in assortative motifs was accompanied by an increase in core-periphery and disassortative motifs.

These additional analyses demonstrate that observed brain networks exhibit different community statistics compared to rewired brain networks. These findings inform the results in the main text. The application of the WSBM to brain networks results in less assortative communities than if Q_{max} had been used to detect communities. This level of disassortativity, however, is not as severe as in random networks, suggesting that the observed brain networks, in fact, maintain an unexpected level of assortative communities. This is an important point, as the functional and evolutionary advantages of assortative community structure have been well-documented [17], indicating that brains may balance these advantages with whatever additional advantages are conferred by possessing a small proportion of non-assortative communities.”

We also discuss these results and their implications in the main text in the **Discussion** section:

- “Moreover, we also show in the supplementary section **Application of the WSBM to rewired networks** that the diversity of communities in the observed brain networks is distinct from that of rewired controls.”

The reviewer also raised a question about the central topic of the paper, namely the existence of non-
assortative community structure in brain networks. We take this opportunity to clarify our aims, and to
restate and restructure our main arguments.

Our motivation for writing this paper was as follows: Our current view of the brain’s meso-scale archi-
tecture might be biased by heavy use of Q_{max} and related algorithms. We felt it necessary to present an
alternative view of the brain’s meso-scale architecture in which we apply a relatively new method (WSBMs,
in this case) capable of detecting more general types of communities. We also wanted to, if possible, demon-
strate the superiority of one method over the other. To this end, we used a well-established cross-validation
procedure in which we compared communities to metadata [31, 30]. This approach assumes there exists
some form of independent metadata at either the level of network nodes or edges that captures a network’s
ground-truth organization better than its structural communities, i.e. divisions of the network estimated
from its topology alone. In our case, we used human resting state functional connectivity and mouse gene
co-expression patterns. We demonstrated that in both cases, communities estimated from the WSBM better
matched the organization of these metadata, indicating that not only did WSBM and Q_{max} communities
differ, but along these dimensions the WSBM was objectively better.

Reviewer #1 raised a similar point, prompting us to write a better explanation of the cross-validation
method, including the underlying assumptions and interpretation of the results. This explanation now
appears in the **Results: Functional relevance of the WSBM:**

•

- • “It is generally agreed upon that brain structural connectivity determines the partners that any given
region can “talk to”, and therefore constrains communication patterns among brain regions, shaping
the correlation pattern of ongoing neural activity, i.e. functional network organization. We reasoned
that if two brain regions receive input from the same set of brain regions and deliver output to the
same set of regions, then their activity over time should be correlated, i.e. those regions would appear
functionally connected to one another. This set of assumptions has a long tradition in the network
neuroscience community. In the past when empirical estimates of FC could not be easily obtained,
measures of similarity between brain regions’ connectivity profiles (e.g., matching index) have been
used as a stand-in [27, 28, 29].

Though through different mechanisms, both the WSBM and Q_{max} produce communities of brain
regions with similar patterns of connections. However, these methods differ in that communities
are defined according to two vastly different topological principles. Q_{max} assumes that the brain’s
meso-scale organization is based on assortative and segregated sub-systems, while the WSBM allows
communities to be both assortative and non-assortative. These differences in meso-scale structure
imply differences in brain function. A strictly assortative brain is aligned with the hypothesis that
the brain is composed of communities operating nearly autonomously, while a brain composed of some
non-assortative communities implies that brain function arises not from independent communities, but
from the interactions between communities.

Here, we test these two hypotheses by cross-validating and comparing WSBM and Q_{max} partitions
using empirical FC as metadata (See **Materials and Methods** for more details on FC reconstruction
from BOLD signals). This approach – cross-validation through metadata – is well-established and has
been used extensively in past studies [31, 30]. In essence, it assumes that metadata better represents
some aspect of a network’s ground truth organization than its structural topology alone. Community
detection methods that are more closely aligned with the metadata may be more sensitive to the
network’s ground truth organization and are considered, in this quantitative and objective sense,
superior to those that do not. We reasoned that if the brain’s correlated activity pattern is better
described by assortative communities behaving autonomously, then the FC network will be more closely
aligned with Q_{max} communities. On the other hand, if the correlation pattern is better described by
interacting, non-assortative communities, the alignment of FC to WSBM communities will be greater.
Here, we quantify this alignment as the mean weight of within-community functional connections minus
the mean weight of between-community connections.”

Comment 2

*I doubt as presented, many readers will understand what the WSBM actually does do, other than being “not*
*maximally pro-assortative”. There is a brief heuristic for the standard block model that seeks to minimise*
*a cost function that penalises the heterogeneity of inter-module connectivity amongst nodes within the same*
*module (that is my reading of equation (1)). The WSBM is more complex, but I assume it is a generalisation*
*of the same principle to accommodate edge weights. I think the authors should provide a better heuristic*
*explanation of the WSBM minimization and put it at the beginning of the Results so that the general reader*
*can understand what is being optimized.*

We thank the reviewer for this point. We have now included a better intuitive description of the WSBM
when we first introduce it to the reader. A longer, more detailed description is included in the methods
section.

- • “Briefly, the WSBM assumes that a network’s nodes can be partitioned into communities and that
both the probability of a connection forming between two nodes and the weight of that connection
are governed by parameterized generative processes. Importantly, these processes depend only on the
communities to which two nodes are assigned. Using the WSBM to uncover a network’s community
structure involves inferring both the parameters of these processes and nodes’ community assignments
that maximize the log-evidence that the WSBM generated the observed network. The resulting com-
munities, therefore, reflect similarities in nodes’ connectivity profiles and are not constrained to be
assortative.”

Comment 3

*I also think the paper would be improved with some groundtruth validations, using growth models, to show that*
*the WSBM algorithm does what the authors assume it does namely that Q_{max} and WSBM should converge*
*if applied to networks that are generated by suitable preferential attachment models, that add new edges to*
*maximize assortativity (the authors are leading experts in such generative modelling); Also that they should*
*strongly diverge when such networks are grown to maximise core-periphery arrangements and for maximally*
*disassortative networks.*

Ground truth validations are essential for any new model to ensure that it does what its creators claim
it does. As we note in the submission, the WSBM is not an entirely novel method (though its application
to connectome data is) and has existed in the literature for several years [34, 35]. We refer the reviewer
to these manuscripts, which introduce the WSBM and in which the authors perform extensive validation
on synthetic and real-world networks, demonstrating that the WSBM is capable of detecting generalized,
blockwise communities (assortative or otherwise) in weighted and directed networks.

Comment 4

*I found the choice of the statistic for comparing functional and structural connectivity somewhat counter-*
*intuitive: Namely of seeing if the corresponding functional networks (when grouped into the corresponding*
*modules) were maximally assortative. Its interesting, although counter-intuitive that WSBM outperforms*
*Q_{max} on this metric. The authors should likely also be cautious that network measures based on linear*
*correlations induce artificial assortativity through the nature of the measure [1].*

This concern is similar to Comment 5 from Reviewer #1. We use this opportunity to restate our reasoning
for why even non-assortative communities should be internally dense in terms of functional connections.

In past work when empirical estimates of FC could not be easily obtained, the similarity of brain region’s
structural connectivity profiles (as measured by the “matching index”) was treated as a measure of their
functional connectedness [27, 28, 29]. Importantly, the matching index can be strong between disconnected
regions, so long as their inputs and outputs are similar. This implies that even bipartite communities with
no internal structural connections will have strong within-community matching. We demonstrate this using
two toy networks: one with bipartite communities and the other with assortative communities (Fig. 10). The
point of this demonstration is to show that there is both an historical and structural rationale for expecting
high levels of within-community FC in both assortative as well as non-assortative communities.

Figure 10: **Matching index.** We show two example adjacency matrices: one for a bipartite network and another with assortative communities. We compute the matching index for all pairs of nodes to show that both networks, despite their vastly different connectivity patterns, result in similar patterns of matching index.

Though through different mechanisms, both the WSBM and Q_{max} produce communities composed of
 brain regions with similar patterns of incoming and outgoing connections. In the case of Q_{max} , this similarity
 is entirely incidental – nodes get grouped into internally dense clusters and as a result they tend to be mutually
 connected, inflating the similarity of their connectivity profiles. The WSBM, on the other hand, assumes
 that the connectivity profiles of the nodes that make up a community are generated by the same statistical
 process and, by definition, should be similar to one another.

Because both methods result in communities composed of nodes with similar connectivity profiles, and
 because this similarity is associated with strong functional connectivity, we expect that two nodes in the same
 community should be more strongly functionally connected to one another than two nodes in different com-
 munities. However, both methods *also* define communities according to two vastly different organizational
 principles. Q_{max} assumes that the brain’s meso-scale organization is based on assortative and segregated
 sub-systems while the WSBM assumes that communities can be segregated, but that they can also form
 cores and peripheries, and sometimes disassortative structures as well. These differences in network orga-
 nization imply differences in brain function, too. A strictly assortative brain is aligned with the hypothesis
 that the brain is composed of communities operating nearly autonomously, while a brain composed of some
 non-assortative communities implies that brain function arises not from independent communities, but from
 the interactions between communities.

We can test these two hypotheses of brain organization and function through cross-validation using
 empirical functional connectivity as metadata. We reason that if the brain’s correlated activity pattern is
 better described by assortative communities behaving autonomously, then the functional network will be
 more closely aligned with those communities. We measure this alignment as the mean weight of within-
 community functional connections minus the mean weight of between-community connections. We can
 compute a similar measure to assess the functional network alignment to WSBM communities. We can
 compare these two measurements to support the claim that one or the other method is better aligned with

the brain’s functional architecture.

We have added the following text to the section **Functional relevance of the WSBM** to better reflect
our assumptions and hypotheses:

- • “It is generally agreed upon that brain structural connectivity determines the partners that any given
region can “talk to”, and therefore constrains communication patterns among brain regions, shaping
the correlation pattern of ongoing neural activity, i.e. functional network organization. We reasoned
that if two brain regions receive input from the same set of brain regions and deliver output to the
same set of regions, then their activity over time should be correlated, i.e. those regions would appear
functionally connected to one another. This set of assumptions has a long tradition in the network
neuroscience community. In the past when empirical estimates of FC could not be easily obtained,
measures of similarity between brain regions’ connectivity profiles (e.g., matching index) have been
used as a stand-in [27, 28, 29].

Though through different mechanisms, both the WSBM and Q_{max} produce communities of brain
regions with similar patterns of connections. However, these methods differ in that communities
are defined according to two vastly different topological principles. Q_{max} assumes that the brain’s
meso-scale organization is based on assortative and segregated sub-systems, while the WSBM allows
communities to be both assortative and non-assortative. These differences in meso-scale structure
imply differences in brain function. A strictly assortative brain is aligned with the hypothesis that
the brain is composed of communities operating nearly autonomously, while a brain composed of some
non-assortative communities implies that brain function arises not from independent communities, but
from the interactions between communities.

Here, we test these two hypotheses by cross-validating and comparing WSBM and Q_{max} partitions
using empirical FC as metadata (See **Materials and Methods** for more details on FC reconstruction
from BOLD signals). This approach – cross-validation through metadata – is well-established and has
been used extensively in past studies [31, 30]. In essence, it assumes that metadata better represents
some aspect of a network’s ground truth organization than its structural topology alone. Community
detection methods that are more closely aligned with the metadata may be more sensitive to the
network’s ground truth organization and are considered, in this quantitative and objective sense,
superior to those that do not. We reasoned that if the brain’s correlated activity pattern is better
described by assortative communities behaving autonomously, then the FC network will be more closely
aligned with Q_{max} communities. On the other hand, if the correlation pattern is better described by
interacting, non-assortative communities, the alignment of FC to WSBM communities will be greater.
Here, we quantify this alignment as the mean weight of within-community functional connections minus
the mean weight of between-community connections.”

- • “We note that the use of Pearson correlation as a measure of FC results in increased transitivity (if
a strong correlation exists between nodes A and B as well as B and C, then A and C will tend to be
strongly correlated), which can reinforce block structure in correlation matrices [32].”

**Comment 5**

*Novelty: As the authors cite, there is already substantial work using SBMs in human connectomes. A brief*
*summary of what is new here would help. Also, there are elements of the current findings that could be*
*unpacked from prior work: for example, [2] previously showed that rich club nodes preferentially existed*
*as apex nodes in open motifs of 3 nodes (that is maximally disassortative motifs [3]). Also, Figure 10a*
*of [4] shows that very high degree nodes are actually less often inter-connected than by chance i.e. have*
*a “cartel-like” disassortative property. The present finding, by very different methods, seems reassuringly*
*convergent.*

This is a good suggestion. Our paper makes several important contributions above and beyond the
papers that the author cites and past applications of SBMs to connectome data. Though there are several
papers that used variations of the SBM with unweighted structural connectome data from other species, e.g.
*C. elegans* [9], and others that have used blockmodels with human functional connectivity networks, e.g.
[36, 37, 38, 39, 40], to our knowledge there are no papers that have applied the SBM (weighted or otherwise)

to human *structural connectivity* data. For the same reason, we are the first to apply any kind of SBM to
macaque, mouse, rat, and *Drosophila* structural connectivity data, as well.

We also make several methodological contributions. For instance, the use of a community morphospace
to study interactions among pairs of communities seems to be a potentially profitable way of studying a
network’s meso-scale structure. Based on the concept of a community motif, we defined a diversity score
that we could map back to individual brain regions. The diversity index, at least in human data, corroborated
some of our hypotheses. Namely, that it should be greatest in poly-functional, association areas (indeed, we
find that control and sub-cortical areas achieve the greatest diversity score). Even more interesting is that
intersubject variability in the regional diversity of precisely these same areas is correlated with performance
on cognitive control tasks. We show that these areas are neither the most highly- nor the most weakly-
connected, suggesting possible functional roles for these “middle class” brain areas. Lastly, we find many of
the same architectural principles in the non-human datasets.

The reviewer points to two important papers linking non-assortative network properties to rich clubs.
While those papers represent important contributions to the field of network neuroscience, they nonetheless
differ from our submission in at least one important way. Whereas our paper focuses on patterns in the
meso-scale structure of neural systems, those two papers focus on properties of individual nodes and edges.
Certainly, both scales matter and the fact that they converge to highlight non-assortative structure in brain
networks is, indeed, comforting. However, non-assortative community structure is not identical to “open”
motifs among nodes. In principle, individual nodes can independently form non-assortative links and edge-
level motifs. At the meso-scale, however, non-assortative interactions among communities indicates collective
and cooperative behavior among *groups* of nodes.

The section **Community and meso-scale connectome analyses** now discusses these issues in greater
detail:

- • “Our study represents one of the first to explore the utility of blockmodels in conjunction with human
and animal structural connectome data (though past studies have investigated blockmodels in the
context of functional connectivity [36, 37, 38, 39, 40]). Furthermore, we demonstrate the potential
benefits of this approach, linking blockmodels to behavior as well as functional connectivity (in the
human) and gene co-expression (in the mouse). Future studies may wish to extend these approaches
to the study of neurodevelopment [41], or the alteration of connectomic structure in psychiatric disease
[42, 43] and neurological disorders [44, 45]”.

**Comment 6**

*I am somewhat (pleasantly) surprised, given the very noisy nature of individual tractography data that I*
*have seen (expect perhaps for carefully reconstructed connectomes from the highly curated human connec-*
*tome project) that individual correlations with performance were discovered. Brief details of acquisition and*
*reconstruction must be provided in the Methods here, since this remains a contentious area. What was the*
*connection density? Also, a very brief summary of these data could be given at the beginning of the Results.*

We apologize for this oversight and now include a more detailed description of the acquisition and recon-
struction procedures for the human connectome data. Additionally, across subjects, the binary connection
density and average node strength were $d = 0.58 \pm 0.04$ and $\langle s \rangle = 85.49 \pm 11.82$, respectively (mean plus/minus
standard deviation across subjects).

In addition to describing these data briefly at the beginning of the **Results** section, we also include a more
comprehensive description of diffusion imaging and tractography in the section **Materials and Methods:**
**Human connectome dataset:**

- • “We fit the weighted stochastic blockmodel (WSBM) to group-representative human connectome data
reconstructed from diffusion spectrum images with state-of-the-art tractography algorithms”.
- • “Diffusion spectrum images (DSI) were acquired for a total of 30 subjects along with a T1-weighted
anatomical scan. We followed a parallel strategy for data acquisition and construction of streamline
adjacency matrices as in previous work [46]. DSI scans sampled 257 directions using a Q5 half-shell
acquisition scheme with a maximum b -value of 5,000, an isotropic voxel size of 2.4 mm, and an axial
acquisition with the following parameters: repetition time (TR) = 5 s, echo time (TE) = 138 ms, 52

slices, field of view (FoV) (231, 231, 125 mm). All procedures were approved in a convened review by the
University of Pennsylvania’s Institutional Review Board, and were carried out in accordance with the
guidelines of the Institutional Review Board/Human Subjects Committee, University of Pennsylvania.
All participants volunteered with informed consent in writing prior to data collection.

DSI data were reconstructed in DSI Studio (www.dsi-studio.labsolver.org) using q -space diffeomorphic
reconstruction (QSDR) [47]. QSDR first reconstructs diffusion-weighted images in native space and
computes the quantitative anisotropy (QA) in each voxel. Then, it warps the images to a template
QA volume in Montreal Neurological Institute (MNI) space using the statistical parametric mapping
(SPM) nonlinear registration algorithm. Once in MNI space, spin density functions were reconstructed
with a mean diffusion distance of 1.25 mm using three fiber orientations per voxel. Fiber tracking was
performed in DSI studio with an angular cutoff of 55° , step size of 1.0 mm, minimum length of 10 mm,
spin density function smoothing of 0.0, maximum length of 400 mm and a QA threshold determined
by DWI signal in the colony-stimulating factor. Deterministic fiber tracking using a modified FACT
algorithm was performed until 1,000,000 streamlines were reconstructed for each individual.

Anatomical scans were segmented using FreeSurfer59 and parcellated using the connectome mapping
toolkit [48]. A parcellation scheme including $n = 129$ regions was registered to the B0 volume from
each subject’s DSI data. The B0 to MNI voxel mapping was used to map region labels from native
space to MNI coordinates. To extend region labels through the grey-white matter interface, the atlas
was dilated by 4 mm [49]. Dilation was accomplished by filling non-labelled voxels with the statistical
mode of their neighbors’ labels. In the event of a tie, one of the modes was selected uniformly at
random. From these data, we constructed a structural connectivity matrix, \mathbf{A} whose element A_{ij}
represented the number of streamlines connecting region i to region j , divided by the sum of volumes
for regions i and j .”

- • “Each individual’s resulting network was undirected (i.e. $A_{ij} = A_{ji}$) with density and mean node
strength of $d = 0.58 \pm 0.04$ and $\langle s \rangle = 85.49 \pm 11.82$), respectively.”

Comment 7

*Section on “Behavioural relevance . . . ”: Why/how were differences in total connection weight partialled out?
Why not normalize the matrices to have uniform weights before the analysis? Were do the differences arise
(e.g. do they correlate with white matter volume)? Also, what was the nature of the FDR correction? How
many tests were performed/corrected for?*

The reviewer raises important details that were not included in our original description. In general, the
reviewer is absolutely correct that differences in coarse, non-specific measures like total connection weight
propagate to local measures. That is, apparent differences or correlations in regional properties of a network
can oftentimes be attributed to less interesting global differences in the network’s density or total weight.
We were interested, specifically, in comparing the diversity index (a regional measure) and subjects’ task
performances. Accordingly, we wished to control for whole-brain measures like total connection weight (row
and column sum of subjects’ connectivity matrices). To do this, we calculated total connection weight for
all subjects and partialled out this variable from the diversity indices of brain regions. We then calculated
the correlation of task performance with the residuals of this regression analysis.

Beyond artifactual sources, we accept that brains are different from one individual to another, and while
we expect subjects to be similar to one another at a coarse scale, we also expect that fine-scale aspects
of their white-matter architecture will differ. These differences could be focal, highly localized effects at
the level of particular tracts, or a brainwide effect in which all (or a majority) of tracts are stronger or
weaker than those of other subjects. Though imperfect, the neurobiological interpretation of white-matter
network architecture is clear, and under ideal settings recapitulates in myelinated fiber tracts the same axonal
projections identified in non-human tract-tracing experiments [50, 51] and microstructural properties noted
in post-mortem studies [52].

Irrespective of the source of individual variation, we would like to note that if we did not correct for
global differences in connectivity, then the spatial pattern of correlations we report in the main text are, in
fact, largely unchanged. In fact, in the absence of a correction, the pattern is nearly identical but the overall

magnitude of correlations is much stronger. We include in this response (as well as the **Supplementary**
**Materials**) a figure indicating precisely this.

This figure is now called out in the main text:

- • “(In the **Supplementary Material** we show that we get similar results without partialing out total
connection weight; Fig. S17.)”

Figure 11: **Effect of total network weight corrections on correlations between diversity scores and task performance.** Each point represents the region-level correlation coefficient and the blue line represents the break-even line. If two brain regions had identical correlations with and without corrections for total network weight then they would fall along this line.

To the reviewer’s final point, we performed an FDR correction for multiple comparisons. Because we
aimed to assess system-level effects, this correction was performed after correlations had been aggregated
and averaged by brain system. Note, that the p -values associated with these correlations were obtained
non-parametrically *via* a permutation test (eight tests in total).

**Comment 8**

*Most of the first paragraph of the section “Connectomes support diverse ...” simply repeats the Intro and*
*could be deleted.*

We thank the reviewer for this suggestion. We have rewritten that paragraph to have less overlap with
introduction.

- • “The human connectome’s ground truth meso-scale structure is unknown. This motivates studying
alternative methods for uncovering communities and characterizing their similarities and differences.
In this section, we compare the results of applying two well-known community detection methods: a
weighted stochastic blockmodel (WSBM) and modularity maximization (Q_{max}).”

**Comment 9**

*I found it impossible to see any differences between Q_{max} and WSBM in Figure 3B.*

We apologize for the lack of visual clarity in this figure. To encapsulate the full range of individual
 community assortativity scores we had to extend the y-axis, which obscured the differences between the
 mean curves. We have included an inset that shows the two curves in the absence of individual points within
 a restricted range. We have included this new figure below (see Fig. 12).

Figure 12: Modularity maximization and the weighted stochastic block-model uncover fundamentally different architectural signatures.

**Comment 10**

*Why are the WSBM networks more assortative than the null distribution in Fig 3C?*

The statistic that we compare to the null distribution is the summed difference between orange and blue
 curves in Fig 3B. The statistic is computed as the difference between the mean assortativity of all size-
 N_r communities detected using the WSBM and Q_{max} summed over all possible values of N_r . Its value is
 negative because we subtract Q_{max} assortativity from WSBM assortativity (note that the observed statistic,
 which is shown in yellow, is large and negative). The null distribution was estimated by randomly permuting
 community assignments, computing mean assortativity for every community of size N_r , and computing the
 summed difference across all N_r . This null model tests whether differences in assortativity can be attributed
 to community size and number.

**Comment 11**

*Suggest delete the interpretive phrase “suggesting the capacity for an equally . . .” from p6 of the Results.*

We have removed the above phrase.

**Comment 12**

*It is the authors own preference, but I found aspects of the Discussion highly speculative for an original
 research article.*

We thank the reviewer for the suggestion and, while we retained the section and its overall spirit, we
 have removed the more far-fetched aspects of the discussion. Specifically, we made the following changes:

- • We removed the phrase “computationally-relevant” in the section **Discussion: Community and**
 **meso-scale connectome analyses.**
- • In the same section, we changed the sentence, “To comprehend the organization of connectome data,
 especially at the cellular scale, requires dimension reduction techniques like community detection that
 can distill the important organizational principles from those that are less useful” so that it now reads

“To comprehend the organization of connectome data, especially at the cellular scale, **may** require
dimension reduction techniques like community detection that can distill the important organizational
principles from those that are less useful.”

- • In the section **Discussion: Connectomes exhibit rich, non-assortative structure**, we changed
the phrase “. . . into a more varied landscape that supports top-down . . .” so that it now reads “. . . into
a more varied landscape that **possibly** supports top-down . . .”

**Comment 13**

*p13: The cite regarding repertoire diversity might also consider [3].*

We had discussed this reference in other parts of the manuscript, but now also include it in the section
**Discussion: Connectomes exhibit rich, non-assortative structure**.

**Comment 14**

*p15: How did the authors go from a whole group consistency-based connectome back to individual subjects?*

We apologize for any confusion. Throughout most of the main text, we analyze a group-representative
matrix that was constructed from 30 subject-level matrices through an averaging procedure. We use this
group matrix to illustrate the basic differences between communities detected using Q_{max} and the WSBM.
For the final section, in which we demonstrate that regional diversity tracks behavior, we no longer analyze
the group-representative matrix, but instead apply the WSBM directly to single-subject matrices.

**Comment 15**

*p16: were the structural and functional connectomic data and the behavioural data all from the same subjects?
Why 30 for structural connectivity and 70 for functional connectivity?*

We tried to locate the reference to 70 subjects, but were unable to do so. In any case, the functional and
structural connectivity data were recorded as part of the same study and included 30 individuals.

**Reviewer references**

References:

- 1. Zalesky, A., Fornito, A., & Bullmore, E. (2012). On the use of correlation as a measure of network
connectivity. *Neuroimage*, 60(4), 2096-2106.
- 2. Harriger L, Van Den Heuvel MP, Sporns O. 2012 Rich club organization of macaque cerebral cortex
and its role in network communication. *PLoS ONE* 7,
- 3. Sporns O, Kotter R (2004) Motifs in brain networks. *PLoS Biology* 2: e369.
- 4. Roberts JA, Perry A, Lord AR, Roberts G, Mitchell PB, Smith RE, Breakspear M (2016) The
contribution of geometry to the human connectome. *Neuroimage* 124: 379-393.

**Reviewer #3**

*I quite enjoyed reading this report, which challenges the classical view of the view of the connectome being
composed of segregated communities and introduces the alternative view on the existence of more heteroge-
neous community configurations.*

*Unlike to the standard methods used to define communities of the structural connectome, the authors
utilize a different technique, weighted stochastic block model (WSBM), one that does not explicitly impose
the assumption of the modularity maximization and hence segregated communities. Using the WSBM, this
work “reveals” other kinds of communities and community interactions, where the newly found communities
show a better overlap with the functional networks of the brain. It is also quite interesting to see that the
intersubject variability in diversity of the community profiles of certain brain regions shows some correlation
with the behavioral performance.*

*I think the paper introduces a novel and quite interesting perspective on the diversity of community*
*organization in the connectome. My major comment is about the lack of true ground truth for the studied*
*problem. As a logical decision, the authors chose to compare to the most commonly used state-of-the art*
*method, modularity maximization, referred as Q_{max} here. However, this algorithm, as also stated by the*
*authors is designed to maximize modularity and hence assortativeness. Hence, removing this particular*
*constraint modularity maximization as in the case of using WSBM, naturally leads to less assortative*
*community structures compared to Q_{max} , which by design extracts segregated communities. My main question*
*is how do we know that the partitions returned by WSBM are more “correct” compared to those detected by*
*Q_{max} ?*

*Having said that, I would acknowledge that this is not a drawback of the method presented here but a*
*general concern about the lack of a true ground truth for the problem at hand. It may be important to at least*
*mention or discuss this point and maybe point out that the results drastically change when the modularity*
*maximization constraint is removed, although a true ground truth for neither algorithm exists.*

We thank the reviewer for their overall positive comments. The reviewer raises an important and timely
question (one that is also repeated by Reviewers 1 & 2): Given two community detection algorithms that
partition a network differently, can we claim that one is more “correct” or “better” than the other? This
is a challenging problem that is being actively researched [53]. One common approach for comparing two
community detection algorithms or sets of partitions is by cross-validation using metadata [31, 30]. In short,
this approach assumes that there exists node-/edge-level metadata that reflect a network’s ground truth
communities better than those estimated from its topology alone. In the present study, we use whole-brain
functional connectivity (human) and gene co-expression patterns (mouse). We can compare communities
detected using the WSBM and Q_{max} by quantifying how well they are aligned to these metadata. If one or
the other community detection method consistently outperforms the other then we can claim that, at least
along these specific dimensions, that method is superior to the other.

The results of our cross-validation procedure, which we document in the section **Functional relevance**
**of the WSBM**, show that the WSBM does, indeed, outperform Q_{max} when we compare their respective
communities to the metadata. Moreover, because we recognize functional connectivity and gene co-expression
as being important to the function of neural systems, we interpret these results as an indication that the
WSBM communities capture functionally relevant patterns of connectivity.

We now include extensive discussion of the cross-validation procedure in the **Results** section:

- • “Here, we test these two hypotheses by cross-validating and comparing WSBM and Q_{max} partitions
using empirical FC as metadata (See **Materials and Methods** for more details on FC reconstruction
from BOLD signals). This approach – cross-validation through metadata – is well-established and has
been used extensively in past studies [31, 30]. In essence, it assumes that metadata better represents
some aspect of a network’s ground truth organization than its structural topology alone. Community
detection methods that are more closely aligned with the metadata may be more sensitive to the
network’s ground truth organization and are considered, in this quantitative and objective sense,
superior to those that do not. We reasoned that if the brain’s correlated activity pattern is better
described by assortative communities behaving autonomously, then the FC network will be more closely
aligned with Q_{max} communities. On the other hand, if the correlation pattern is better described by
interacting, non-assortative communities, the alignment of FC to WSBM communities will be greater.
Here, we quantify this alignment as the mean weight of within-community functional connections minus
the mean weight of between-community connections.”

**Comment 1**

*How does the approach followed by the authors differ from the hierarchical clustering method, cited as ref*
*[35] in the manuscript, as to my knowledge this method also falls outside of the modularity maximization*
*framework. Also, what was the motivation behind the choice of WSBM instead of for instance the hierarchical*
*clustering approach as in [35]?*

The hierarchical method proposed by Clauset et al. shares many properties with the WSBM, most
prominently the use of a statistical model and maximization of a likelihood function to infer communities.
However, it differs in a few important ways. For example, the hierarchical method is not compatible with

weighted networks. It does, however, have the distinct advantage of inferring a hierarchy of communities
rather than a single partition. So the method of Clauset et al. will identify an entire tree of community
partitions. Nonetheless, we decided to use the weighted version of the stochastic blockmodel presented by
[34, 35]. This decision was motivated by the fact that most brain network data are weighted in some way
and because we, as neuroscientists, believe that those weights are of neurobiological relevance.

Comment 2

*Page 2, first paragraph: “Next, we define a node-level diversity index that quantifies the extent to which*
*individual neural elements participate in communities of all classes.” Can a node (neural element) belong to*
*multiple clusters; i.e. the communities can be overlapping and are not disjoint?*

We apologize for any confusion. The WSBM results in a hard partition of network nodes (neural ele-
ments) into one and only one community. Given a single partition, we classified the interactions between
pairs of communities as either assortative, core-periphery, or disassortative. We then counted, for each com-
munity, the number of times it participated in each type of interaction. The diversity index of a community
is quantified as the entropy over that distribution and assigned, uniformly, to all nodes comprising that
community. Thus, communities whose inter-community interactions are of one type, e.g. only assortative,
contain nodes of low diversity. Conversely, if a community’s inter-community interactions are varied, then
its constituent nodes will have high diversity.

We have now added a subsection in **Materials and Methods** further detailing the calculation of the
diversity index.

- • “A partition of a network into communities induces a set of two-community motifs based on connection
densities. In the previous section, we presented rules for classifying those motifs into one of three
classes. For a K -community partition, community r participates in $K - 1$ interactions. We can
calculate for each motif class (now differentiating between cores and peripheries, resulting in four
distinct classes), how frequently it appears among community r ’s $K - 1$ interactions. If we express these
frequencies as probabilities, P_a , P_c , P_p , and P_d (subscripts indicate “assortative”, “core”, “periphery”,
and “disassortative” motif frequencies, respectively), we can then calculate an entropy:

$$H_r = -[P_a \log_2 P_a + P_c \log_2 P_c + P_p \log_2 P_p + P_d \log_2 P_d]. \quad (1)$$

This entropy is zero if community r participates in only one motif class and is maximized when r
participates in all classes equally. We then assign this score to all nodes $i \in r$. The resulting vector of
length $[N \times 1]$ specifies the single-partition diversity index for each node. We can calculate this vector
for all K -community partitions and estimate mean diversity indices for each node by averaging across
partitions.”

Comment 3

*Fig. 3a: Are the within-technique variation of information (VI) scores based on the comparison of partitions*
*with the same number of communities on two different subjects datasets?*

*As the comparisons are performed on partitions with the same number of communities, I assume two*
*different partitions using the same technique can come from the use of different datasets. However, that*
*is not clear at that point of the manuscript, as any information on different subjects data etc. has been*
*provided yet. The authors may want to explain what leads to different partitions with the same number of*
*communities, which are used for comparison.*

We apologize for any confusion. We only calculate VI for pairs of partitions that result in the same
number of communities. However, the comparisons are not carried out at the single-subject level. Because
both Q_{max} and the WSBM algorithms are non-deterministic – i.e. repeated runs of the algorithm usually
result in slightly different solutions – we computed VI between pairs of partitions uncovered using the same
community detection algorithm and also between algorithms. We now clarify this in the manuscript and
figure caption.

In the section **Connectomes support diverse meso-scale architecture** we now include the state-
ment:

- “Specifically, we computed VI separately for three different subsets of partitions: partitions detected using WSBM with other WSBM partitions; partitions detected using Q_{max} with other Q_{max} partitions; partitions detected using the WSBM with Q_{max} partitions.”

Comment 4

Fig. 3A: Also, the authors mention that both techniques, WSBM and Q_{max} lead to self-similar partitions that are statistically different between techniques. In Fig. 3A, the variation of information (VI) scores of WSBM are much higher than those of Q_{max} and for $K=10$, the within-technique VI for WSBM is even higher than between-technique VI. What is the cause of such difference in within-technique VI observed between the two techniques? Are the WSBM partitions reliable, if they show such high within technique VI scores?

We thank the reviewer for pointing this out. The reviewer has correctly interpreted the figure – on average, partition similarity is greater for repeated runs of Q_{max} than for the WSBM and for $K = 10$ the between technique similarity is greater than within-technique similarity of the WSBM. We believe that this may be a peculiarity of the human connectome dataset – we see more comparable levels of similarity when investigating the non-human connectome data. See Mouse and *Drosophila* in Fig. S2.

It is also important to note that, like Q_{max} , the WSBM must infer the community assignments of N nodes. In addition, however, the WSBM must also estimate the parameters for each of $K(K - 1)$ within/between community blocks. This results in a larger parameter space and may lead to more variability from run to run.

Despite this, the communities uncovered using WSBM converge across species to paint a picture of a non-assortative brain, offer superior predictions meta-data (FC and gene co-expression) compared to Q_{max} , and can be used to predict behavioral measures. This highlights the utility and reliability of the WSBM and paves the way for future studies.

Comment 5

The authors mention: “We compared these curves using functional data analysis, which is a set of statistical tools for comparing continuous curves [47,48]. We found that the observed scores were smaller than those obtained under the null model ($p < 10^3$), confirming that WSBM communities tend to be less assortative than Q_{max} (Fig. 3C)”. Arent these curves discrete set of measurements, hence allowing for a comparison for instance by Monte-Carlo approach; simply by shuffling the labels of assortativity scores between two methods over multiple comparisons?

The reviewer is correct. An alternative approach for comparing community assortativity is to proceed point by point (where each point represents community size), and compute a point-wise p -value by randomly permuting community labels. This would result in a series of p -values, which would allow us to independently assess whether communities of a given size differ in their assortativity. However, it would also pose a multiple comparison problem, as separate tests are performed at each value of community size. By contrast, our aim was simply to test whether community assortativity, on average, differed between techniques, which motivates the use of FDA, which is a tool for the statistical comparison of curve shapes that circumvents the multiple comparisons problem by performing tests between the full curves rather than at many points.

Comment 6

I fail to understand Fig. 3C and the stats performed with functional data analysis. What does the y-axis labeled as “Probability” represent? The authors mention “Specifically, we generated a statistic by performing a pointwise subtraction and summation of the curves $A(N)$ obtained for the WSBM and Q_{max} . The value of this statistic quantifies the difference between mean community assortativity across communities of all sizes and is negative when communities detected using Q_{max} are more assortative than WSBMs. We compared this statistic against a null distribution obtained from a null model wherein we preserved the number and size of communities in a given partition but permute nodes assignments uniformly and randomly (1000 repetitions).”

What does the pointwise subtraction and summation of the curves $A(N)$ yield, is it average difference between the two curves? For the null distribution, doesnt the permutation of nodes result in non-continuous clusters, similar to a random assignment? Would that be a fair comparison to use?

Again, we apologize for any confusion. The FDA computes the difference in mean assortativity of
communities detected using the WSBM and Q_{max} . These differences are then summed over all possible
community sizes. This sum is treated as a test statistic. We compare this statistic against a null distribution
generated after permuting nodes' community assignments and recomputing community assortivity scores.
This null model tests whether we would expect the observed test statistic given communities of the same
size and number, but randomly assigned. The "probability" label in the figure represents an estimate of the
probability that we observe a particular test statistic under the null model.

The reviewer's question about non-continuous clusters is an interesting one. In general, the answer is
"yes" – randomly permuting community assignments will oftentimes result in spatially-disjoint communities.
It is also the case that, in practice, most (but not all) community detection algorithms applied to structural
connectome data result in spatially-contiguous communities. However, the origin of this spatial contiguity
has been debated, and it remains unclear whether the spatial contiguity is a consequence of biases in
tractography or a feature of the network [25, 54]. While we agree that it would be potentially interesting
to test a null model that results in comparable spatial distributions of communities, (1) it is not usually
possible to permute community assignments while preserving the same spatial distribution of the observed
communities; (2) performing such a test would mean engaging a contentious literature whose topic is beyond
the scope of the present study.

Lastly, we agree with the reviewer that there are certainly different null models that we could test.
However, in the absence of an explicit hypothesis, the test we used represents a reasonable initial point
of comparison. We now include a more detailed description of these analyses in the section **Results:**
**Connectomes support diverse meso-scale architecture:**

- • "Next, we wished to confirm that the WSBM uncovered non-assortative communities, specifically. To
test this hypothesis, we computed for each community r , its size, N_r , and assortativity score, \mathcal{A}_r ,
which measured its internal density of connections less its maximum density of connections to any
other community (See **Materials and Methods**). We then aggregated all detected communities
and computed the mean assortativity score as a function of community size, $\bar{\mathcal{A}}(N)$ (Fig. 3B). These
procedures were performed separately for the WSBM and Q_{max} . We compared these curves using
functional data analysis, which is a set of statistical tools for comparing continuous curves [55, 56].
Specifically, we computed the summed pointwise difference in both curves, which we treated as a test
statistic. We found that the observed statistic was smaller than those obtained under a permutation-
based null model ($p < 10^{-3}$), confirming that WSBM communities tend to be less assortative than
Q_{max} (Fig. 3C). Again, these findings are consistent across connectome data obtained from all species
(Fig. S3).

Comment 7

*The communities detected by WSBM more closely reflect the functional networks. However, I believe it is*
*important to point out that functional networks emerge from the dynamics and interactions between neural*
*elements that is constrained by the structural connections but not purely determined by them. Hence, although*
*some degree of overlap between structure and function is expected, it is not expected that they will be the*
*same or very similar. The effect of the dynamics would play a crucial role in the emergence of the functional*
*networks.*

The reviewer is exactly correct. The organization of FC depends a great deal on the underlying config-
uration of structural connections, though the extent to which FC comes to resemble SC is also dependent
upon the nature of the network's dynamics – i.e. the evolution operator that propagates each brain regions'
state at time t to a new state at time $t + \Delta t$. The measure used to estimate FC also plays a role; correla-
tion measures are known to induce transitive functional connections. Our approach, in line with the aims
of this paper, was to focus on the role of the brain's underlying structural connectivity in influencing FC.
Though over short time intervals, the mapping of structure to function is less constrained [57], there is a
long-standing expectation that over longer time intervals the correlation pattern of the brain's spontaneous,
resting activity will come to resemble its underlying anatomical structure [58].

However, we also agree with that, as written, our explanation for why we might expect a high density of
functional connections within non-assortative communities is unclear. We take this opportunity to detail our

rationale. Specifically, we hypothesized that brain regions with similar incoming and outgoing connections
receive and deliver similar input and output signals, and should therefore exhibit temporally correlated
activity. The existence of this relation is a long-standing assumption in the network neuroscience community.
In fact, in past studies where empirical estimates of FC could not be easily obtained, a measure called the
“matching index” (which calculates the similarity of regions’ connectivity profiles) has been used as a stand-in
[59].

Though through different mechanisms, both the WSBM and Q_{max} produce communities of brain regions
with similar patterns of connections. However, these methods differ in that communities are defined according
to two vastly different topological principles. Q_{max} assumes that the brain’s meso-scale organization is based
on assortative and segregated sub-systems, while the WSBM allows communities to be both assortative and
non-assortative. These differences in meso-scale structure imply differences in brain function. A strictly
assortative brain is aligned with the hypothesis that the brain is composed of communities operating nearly
autonomously, while a brain composed of some non-assortative communities implies that brain function
arises not from independent communities, but from the interactions between communities.

Here, we test these two hypotheses by cross-validating and comparing WSBM and Q_{max} partitions using
empirical FC as metadata (See **Materials and Methods** for more details on FC reconstruction from
BOLD signals). This approach – cross-validation through metadata – is well-established and has been used
extensively in past studies [31, 30]. In essence, it assumes that metadata better represents some aspect of
a network’s ground truth organization than its structural topology alone. Community detection methods
that are more closely aligned with the metadata may be more sensitive to the network’s ground truth
organization and are considered, in this quantitative and objective sense, superior to those that do not.
We reasoned that if the brain’s correlated activity pattern is better described by assortative communities
behaving autonomously, then the FC network will be more closely aligned with Q_{max} communities. On the
other hand, if the correlation pattern is better described by interacting, non-assortative communities, the
alignment of FC to WSBM communities will be greater. Here, we quantify this alignment as the mean weight
of within-community functional connections minus the mean weight of between-community connections.

In addition to noting the limitations of assuming that FC is shaped by structure alone, we have also
amended the **Results: Functional relevance of the WSBM** to better reflect our assumptions and
hypotheses:

- • “While the results of this section suggest that the WSBM is closely aligned with human FC (and mouse
gene-coexpression; see the **Supplementary Materials**), we report several caveats. First, our analysis
assumes a close relationship of FC with the underlying structure. While structure constrains FC, the
mapping between the two is imperfect and fluctuates over shorter timescales [57] and can vary when
different measures of FC are used. The use of a Pearson correlation, for example, induces transitive
functional connections by placing statistical bounds on correlations among triplets of nodes [32]. This
implies that the correlation values are not independent, which may influence our estimates of mean
within- and between-community FC magnitude.”
- • “It is generally agreed upon that brain structural connectivity determines the partners that any given
region can “talk to”, and therefore constrains communication patterns among brain regions, shaping
the correlation pattern of ongoing neural activity, i.e. functional network organization. We reasoned
that if two brain regions receive input from the same set of brain regions and deliver output to the
same set of regions, then their activity over time should be correlated, i.e. those regions would appear
functionally connected to one another. This set of assumptions has a long tradition in the network
neuroscience community. In the past when empirical estimates of FC could not be easily obtained,
measures of similarity between brain regions’ connectivity profiles (e.g., matching index) have been
used as a stand-in [27, 28, 29].

Though through different mechanisms, both the WSBM and Q_{max} produce communities of brain
regions with similar patterns of connections. However, these methods differ in that communities
are defined according to two vastly different topological principles. Q_{max} assumes that the brain’s
meso-scale organization is based on assortative and segregated sub-systems, while the WSBM allows
communities to be both assortative and non-assortative. These differences in meso-scale structure
imply differences in brain function. A strictly assortative brain is aligned with the hypothesis that

the brain is composed of communities operating nearly autonomously, while a brain composed of some
non-assortative communities implies that brain function arises not from independent communities, but
from the interactions between communities.

Here, we test these two hypotheses by cross-validating and comparing WSBM and Q_{max} partitions
using empirical FC as metadata (See **Materials and Methods** for more details on FC reconstruction
from BOLD signals). This approach – cross-validation through metadata – is well-established and has
been used extensively in past studies [31, 30]. In essence, it assumes that metadata better represents
some aspect of a network’s ground truth organization than its structural topology alone. Community
detection methods that are more closely aligned with the metadata may be more sensitive to the
network’s ground truth organization and are considered, in this quantitative and objective sense,
superior to those that do not. We reasoned that if the brain’s correlated activity pattern is better
described by assortative communities behaving autonomously, then the FC network will be more closely
aligned with Q_{max} communities. On the other hand, if the correlation pattern is better described by
interacting, non-assortative communities, the alignment of FC to WSBM communities will be greater.
Here, we quantify this alignment as the mean weight of within-community functional connections minus
the mean weight of between-community functional connections.”

- • “We note that the use of Pearson correlation as a measure of FC results in increased transitivity (if
a strong correlation exists between nodes A and B as well as B and C, then A and C will tend to be
strongly correlated), which can reinforce block structure in correlation matrices [32].”

Comment 8

*The authors state “To test whether this was the case, we imposed partitions obtained from the WSBM and*
*Q_{max} applied to the structural connectome onto the FC matrix and computed the difference of within- and*
*between-community FC density. We found that over a range $K = 2, . . . , 10$, the WSBM consistently*
*uncovered communities whose internal FC density exceeded their between-community density (Fig. 5A).” The*
*functional networks are defined on the FC matrix, hence I would have thought that they would superimpose*
*FC matrix parcellation onto the structural connectome (SC) partitions. For instance Fig. 5A caption states:*
*“Functional connectivity (FC) matrix ordered by functional system”. As both, FC connectivity and the*
*labeling of the functional systems come from the functional connectivity, how does this figure capture the*
*partitions of the structural connectome? Is it a misunderstanding on my side or is there a confusion between*
*SC and FC in the wording here?*

The reviewer is correct. The FC matrix depicted in Fig. 5A is ordered according to functional systems
so that the reader can develop some intuition for the matrix’s structure. We chose not to order the matrix
by partitions detected by either WSBM or Q_{max} because there were thousands of such partitions and
choosing a representative partition from among those was not trivial. We note that, in general, discerning
the differences between the two techniques based on a visual comparison is not especially illuminating; it
was only by performing detailed statistical comparisons that we were able to confirm that the WSBM better
segregates FC compared to Q_{max} .

In any case, we show here, an example of the FC matrix with its nodes ordered according to consensus
communities and with $k = 5$ for both the WSBM and Q_{max} (Fig. 13). We obtained the consensus commu-
nities by reclustering an association matrix, which we constructed separately for partitions detected using
either method.

Because we believe that this figure does not contribute much beyond what we already mention in the
main text, we opted to not include it in the manuscript. We have, however, edited the caption for Figure
5A to make clear that the ordering of nodes represents the functional systems described in [60].

- • “Note that the order of nodes shown in this panel does not correspond to partitions generated by either
the WSBM or Q_{max} .”

Comment 9

*I think the correlations between the diversity index and performance categorized according to functional*
*networks is very interesting. From what I can see in Fig. 8B, one can conclude that some networks require*

Figure 13: **FC matrix ordered by WSBM and Q_{max} partitions.** Rows and columns of the FC matrix are reordered so that nodes assigned to the same community are next to one another.

*certain type of motifs (interactions) such as the visual network and the DAN, whereas others such as the*
 *control network, subcortical areas and maybe also the default mode network contain interactions of various*
 *kinds for a good cognitive performance. What about the whole brain diversity of connections? Would that*
 *make any inference on the cognitive performance?*

We agree that it would be interesting to test whether diversity as a global statistic was related to subject
 performance. To do so, we calculated the Pearson and Spearman correlation of performance on the Stroop
 and Navon tasks (we tested their performance separately and also their average performance on both tasks)
 with a node-averaged measure of diversity. The greatest magnitude correlation we observed was using the
 Spearman measure to relate total accuracy on the Stroop task with average diversity ($\rho = 0.18$; $p > 0.05$).
 All other correlations were weaker and also not significant.

We note that while global diversity does not appear to track Stroop or Navon task performance, it may
 be useful for future studies. Accordingly, we now note this in the main text in **Materials and Methods:**
 **Diversity index:**

- • “Note that while we define the diversity index at the level of individual brain regions (network nodes), it
 would be straightforward to average node-level diversity scores to compute a global diversity score that
 could serve to characterize the diversity of meso-scale structure in the network as whole. Alternatively,
 a global diversity index could be computed straightforwardly as an entropy based on the complete set
 of community motif frequencies.”

**Comment 10**

*Page1: What is the difference between clusters and communities? I found the illustration of different con-*
 *nectivity profiles in Fig. 1 very useful. It may be very helpful to illustrate the concepts of region, community,*
 *partition in a similar manner for the nave reader, if possible, at least as supplementary material.*

We apologize for any confusion. In many applications, the terms cluster, community, module, and group
 (among others) have come to mean the same thing. So when we refer to a “cluster” or “community” we are
 referring to a set of brain regions grouped together according to some topological principle, e.g. by maximizing
 Q or using the WSBM. We have added a figure in the supplement illustrating the different topological scales
 of a network (node \rightarrow community \rightarrow whole network).

**Comment 11**

*Page 1: what do individual network nodes represent? Brain regions? It would be useful to specify here.*

The reviewer is correct – in all five connectome datasets, nodes represent brain areas whose boundaries
 are delineated based on their function, morphology, cyto-architecture, or related measures. We note this in

the main text in the section **Human connectome dataset** and in the **Supplementary Material** under
the section **Non-human connectome data**.

**Comment 12**

*Is assortative architecture the same as small-world, as used in some reports on connectomes architecture?*

Assortative architecture and small-worldness are distinct concepts. Small-worldness refers to a global
property of a network in which it simultaneously exhibits high levels of clustering (nodes' neighbors tend
to be connected to one another) and short path length (the mean number of steps between nodes is small).
Assortative architecture refers to a property of small groups or communities of nodes (so it is not a global
property of a network) in which nodes that belong to a group prefer to connect to other nodes in the same
group compared to nodes in different groups. While it is possible for a network with assortative structure to
also possess small-world qualities (dense communities with a few links between communities) and *vice versa*,
in general that is not the case.

**Comment 13**

*Fig. 2: Community labeled with purple seems to consist of only one small brain region in the right hemi-*
*sphere, which does not seem to have any correspondence in the left hemisphere, whether labeled as the same*
*community or not. Where does this asymmetry stem from? Is it an algorithmic artifact?*

We appreciate the reviewer's attention to detail. In this case, the purple community and its relatively
small size is a result of the stochasticity of the WSBM algorithm. That is, in attempting to optimize their
respective objective functions, the output of both the WSBM and Q_{max} will vary somewhat. The commu-
nities shown in Figure 2 represent the outputs of single runs of the algorithm and should not be treated as
necessarily representative of the network's ground truth communities. It is not difficult to identify a different
partition of the same network into the same number of communities with comparably-sized communities.
To demonstrate this, we show an alternative partition of the network into five communities. Specifically, we
chose the partition with the most similarly-sized communities (25.6 ± 2.07 nodes per community). We have
also replaced the WSBM partition in Figure 2 with the communities shown here.

Figure 14: **Surface plot showing similar-sized communities detected using the WSBM.** Colors represent different community labels.

It is essential to note that in the main text we intentionally avoid defining a single “representative”
 partition from among the ensemble of detected partitions. In general, most networks have “fuzzy” meso-
 scale structure, with a near-degeneracy of optimal partitions. That is, there may be many partitions judged
 to be of similar quality but which possibly differ a bit from one another. In the main text, we embrace this
 variability and focus on the statistical properties of this ensemble of near-optimal partitions.

**Comment 14**

*Fig. 3. Caption: Q_{max} should be Q_{max} in latex notation, “perserved” should be “preserved”.*
 We thank the reviewer for pointing this out. We have corrected both typos.

**Comment 15**

*Fig. 3F, what do the upper and lower limits of the box plot represent?*
 The upper and lower limits of each box represent the 25th and 75th percentiles of each system’s assorta-
 tivity.

**Comment 16**

*Fig. 4C, how were the rich club nodes labeled/estimated?*
 To identify putative rich clubs, we maximized a weighted rich club coefficient [61]. This coefficient
 is calculated at different levels, k , corresponding to nodes’ degrees. For a given, k , we first identify all
 nodes of degree k or greater, the number of connections among those nodes ($E_{>k}$), and the total weight of
 those connections ($W_{>k}$). We divide $W_{>k}$ by the total weight of the strongest $E_{>k}$ edges in the network,
 $W_{max} = \sum_{l=1}^{E_{>k}} w_l^{rank}$, where w_l^{rank} is the set of all network edge weights ordered from strongest to weakest.
 This measure defines the weighted rich club coefficient:

$$\phi^w(k) = \frac{W_{>k}}{W_{max}} . \quad (2)$$

This coefficient measures, for every possible node degree, k , the total weight of connections among nodes
 whose degrees are greater than k divided by the maximum possible value of the same number of connections.
 We compared $\phi^w(k)$ for the observed network against an ensemble of 100 randomized networks with the same
 degree sequence as the observed network. For every possible k , we calculated the fraction of all randomized
 networks whose rich club coefficient was in excess of the observed network’s. This fraction served as a p -value
 for associated statistical tests and made it possible to identify statistically significant rich clubs ($p < 0.05$).

In practice, this procedure often leads to a range of k over which rich clubs are considered statistically
 significant. Rather than explore this entire range, we focused on a 20-80 split of network nodes assigned to
 and not assigned to the rich club. We justify this split on the grounds that (i) all networks we observed
 exhibited a statistically significant rich club in this range, making it unnecessary to develop separate criteria
 for studying rich clubs across species, and (ii) a rich club composed of 20% of a network’s nodes is exclusive
 enough to be of interest but not so large as to be trivial.

We now include a more detailed explanation of these procedures in our manuscript in the **Materials**
 **and Methods** section.

- • “We identified putative rich club nodes by maximizing a weighted rich club coefficient, $\phi^w(k)$, where
 k is node degree [61]. Intuitively, a weighted rich club is composed of highly connected nodes linked to
 one another by connections with strong weights. To calculate $\phi^w(k)$, we first identify the sub-network
 composed only of nodes whose degree is k or greater, the number of connections among those nodes,
 $E_{>k}$, and the total weight of those connections $W_{>k}$. We also calculate $W_{k>}^{max} = \sum_{l=1}^{E_{>k}} w_l^{rank}$, which
 measures the maximum possible value that $E_{>k}$ connections could have given the edge weights present
 in the network.

$$\phi^w(k) = \frac{W_{>k}}{W_{k>}^{max}} . \quad (3)$$

We compared $\phi^w(k)$ for the observed network against the same measure made over an ensemble of 100 randomized networks with the same degree sequence. For every possible k , we calculated the fraction of all randomized networks whose rich club coefficient was in excess of the observed network's. This fraction served as a p -value for performing statistical tests and made it possible to identify statistically significant rich clubs ($p < 0.05$).

This procedure results in a range of k over which rich clubs are considered statistically significant. Rather than characterize this entire range, we focused on a 20-80 split of network nodes into rich and non-rich groups. We justify this split on the grounds that (i) all of the networks we studied exhibited a statistically significant rich club in this range, making it unnecessary to develop separate criteria for studying rich clubs across species, and (ii) a rich club composed of 20% of a network's nodes is exclusive enough to be of interest but not so large as to be trivial (Fig. 15)."

We also include the following figure:

Figure 15: **Rich club analysis.** (A) Here we display p -values for rich clubs across all five species. (B) The number of statistically significant rich clubs as a fraction of network size.

Comment 17

What is the difference between a core community and a hub?

Though the definition of a “hub” region is not settled upon, it generally refers to a node with a high level of connectivity (high-degree and/or high-strength) that occupies a position of centrality and influence in the

network. A core community refers to a group or community of nodes, all with similar connectivity profiles,
that interacts with a peripheral community. Specifically, the core nodes connect both to one another and
also to the periphery, while the peripheral nodes do not connect to one another but do connect to the core.
Cores, like hubs, represent structures in the network that are associated with influence and centrality. Unlike
hubs, however, cores explicitly refer to groups of nodes, rather than to any particular node.

**Comment 18**

*Page 13, first paragraph: I would say “functional connectivity” (FC) instead of “functional dynamics”, as the*
*comparison was done to FC and with the emergence of new methods such as dynamic functional connectivity*
*etc., “functional dynamics” is now understood as changing functional connectivity.*

We have made this change.

**Comment 19**

*What was the motivation behind using a 128 parcellation and can the authors comment on if/how the change*
*of parcellation may change the observed effects?*

The 128-node parcellation is a sub-division of the well-known Desikan-Killiany atlas [62]. This particular
sub-division is implemented in the Connectome Mapper Toolkit [48]. Though this software includes both
coarser and finer sub-divisions, the division into 129 nodes (128 after excluding brainstem) is particularly
appealing, as cortical and sub-cortical regions have approximately the same volume, which reduces potential
volume-related biases in tractography and network reconstruction.

As the reviewer correctly notes, choice of parcellations can induce biases in the structure of the network.
Because there is considerable debate about what parcellation is the best (especially when used to define
the nodes of a structural connectivity network), dealing with this issue is non-trivial. One strategy to deal
with this issue is to demonstrate that one’s results are robust to reasonable variation of parcellation scheme.
In our case, the nested sub-divisions of the Desikan-Killiany are beneficial, because we can test whether
results obtained using any particular sub-division generalize to the next-coarser and next-finer parcellations.
Here, we show that our main results remain qualitatively the same when we change the number of nodes
from 128 to 82 and to 233. Specifically, we find across different numbers of nodes, the mean system-level
diversity scores are correlated with scores obtained from the 128-node network described in the main text
(Fig. 16B,D). We note that these correlations are intended to be qualitative demonstrations of the robustness
of our results. With only eight systems (corresponding to eight observations), neither correlation passes a
$p < 0.05$ threshold ($p = 0.09$ and $p = 0.43$ for the 82- and 233-node networks). Nonetheless, these findings
suggest a broad correspondence across scales.

We also repeated the behavioral analysis and calculated the correlation of regional diversity scores with
the Stroop and Navon task accuracy. Comparing the 82-, 128-, and 233-node results was complicated by the
fact that network nodes were defined differently in each case. To facilitate comparison across the differently
sized networks we focused on system-level statistics [60]. This entailed aggregating all nodes assigned to the
same system and averaging their diversity-by-behavior correlations to obtain a system-level mean. Because
the number and identities of systems were consistent across the different-sized networks, this enabled us to
relate the system-level scores between networks. We obtained mean system-level scores as we varied the
number of communities from $k = 2$ to $k = 10$, aggregated all system scores and computed two correlations.
First we computed the correlation of system-level scores for the 82- and 128-node networks ($r = 0.32$,
$p < 0.01$). We then computed a similar correlation using system-level scores obtained for the 128- and
233-node networks ($r = 0.32$, $p < 0.01$) (Fig. 17A, B). As with the previous section, the comparison between
scales was not perfect, but confirmed similar overall patterns, suggesting that our results were robust to
reasonable variation in choice of parcellation.

**Comment 20**

*Page 18: In section “Community and regional assortativity” the authors provide the equations for both*
*directed and undirected graphs. Is that done so for the generalization of the provided methods for directed*
*graphs? As far as I understand the results in the actual manuscript are based on undirected graphs. I believe*

Figure 16: **Comparison of system-level diversity scores with 82- and 233-node networks.** (A) System-level diversity scores for 82-node network. (B) Rank correlation of system-level scores obtained for the 82-node and 128-node networks. Panels (C) and (D) are the same as (A) and (B) but for the 233-node network.

*the directed graph use may be necessary for the mouse data? If so, it may be worth mentioning this in the*
 *manuscript.*

With the exception of the human connectome data, all networks we analyze are directed. In general,
 our measures generalize to directed networks. In the section **Materials and methods: Community and**
 **regional assortativity**, we describe how we deal with directed networks. In short, we have the option of
 considering for a node or community the density of its incoming or outgoing connections to other communities.
 Our solution was to take the maximum of the two density measurements as a sort of “worst-case” scenario.
 That is, we consider a community disassortative if *either* its incoming or outgoing connections would lead
 to such a classification.

**Comment 21**

*Eq. (8): it could be easier for the reader if a different notation instead of double indexing was used to refer*
 *a_i and a_{iz_i} . It is not clear to me what a_{iz_i} represents.*

We agree with the reviewer that this notation is confusing. The variable a_{i,z_i} represents the density of
 node i 's connections to its own community, z_i . Similarly, $a_{i,r}$ represents the density of node i 's connections
 to community r . We leave these definitions intact but have changed the regional assortativity variable name
 from a_i to η_i .

**Comment 22**

*It may also be useful to express the diversity index mathematically.*

We have now added a subsection in **Materials and Methods** detailing the calculation of the diversity
 index.

Figure 17: **Comparison of system-level correlations of diversity and task performance with 82- and 233-node networks.** (A) Comparison of system-level correlations between the 82- and 128-node networks for all $K = 2$ to $K = 10$. (B) Same as panel (A), but for the 233-node network.

- “A partition of a network into communities induces a set of two-community motifs based on connection densities. In the previous section we presented rules for classifying those motifs into one of three classes. For a K -community partition, community r participates in $K - 1$ interactions. We can calculate for each motif class (now differentiating between cores and peripheries, resulting in four distinct classes), how frequently it appears among community r ’s $K - 1$ interactions. If we express these frequencies as probabilities, P_a , P_c , P_p , and P_d (subscripts indicate “assortative”, “core”, “periphery”, and “disassortative” motif frequencies, respectively), we can then calculate an entropy:

$$H_r = -[P_a \log_2 P_a + P_c \log_2 P_c + P_p \log_2 P_p + P_d \log_2 P_d]. \quad (4)$$

This entropy is zero if community r participates in only one motif class and is maximized when r participates in all classes equally. We then assign this score to all nodes $i \in r$. The resulting vector of length $[N \times 1]$ specifies the single-partition diversity index for each node. We can calculate this vector for all K -community partitions and estimate mean diversity indices for each node by averaging across partitions.”

Comment 23

*The macaque connectome results seem to show the opposite trade in terms of being assigned to maximally*
*assortative set as rich club and non-rich club members (Fig. S4O). Do the authors have any speculative idea*
*on what may be the reason for this opposite trade?*

The macaque connectome is peculiar in several ways, all of which could lead to atypical results. First,
it is the smallest network we study. Consequently, the network’s global structure can be disproportionately
influenced by the behavior of one or two nodes. Second, whereas the other networks are either whole-brain
or whole-hemisphere, the macaque network is incomplete; connectivity information is available for 29 of 93
total nodes. This means that macaque network properties will likely change as more data becomes available.

References

- [1] Harrison C White, Scott A Boorman, and Ronald L Breiger. Social structure from multiple networks.
i. blockmodels of roles and positions. *American journal of sociology*, 81(4):730–780, 1976.

- [2] Paul W Holland, Kathryn Blackmond Laskey, and Samuel Leinhardt. Stochastic blockmodels: First
steps. *Social networks*, 5(2):109–137, 1983.
- [3] Mark EJ Newman. Communities, modules and large-scale structure in networks. *Nature Physics*,
8(1):25–31, 2012.
- [4] Mark EJ Newman and Michelle Girvan. Finding and evaluating community structure in networks.
*Physical review E*, 69(2):026113, 2004.
- [5] Martin Rosvall and Carl T Bergstrom. Maps of random walks on complex networks reveal community
structure. *Proceedings of the National Academy of Sciences*, 105(4):1118–1123, 2008.
- [6] Feng Luo, Bo Li, Xiu-Feng Wan, and Richard H Scheuermann. Core and periphery structures in protein
interaction networks. *Bmc Bioinformatics*, 10(4):S8, 2009.
- [7] Serguei Saavedra, Felix Reed-Tsochas, and Brian Uzzi. A simple model of bipartite cooperation for
ecological and organizational networks. *Nature*, 457(7228):463–466, 2009.
- [8] Patrick Doreian. Structural equivalence in a psychology journal network. *Journal of the American
Society for Information Science*, 36(6):411–417, 1985.
- [9] Dragana M Pavlovic, Petra E Vértés, Edward T Bullmore, William R Schafer, and Thomas E Nichols.
Stochastic blockmodeling of the modules and core of the caenorhabditis elegans connectome. *PloS one*,
9(7):e97584, 2014.
- [10] César A Hidalgo and Ricardo Hausmann. The building blocks of economic complexity. *proceedings of
the national academy of sciences*, 106(26):10570–10575, 2009.
- [11] Kwang-Il Goh, Michael E Cusick, David Valle, Barton Childs, Marc Vidal, and Albert-László Barabási.
The human disease network. *Proceedings of the National Academy of Sciences*, 104(21):8685–8690, 2007.
- [12] Phuong Anh T Nguyen, Willisa Liou, David H Hall, and Michel R Leroux. Ciliopathy proteins establish
a bipartite signaling compartment in a c. elegans thermosensory neuron. *J Cell Sci*, 127(24):5317–5330,
2014.
- [13] Esteban Real, Hiroki Asari, Tim Gollisch, and Markus Meister. Neural circuit inference from function
to structure. *Current Biology*, 2017.
- [14] Kevin L Briggman, Moritz Helmstaedter, and Winfried Denk. Wiring specificity in the direction-
selectivity circuit of the retina. *Nature*, 471(7337):183–188, 2011.
- [15] Huayu Ding, Robert G Smith, Alon Polog-Polsky, Jeffrey S Diamond, and Kevin L Briggman. Species-
specific wiring for direction selectivity in the mammalian retina. *Nature*, 2016.
- [16] Lav R Varshney, Beth L Chen, Eric Paniagua, David H Hall, and Dmitri B Chklovskii. Structural
properties of the caenorhabditis elegans neuronal network. *PLoS Comput Biol*, 7(2):e1001066, 2011.
- [17] Olaf Sporns and Richard F Betzel. Modular brain networks. *Annual review of psychology*, 67:613–640,
2016.
- [18] Roger Guimera and Luis A Nunes Amaral. Functional cartography of complex metabolic networks.
*Nature*, 433(7028):895–900, 2005.
- [19] Gustavo Deco, Giulio Tononi, Melanie Boly, and Morten L Kringelbach. Rethinking segregation and
integration: contributions of whole-brain modelling. *Nature Reviews Neuroscience*, 16(7):430–439, 2015.
- [20] David C Van Essen, Charles H Anderson, and Daniel J Felleman. Information processing in the primate
visual system: an integrated systems perspective. *Science*, 255(5043):419, 1992.
- [21] H Sebastian Seung. Reading the book of memory: sparse sampling versus dense mapping of connectomes.
*Neuron*, 62(1):17–29, 2009.

- [22] H Sebastian Seung and Uygur Sümbül. Neuronal cell types and connectivity: lessons from the retina.
*Neuron*, 83(6):1262–1272, 2014.
- [23] Hamid R. Noori, Judith Schottler, Maria Ercsey-Ravasz, Alejandro Cosa-Linan, Melinda Varga, and
Zoltan Toroczkai. A multiscale cerebral neurochemical connectome of the rat brain. *PLOS Biology*,
page <https://doi.org/10.1371/journal.pbio.2002612>, 2017.
- [24] Szabolcs Horvát, Răzvan Gămănu, Mária Ercsey-Ravasz, Loïc Magrou, Bianca Gămănu, David C
Van Essen, Andreas Burkhalter, Kenneth Knoblauch, Zoltán Toroczkai, and Henry Kennedy. Spa-
tial embedding and wiring cost constrain the functional layout of the cortical network of rodents and
primates. *PLoS biology*, 14(7):e1002512, 2016.
- [25] Mária Ercsey-Ravasz, Nikola T Markov, Camille Lamy, David C Van Essen, Kenneth Knoblauch, Zoltán
Toroczkai, and Henry Kennedy. A predictive network model of cerebral cortical connectivity based on
a distance rule. *Neuron*, 80(1):184–197, 2013.
- [26] James A Roberts, Alistair Perry, Anton R Lord, Gloria Roberts, Philip B Mitchell, Robert E Smith,
Fernando Calamante, and Michael Breakspear. The contribution of geometry to the human connectome.
*Neuroimage*, 124:379–393, 2016.
- [27] Richard E Passingham, Klaas E Stephan, and Rolf Kötter. The anatomical basis of functional localiza-
tion in the cortex. *Nature Reviews Neuroscience*, 3(8):606–616, 2002.
- [28] Claus C Hilgetag, Rolf Kötter, Klaas E Stephan, and Olaf Sporns. Computational methods for the
analysis of brain connectivity. In *Computational neuroanatomy*, pages 295–335. Springer, 2002.
- [29] Giorgio A Ascoli. *Computational neuroanatomy: Principles and methods*. Springer Science & Business
Media, 2002.
- [30] Darko Hric, Richard K Darst, and Santo Fortunato. Community detection in networks: Structural
communities versus ground truth. *Physical Review E*, 90(6):062805, 2014.
- [31] Jaewon Yang, Julian McAuley, and Jure Leskovec. Community detection in networks with node at-
tributes. In *Data Mining (ICDM), 2013 IEEE 13th international conference on*, pages 1151–1156. IEEE,
2013.
- [32] Andrew Zalesky, Alex Fornito, and Ed Bullmore. On the use of correlation as a measure of network
connectivity. *Neuroimage*, 60(4):2096–2106, 2012.
- [33] Sergei Maslov and Kim Sneppen. Specificity and stability in topology of protein networks. *Science*,
296(5569):910–913, 2002.
- [34] Christopher Aicher, Abigail Z Jacobs, and Aaron Clauset. Adapting the stochastic block model to
edge-weighted networks. *arXiv preprint arXiv:1305.5782*, 2013.
- [35] Christopher Aicher, Abigail Z Jacobs, and Aaron Clauset. Learning latent block structure in weighted
networks. *Journal of Complex Networks*, 3(2):221–248, 2015.
- [36] Sean L Simpson, Satoru Hayasaka, and Paul J Laurienti. Exponential random graph modeling for
complex brain networks. *PloS one*, 6(5):e20039, 2011.
- [37] Sean L Simpson, Malaak N Moussa, and Paul J Laurienti. An exponential random graph model-
ing approach to creating group-based representative whole-brain connectivity networks. *Neuroimage*,
60(2):1117–1126, 2012.
- [38] Daniel Moyer¹², Boris Gutman, Gautam Prasad, Greg Ver Steeg, and Paul Thompson. Mixed mem-
bership stochastic blockmodels for the human connectome. *neuro-degenerative diseases*, 5:6, 2015.

- [39] Andrew C Murphy, Shi Gu, Ankit N Khambhati, Nicholas F Wymbs, Scott T Grafton, Theodore D
Satterthwaite, and Danielle S Bassett. Explicitly linking regional activation and function connec-
tivity: Community structure of weighted networks with continuous annotation. *arXiv preprint*
*arXiv:1611.07962*, 2016.
- [40] Christopher Baldassano, Diane M Beck, and Li Fei-Fei. Parcellating connectivity in spatial maps. *PeerJ*,
3:e784, 2015.
- [41] A Di Martino, D A Fair, C Kelly, T D Satterthwaite, F X Castellanos, M E Thomason, R C Crad-
dock, B Luna, B L Leventhal, X N Zuo, and M P Milham. Unraveling the miswired connectome: a
developmental perspective. *Neuron*, 83(6):1335–1353, 2014.
- [42] M Rubinov and E Bullmore. Fledgling pathoconnectomics of psychiatric disorders. *Trends Cogn Sci*,
17(12):641–647, 2013.
- [43] A Fornito, A Zalesky, C Pantelis, and E T Bullmore. Schizophrenia, neuroimaging and connectomics.
*Neuroimage*, 62(4):2296–2314, 2012.
- [44] D S Bassett and E T Bullmore. Human brain networks in health and disease. *Curr Opin Neurol*,
22(4):340–347, 2009.
- [45] C J Stam. Modern network science of neurological disorders. *Nat Rev Neurosci*, 15(10):683–695, 2014.
- [46] Shi Gu, Fabio Pasqualetti, Matthew Cieslak, Qawi K Telesford, B Yu Alfred, Ari E Kahn, John D
Medaglia, Jean M Vettel, Michael B Miller, Scott T Grafton, et al. Controllability of structural brain
networks. *Nature communications*, 6, 2015.
- [47] Fang-Cheng Yeh, Van Jay Wedeen, and Wen-Yih Isaac Tseng. Estimation of fiber orientation and spin
density distribution by diffusion deconvolution. *Neuroimage*, 55(3):1054–1062, 2011.
- [48] Leila Cammoun, Xavier Gigandet, Djalel Meskaldji, Jean Philippe Thiran, Olaf Sporns, Kim Q Do,
Philippe Maeder, Reto Meuli, and Patric Hagmann. Mapping the human connectome at multiple scales
with diffusion spectrum mri. *Journal of neuroscience methods*, 203(2):386–397, 2012.
- [49] M Cieslak and ST Grafton. Local termination pattern analysis: a tool for comparing white matter
morphology. *Brain imaging and behavior*, 8(2):292–299, 2014.
- [50] Evan Calabrese, Alexandra Badea, Gary Cofer, Yi Qi, and G Allan Johnson. A diffusion mri trac-
tography connectome of the mouse brain and comparison with neuronal tracer data. *Cerebral Cortex*,
25(11):4628–4637, 2015.
- [51] Chad J Donahue, Stamatios N Sotiropoulos, Saad Jbabdi, Moises Hernandez-Fernandez, Timothy E
Behrens, Tim B Dyrby, Timothy Coalson, Henry Kennedy, Kenneth Knoblauch, David C Van Essen,
et al. Using diffusion tractography to predict cortical connection strength and distance: a quantitative
comparison with tracers in the monkey. *Journal of Neuroscience*, 36(25):6758–6770, 2016.
- [52] Karla L Miller, Charlotte J Stagg, Gwenaëlle Douaud, Saad Jbabdi, Stephen M Smith, Timothy EJ
Behrens, Mark Jenkinson, Steven A Chance, Margaret M Esiri, Natalie L Voets, et al. Diffusion imaging
of whole, post-mortem human brains on a clinical mri scanner. *Neuroimage*, 57(1):167–181, 2011.
- [53] Leto Peel, Daniel B Larremore, and Aaron Clauset. The ground truth about metadata and community
detection in networks. *Science Advances*, 3(5):e1602548, 2017.
- [54] Richard F Betzel, John D Medaglia, Lia Papadopoulos, Graham L Baum, Ruben Gur, Raquel Gur,
David Roalf, Theodore D Satterthwaite, and Danielle S Bassett. The modular organization of human
anatomical brain networks: Accounting for the cost of wiring. *Network Neuroscience*, 2017.
- [55] James O Ramsay and Bernard W Silverman. *Applied functional data analysis: methods and case studies*,
volume 77. Citeseer, 2002.

- [56] James O Ramsay. *Functional data analysis*. Wiley Online Library, 2006.
- [57] Christopher J Honey, Rolf Kötter, Michael Breakspear, and Olaf Sporns. Network structure of cerebral
cortex shapes functional connectivity on multiple time scales. *Proceedings of the National Academy of*
*Sciences*, 104(24):10240–10245, 2007.
- [58] Gustavo Deco, Viktor K Jirsa, and Anthony R McIntosh. Emerging concepts for the dynamical orga-
nization of resting-state activity in the brain. *Nature Reviews Neuroscience*, 12(1):43–56, 2011.
- [59] Gorka Zamora-López, Changsong Zhou, and Jürgen Kurths. Cortical hubs form a module for mul-
tisensory integration on top of the hierarchy of cortical networks. *Frontiers in neuroinformatics*, 4,
2010.
- [60] BT Thomas Yeo, Fenna M Krienen, Jorge Sepulcre, Mert R Sabuncu, Danial Lashkari, Marisa
Hollinshead, Joshua L Roffman, Jordan W Smoller, Lilla Zöllei, Jonathan R Polimeni, et al. The
organization of the human cerebral cortex estimated by intrinsic functional connectivity. *Journal of*
*neurophysiology*, 106(3):1125–1165, 2011.
- [61] Tore Opsahl, Vittoria Colizza, Pietro Panzarasa, and Jose J Ramasco. Prominence and control: the
weighted rich-club effect. *Physical review letters*, 101(16):168702, 2008.
- [62] Rahul S Desikan, Florent Ségonne, Bruce Fischl, Brian T Quinn, Bradford C Dickerson, Deborah
Blacker, Randy L Buckner, Anders M Dale, R Paul Maguire, Bradley T Hyman, et al. An automated
labeling system for subdividing the human cerebral cortex on mri scans into gyral based regions of
interest. *Neuroimage*, 31(3):968–980, 2006.

Reviewers' comments:

Reviewer #1 (Remarks to the Author):

The authors responded to the reviewer comments thoroughly and, as a result, the manuscript is much stronger. I have a few minor comments, but no remaining substantial concerns.

Your reference to supplementary figures in the main text needs to be updated to reflect additional figures.

In the Results section "Connectomes support diverse meso-scale architecture", I believe that the asterisks in Figure 3A represent 1-tailed t-tests for each K relating $VI(Q_{max}\text{-WSBM})$ to the average of $VI(Q_{max})$ and $VI(\text{WSBM})$. Given how much higher $VI(\text{WSBM})$ is, that seems misleading. If you conduct 1-tailed t-tests relating $VI(Q_{max}\text{-WSBM})$ to $VI(\text{WSBM})$, and also $VI(Q_{max})$ to $VI(\text{WSBM})$, are there consistent significant differences across K? A conclusion that Q_{max} results in more consistent partitions than WSBM or the $Q_{max}\text{-WSBM}$ comparison is different than Q_{max} and WSBM both result in equivalently consistent partitions that are different from each other. I realize there is some variability in the other connectomes, but of the other four, it looks like 2/4 show similar results (where $VI(Q_{max}\text{-WSBM})$ and $VI(\text{WSBM})$ are relatively similar), and the other 2 do in a K-dependent manner (Figure S5).

Also in the Results section, in "Functional relevance of the WSBM", I think this sentence should be rephrased: "A strictly assortative brain is aligned with the hypothesis that the brain is composed of communities operating independently, while a brain that allows for some non-assortative communities implies that brain function arises not solely from contributions of independent communities, but from the interactions between communities." No one would argue that Q_{max} would ever find that a brain is "strictly assortative", or has zero connections across communities. Brain community structure derived from Q_{max} would find that brain function arises both from contributions of independent communities, as well as interactions across those communities. Even if all communities are assortative, they are not strictly assortative with no between-network connections. Perhaps rephrase to emphasize that WSBM-derived partitions allow for more types of interactions that are thought to be important for brain function. Further, I still think this section could use some clarification. I believe the point, which is not explicitly stated, is that both algorithms maximize within > between connectivity, so the algorithm that matches it is assumed to more accurately reflect true underlying connectivity. If a statement like this is the last sentence of the last paragraph on page 7, it will make that last logical step more clear to the readers.

Figure 6: I appreciate the clarification about the existence of disassortative communities from your response to the reviews and more opaque coloring of the disassortative communities in 6D. I think that you should explicitly state the value of WBSM- Q_{max} disassortative communities in 6E, since it looks like the value is 0 even though in your response you mentioned that it is slightly positive. Including that in the caption would clarify.

Reviewer #2 (Remarks to the Author):

I thank the authors for their comprehensive and very clear response to my prior concerns.

Reviewer #3 (Remarks to the Author):

In the revised version, I see that the authors included two major revisions:

- 1) comparison of the main results to those of a randomised network,
- 2) comparison of the main results to those of different parcellations.

In my opinion, these new extensions of the validation, clearly improve the quality of the analysis. The conclusions drawn from the randomized network analysis support the general hypothesis of the paper quite nicely; i.e. that the brain networks show some level of disassortativity, which, as shown with this new comparison, is not as severe as in random networks. Although, the comparison across different parcellations yields insignificant correlations between the results of different parcellations, there exist positive correlations and the insignificance can be attributed to the small number of samples as claimed by the authors.

Based on the new clarifications of the authors, my major concern is however, the seeming lack of convergence of WSBM (and potentially also of Qmax) across different trials:

In their response to my previous Comment 3, the authors state: 'Because both Qmax and the WSBM algorithms are non-deterministic – i.e. repeated runs of the algorithm usually result in slightly different solutions – we computed VI between pairs of partitions uncovered using the same community detection algorithm and also between algorithms. We now clarify this in the manuscript and figure caption. In the section Connectomes support diverse meso-scale architecture we now include the statement: "Specifically, we computed VI separately for three different subsets of partitions: partitions detected using WSBM with other WSBM partitions; partitions detected using Qmax with other Qmax partitions; partitions detected using the WSBM with Qmax partitions."'

Firstly, the newly added part and the Figure caption do not clearly state what the authors explain in their reply to the comment. For clarity of presentation and to aid the understanding of the general reader, I would suggest stating clearly in the manuscript, as in their reply above, such as: 'As both Qmax and the WSBM algorithms are non-deterministic – i.e. repeated runs of the algorithm usually result in slightly different solutions – we run both algorithms <nr of trials> times and computed VI between pairs of partitions across different trials uncovered using the same community detection algorithm and also between algorithms.'

Secondly, the large within-technique differences across different trials of WSBM algorithm (see my previous Comment 4) naturally raise the question about the potential lack of convergence of the algorithm. I assume, as in all non-deterministic algorithms, although the method may be expected to yield slightly different results, if it converges, these differences are not expected to vary dramatically.

Furthermore, in their replies to my previous comments 8 and 13, the authors state:

'We chose not to order the matrix by partitions detected by either WSBM or Qmax because there were thousands of such partitions and choosing a representative partition from among those was not trivial.'

'We appreciate the reviewer's attention to detail. In this case, the purple community and its relatively small size is a result of the stochasticity of the WSBM algorithm. That is, in attempting to optimize their respective objective functions, the output of both the WSBM and Qmax will vary somewhat. The communities shown in Figure 2 represent the outputs of single runs of the algorithm and should not be treated as necessarily representative of the network's ground truth communities. It is not difficult to identify a different partition of the same network into the same number of communities with comparably-sized communities. To demonstrate this, we show an alternative partition of the network

into five communities. Specifically, we chose the partition with the most similarly-sized communities. We have also replaced the WSBM partition in Figure 2 with the communities shown here.'

The communities in Figure 14 in the response to reviews and Figure 2 in the current manuscript seem more anatomically and functionally meaningful to me. However, the significant variation between different community assignments (see Figure 2 in the current and previous versions of the manuscript) resulting from two different runs of the same algorithm with the same initial parameters as well as the statement that any such partitioning could not be considered representative raise my concern about the potential lack of convergence and the reliability of the results. Can the authors please clarify this?

Minor comments:

- Page 3, in the newly added paragraph: 'Higher order cognitive processes, for example, are thought to emerge through integration of information originating in different brain systems [44], which can only occur via the interaction of communities with one another.' I wouldn't include the word "only" here, as the exact mechanism underlying the integration of information in the brain is currently unknown and there may be other possibilities than the one-to-one interaction between communities, as claimed here.

- Figure 3F: please state that the upper and lower limits of each box represent the 25th and 75th percentiles of each system's assortativity in the figure caption.

- Page 7, Functional relevance of the WSBM: "In the past when empirical estimates of FC could not be easily obtained, measures of similarity between brain regions' connectivity profiles (e.g., matching index) have been used as a stand-in [27, 28, 29]" Why would they not be easily obtained, as the FC estimates are simple correlations between different brain regions. Do the authors maybe mean 'before the empirical estimates of FC have been discovered'?

- Supplementary material, Page 9: '... and calculated the correlation of all system-level scores, obtaining coefficients of $r=0.32$ and $r=32$ ($p<0.01$)'. I believe $r=32$ is a typo here.

- Figure S7: The abbreviations RC and nonRC used in the figure caption are not defined in the supplementary material as rich club and non-rich club.

Reviewer #1

The authors responded to the reviewer comments thoroughly and, as a result, the manuscript is much stronger. I have a few minor comments, but no remaining substantial concerns.

We thank the reviewer for the positive remark.

Comment 1

Your reference to supplementary figures in the main text needs to be updated to reflect additional figures.

We apologize for this oversight. We now include references to the new supplementary figures throughout the main text and make sure that each Supplementary Figure is called out. In the beginning of the **Results** section (p.4), we now note that Figures S1 - S3 deal with null models and that we also test a cellular level, *C. elegans* network, and present the results in Figures S13 - S14. Later in the sections **Many (but not all) communities are assortative** and **Behavioral relevance of motif diversity** we now call out Figures S12, S15 - S17, which discuss rich clubs and the influence of brain parcellation.

Comment 2

In the Results section “Connectomes support diverse meso-scale architecture”, I believe that the asterisks in Figure 3A represent 1-tailed *t*-tests for each K relating $VI(Q_{max}\text{-WSBM})$ to the average of $VI(Q_{max})$ and $VI(\text{WSBM})$. Given how much higher $VI(\text{WSBM})$ is, that seems misleading. If you conduct 1-tailed *t*-tests relating $VI(Q_{max}\text{-WSBM})$ to $VI(\text{WSBM})$, and also $VI(Q_{max})$ to $VI(\text{WSBM})$, are there consistent significant differences across K ? A conclusion that Q_{max} results in more consistent partitions than WSBM or the $Q_{max}\text{-WSBM}$ comparison is different than Q_{max} and WSBM both result in equivalently consistent partitions that are different from each other. I realize there is some variability in the other connectomes, but of the other four, it looks like 2/4 show similar results (where $VI(Q_{max}\text{-WSBM})$ and $VI(\text{WSBM})$ are relatively similar), and the other 2 do in a K -dependent manner (Figure S5).

We apologize for any confusion. The tests performed in the main text were, in fact, two separate 1-tailed *t*-tests: the first compared $VI(Q_{max})$ with $VI(Q_{max}, \text{WSBM})$ while the second compared $VI(\text{WSBM})$ with $VI(Q_{max}, \text{WSBM})$. The asterisks represent values of K at which *both* tests were statistically significant. The reviewer is correct in noting that there are cases where the difference in means of $VI(\text{WSBM})$ and $VI(Q_{max}, \text{WSBM})$ are small, but because each group comprises 250^2 elements, the *t*-tests are powered enough to discern statistical differences. We have now added additional clarifying remarks:

In the section **Connectomes support diverse meso-scale architecture**:

- This procedure resulted in a series of within- and between- technique VI scores as a function of K . At each K , we computed one-tailed *t*-tests to assess whether the mean within-technique dissimilarity of partitions detected with either the WSBM or Q_{max} was smaller than the between-technique dissimilarity. We observed that from $K = 2, \dots, 9$, both the WSBM and Q_{max} uncovered partitions that were self-consistent yet distinct from one another (maximum $p < 10^{-15}$) (Note: asterisks in Fig. 3A indicate that *both t*-tests were statistically significant). This observation was consistent across the non-human connectome data as well (Fig. S5), confirming that the WSBM and Q_{max} generate statistically different estimates of connectome community structure.

The reviewer’s comments also speak more broadly to the issue of variability in the partitions detected by the WSBM and are therefore similar to Comment 2 made by Reviewer #3 concerning the performance of the WSBM algorithm. To address this point, we have provided two additional analyses of the human connectome dataset. First, we demonstrate qualitatively and visually that partitions detected using the WSBM are similar to one another. Second, we use statistical methods to quantify the observed level of

76 similarity and show that it is much greater than what would be expected under a permutation-based null
model. These results demonstrate that the WSBM partitions are statistically reliable, supporting the use of
the WSBM for community detection in network neuroscience.

In the main text we used the WSBM to partition brain networks into $K = 2, \dots, 10$ communities. Because
the WSBM algorithm was non-deterministic, we ran it multiple times from different initial conditions (250
repetitions), generating partition ensembles at each value of K . Here, we provide visual evidence suggesting
that the partitions comprising each ensemble are, in fact, similar to one another. Specifically, we compute
for each partition ensemble its association matrix, $\mathbf{T} \in \mathbb{R}^{N \times N}$, whose element, T_{ij} , is equal to the fraction
of partitions in which nodes, i and j , are assigned to the same community. If partitions were dissimilar to
one another, the association matrix would exhibit no structure. In Figure 3 we show examples of association
matrices generated from partitions obtained using the WSBM. Note that these matrices exhibit structure in
the form of non-uniform community co-assignment, providing visual confirmation that partitions generated
by the WSBM are relatively consistent across multiple runs of the algorithm.

Next, we quantify the average similarity of partitions to one another and show that this level of similarity
is much greater than what is expected by chance. As in the main text, we use variation of information (VI)
to quantify the similarity of two partitions to one another. We define the average similarity of each partition
ensemble as the mean pairwise VI across all possible pairs of partitions. To show that the detected partitions
are more similar to one another than expected by chance, we compare the observed mean pairwise VI of
each partition ensemble against a null distribution generated by a permutation-based null model in which
a node's community assignment is swapped with that of another in the same partition with probability r .
We vary the value of r from $r \approx 0.0017$ to $r = 1$ in 30 logarithmically-spaced steps and generate 100 null
values at each step. We then perform one-tailed non-parametric t -tests that the observed mean pairwise VI
is less than that of the null distribution and find that even when r is small, the observed value is statistically
smaller than expected by chance ($p < 10^{-2}$; corrected for multiple comparisons with a false-discovery rate
of 0.05) (Fig. 4). These results provide statistical evidence that the partitions detected using the WSBM
are, in fact, more similar to one another than expected. These results extend and complement the visual
evidence presented earlier.

Finally, it is worth noting that there are some reasons that we might expect partitions detected using the
WSBM to be more variable than those detected using Q_{max} . Both algorithms are tasked with estimating
nodes' community assignments. This problem is, of course, computationally intractable for all but the most
trivial cases [1]. However, the problem is compounded for the WSBM, which must also estimate for every
pair of communities a binary connection probability and the mean/variance of edges that fall between those
communities. All else being equal, this means that the space of possible solutions is much larger for the
WSBM than Q_{max} , leading to many near-optimal solutions on repeated runs.

We include these analyses in the Supplementary Material:

- • Here we summarize additional analyses of the human connectome dataset to characterize the variance
of solutions obtained using the WSBM. First, we demonstrate qualitatively and visually that partitions
detected using the WSBM are similar to one another. Second, we use statistical methods to quantify
the observed level of similarity and show that it is much greater than what would be expected under a
permutation-based null model. These results demonstrate that the WSBM partitions are statistically
reliable, supporting the use of the WSBM for community detection in network neuroscience.

In the main text we used the WSBM to partition brain networks into $K = 2, \dots, 10$ communities.
Because the WSBM algorithm was non-deterministic, we ran it multiple times from different initial
conditions (250 repetitions), generating partition ensembles at each value of K . Here, we provide
visual evidence suggesting that the partitions comprising each ensemble are, in fact, similar to one
another. Specifically, we compute for each partition ensemble its association matrix, $\mathbf{T} \in \mathbb{R}^{N \times N}$,
whose element, T_{ij} , is equal to the fraction of partitions in which nodes, i and j , are assigned to the
same community. If partitions were dissimilar to one another, the association matrix would exhibit no
structure. In Figure 3 we show examples of association matrices generated from partitions obtained
using the WSBM. Note that these matrices exhibit structure in the form of non-uniform community
co-assignment, providing visual confirmation that partitions generated by the WSBM are relatively

Figure 1: **Association matrices computed from partitions output by WSBM.** Each panel depicts a square, brain region \times brain region association matrix, whose elements indicate the fraction of all partitions in which two nodes were co-assigned to the same community. Brain areas are ordered according to a randomly selected partition. Sub-panels correspond to different numbers of communities, $K = 2, \dots, 10$.

consistent across multiple runs of the algorithm.

Next, we quantify the average similarity of partitions to one another and show that this level of similarity is much greater than what is expected by chance. As in the main text, we use variation of information (VI) to quantify the similarity of two partitions to one another. We define the average similarity of each partition ensemble as the mean pairwise VI across all possible pairs of partitions. To show that the detected partitions are more similar to one another than expected by chance, we compare the observed mean pairwise VI of each partition ensemble against a null distribution generated by a permutation-based null model in which a node's community assignment is swapped with that of another in the same partition with probability r . We vary the value of r from $r \approx 0.0017$ to $r = 1$ in 30 logarithmically-spaced steps and generate 100 null values at each step. We then perform one-tailed non-parametric t -tests that the observed mean pairwise VI is less than that of the null distribution and find that even when r is small, the observed value is statistically smaller than expected by chance ($p < 10^{-2}$; corrected for multiple comparisons with a false-discovery rate of 0.05) (Fig. 4). These results provide statistical evidence that the partitions detected using the WSBM are, in fact, more similar to one another than expected. These results extend and complement the visual evidence presented earlier.

144

145

Finally, it is worth noting that there are some reasons that we might expect partitions detected using

Figure 2: **Mean pairwise variation of information (VI) of original and randomized partitions.** For a given number of communities, K , we estimated the mean pairwise VI, which serves as a measure of partition similarity. Lower values of VI imply greater similarity. The VI of the original partitions is shown as a red line. Using a parameterized permutation-based null model, we generate randomized partitions and compute null distributions for the mean pairwise VI. For all values of $K = 2, \dots, 10$, and for all parameter values, even small changes to community assignments result in statistically significant increases in mean pairwise VI. These observations support the hypothesis that the WSBM algorithm is converging to a set of solutions that are consistent and self-similar.

the WSBM to be more variable than those detected using Q_{max} . Both algorithms are tasked with estimating nodes' community assignments. This problem is, of course, computationally intractable for all but the most trivial cases [1]. However, the problem is compounded for the WSBM, which must also estimate for every pair of communities a binary connection probability and the mean/variance of edges that fall between those communities. All else being equal, this means that the space of possible solutions is much larger for the WSBM than Q_{max} , leading to many near-optimal solutions on repeated runs.

We also call out these analyses and figures in the main text in the section **Weighted stochastic block-model**:

- We explore the convergence of the WSBM across multiple repetitions and the similarity of detected partitions in the **Supplementary Material** (Figs. S18, S19).

Comment 3

Also in the Results section, in “Functional relevance of the WSBM”, I think this sentence should be rephrased: “A strictly assortative brain is aligned with the hypothesis that the brain is composed of communities operating independently, while a brain that allows for some non-assortative communities implies that brain function arises not solely from contributions of independent communities, but from the interactions between communities.” No one would argue that Q_{max} would ever find that a brain is “strictly assortative”, or has zero connections across communities. Brain community structure derived from Q_{max} would find that brain

*function arises both from contributions of independent communities, as well as interactions across those com-*
*munities. Even if all communities are assortative, they are not strictly assortative with no between-network*
*connections. Perhaps rephrase to emphasize that WSBM-derived partitions allow for more types of interac-*
*tions that are thought to be important for brain function. Further, I still think this section could use some*
*clarification. I believe the point, which is not explicitly stated, is that both algorithms maximize within >*
*between connectivity, so the algorithm that matches it is assumed to more accurately reflect true underlying*
*connectivity. If a statement like this is the last sentence of the last paragraph on page 7, it will make that*
*last logical step more clear to the readers.*

We agree with the reviewer that even when brain network communities are assortative, in practice we tend
to find a small fraction of brain areas whose links span communities. The point that we intended to make
was that non-assortative community structure implies that there exist entire *groups* of brain areas (not
just individual areas) whose collective connectivity pattern may predispose them to integrative function,
rather than functioning in isolation. The reviewer is also correct in noting that both algorithms, in theory,
detect communities that we would expect to exhibit greater within-community *functional* connectivity than
between.

In line with the reviewer’s suggestion, we have added clarifying remarks to this section.

• Though *via* different mechanisms, both the WSBM and Q_{max} produce communities composed of
brain regions with similar patterns of incoming and outgoing connections **and so we would expect the**
**resulting communities to be internally dense in terms of functional connectivity.** In the case of Q_{max} ,
this similarity is entirely incidental – nodes get grouped into internally dense, mutually-connected
clusters, inflating their similarity. The WSBM, on the other hand, explicitly defines communities as
clusters of nodes whose connections were generated by the same statistical process; by definition pairs
of nodes in the same community will have similar connectivity patterns even if they, themselves, are
not directly connected.

• Because the similarity of regions’ structural connectivity is associated with strong functional connectiv-
ity, we expect that two nodes in the same community should be more strongly functionally connected
to one another than two nodes in different communities, irrespective of which technique was used to de-
fine the communities. However, the WSBM and Q_{max} represent vastly different hypotheses about how
brain networks function. **An assortative brain is aligned with the hypothesis that communities function**
**and process information relatively independently from one another,** while a brain that allows for some
non-assortative communities implies that function arises not solely from contributions of independent
communities, but from the interactions between communities. **Whereas past work has emphasized the**
**assortative model of brain function, in which integration is performed by a few outlying nodes whose**
**connections span community boundaries, the non-assortative model holds that integration is funda-**
**mentally a community-level action performed by clusters of brain areas with similar (non-assortative)**
**connectivity profiles.**

**Comment 4**

*Figure 6: I appreciate the clarification about the existence of disassortative communities from your response*
*to the reviews and more opaque coloring of the disassortative communities in 6D. I think that you should*
*explicitly state the value of WBSM- Q_{max} disassortative communities in 6E, since it looks like the value is 0*
*even though in your response you mentioned that it is slightly positive. Including that in the caption would*
*clarify.*

We have now included a clarifying remark in the caption of Figure 6D.

• **Note: The WSBM does, in fact, generate a small fraction of disassortative communities and so points**
**on the red curves in D and E are not equal to zero.**

Reviewer #2

*I thank the authors for their comprehensive and very clear response to my prior concerns.*

We appreciate the reviewer’s response.

Reviewer #3

*In the revised version, I see that the authors included two major revisions:*

- • *comparison of the main results to those of a randomised network.*
- • *comparison of the main results to those of different parcellations.*

*In my opinion, these new extensions of the validation, clearly improve the quality of the analysis. The*
*conclusions drawn from the randomized network analysis support the general hypothesis of the paper quite*
*nicely; i.e. that the brain networks show some level of disassortativity, which, as shown with this new*
*comparison, is not as severe as in random networks. Although, the comparison across different parcellations*
*yields insignificant correlations between the results of different parcellations, there exist positive correlations*
*and the insignificance can be attributed to the small number of samples as claimed by the authors.*

*Based on the new clarifications of the authors, my major concern is however, the seeming lack of con-*
*vergence of WSBM (and potentially also of Q_{max}) across different trials:*

*In their response to my previous Comment 3, the authors state: “Because both Q_{max} and the WSBM*
*algorithms are non-deterministic i.e. repeated runs of the algorithm usually result in slightly different solu-*
*tions we computed VI between pairs of partitions uncovered using the same community detection algorithm*
*and also between algorithms. We now clarify this in the manuscript and figure caption. In the section Con-*
*nectomes support diverse meso-scale architecture we now include the statement: “Specifically, we computed*
*VI separately for three different subsets of partitions: partitions detected using WSBM with other WSBM*
*partitions; partitions detected using Q_{max} with other Q_{max} partitions; partitions detected using the WSBM*
*with Q_{max} partitions.”*

Comment 1

*Firstly, the newly added part and the Figure caption do not clearly state what the authors explain in their*
*reply to the comment. For clarity of presentation and to aid the understanding of the general reader, I*
*would suggest stating clearly in the manuscript, as in their reply above, such as: “As both Q_{max} and the*
*WSBM algorithms are non-deterministic i.e. repeated runs of the algorithm usually result in slightly different*
*solutions we run both algorithms <nr of trials> times and computed VI between pairs of partitions across*
*different trials uncovered using the same community detection algorithm and also between algorithms.”*

We apologize for the lack of clarification and agree with the reviewer that the manuscript should clearly
reflect the number of trials and partition pairs over which VI was computed. We have now included the
following statement in the main text:

In the first paragraph of **Results**:

- • *As both the Q_{max} and WSBM algorithms are non-deterministic – i.e. repeated runs of the algorithm*
*usually result in slightly different solutions – we varied the number of communities from $K = 2$ to*
*$K = 10$ and repeated both algorithms 250 times for each K .*

And in the section **Connectomes support diverse meso-scale structure**:

- • *Specifically, we computed pairwise VI among all 250 partitions detected using Q_{max} and separately for*
*partitions detected using the WSBM. We also computed pairwise VI between the 250 Q_{max} partitions*
*and the 250 WSBM partitions. This process was repeated separately for different values of K , the*
*number of detected communities, which made the comparison as fair as possible.*

Comment 2

Secondly, the large within-technique differences across different trials of WSBM algorithm (see my previous Comment 4) naturally raise the question about the potential lack of convergence of the algorithm. I assume, as in all non-deterministic algorithms, although the method may be expected to yield slightly different results, if it converges, these differences are not expected to vary dramatically.

Furthermore, in their replies to my previous comments 8 and 13, the authors state: “We chose not to order the matrix by partitions detected by either WSBM or Qmax because there were thousands of such partitions and choosing a representative partition from among those was not trivial.”

“We appreciate the reviewer’s attention to detail. In this case, the purple community and its relatively small size is a result of the stochasticity of the WSBM algorithm. That is, in attempting to optimize their respective objective functions, the output of both the WSBM and Qmax will vary somewhat. The communities shown in Figure 2 represent the outputs of single runs of the algorithm and should not be treated as necessarily representative of the network’s ground truth communities. It is not difficult to identify a different partition of the same network into the same number of communities with comparably-sized communities. To demonstrate this, we show an alternative partition of the network into five communities. Specifically, we chose the partition with the most similarly-sized communities. We have also replaced the WSBM partition in Figure 2 with the communities shown here.”

The communities in Figure 14 in the response to reviews and Figure 2 in the current manuscript seem more anatomically and functionally meaningful to me. However, the significant variation between different community assignments (see Figure 2 in the current and previous versions of the manuscript) resulting from two different runs of the same algorithm with the same initial parameters as well as the statement that any such partitioning could not be considered representative raise my concern about the potential lack of convergence and the reliability of the results. Can the authors please clarify this?

We agree with the reviewer that the convergence of the WSBM algorithm is an important technical point and one that we wish to clarify. We also note that the Reviewer’s comment – dealing with the variability of optimal partitions – is similar to Comment 2 made by Reviewer #1. Here, the reviewer asks whether the WSBM is arriving at dissimilar solutions over different runs. To address this point, we have provided two additional analyses of the human connectome dataset. First, we demonstrate qualitatively and visually that partitions detected using the WSBM are similar to one another. Second, we use statistical methods to quantify the observed level of similarity and show that it is much greater than what would be expected under a permutation-based null model. These results demonstrate that the WSBM partitions are statistically reliable, supporting the use of the WSBM for community detection in network neuroscience.

In the main text we used the WSBM to partition brain networks into $K = 2, \dots, 10$ communities. Because the WSBM algorithm was non-deterministic, we ran it multiple times from different initial conditions (250 repetitions), generating partition ensembles at each value of K . Here, we provide visual evidence suggesting that the partitions comprising each ensemble are, in fact, similar to one another. Specifically, we compute for each partition ensemble its association matrix, $\mathbf{T} \in \mathbb{R}^{N \times N}$, whose element, T_{ij} , is equal to the fraction of partitions in which nodes, i and j , are assigned to the same community. If partitions were dissimilar to one another, the association matrix would exhibit no structure. In Figure 3 we show examples of association matrices generated from partitions obtained using the WSBM. Note that these matrices exhibit structure in the form of non-uniform community co-assignment, providing visual confirmation that partitions generated by the WSBM are relatively consistent across multiple runs of the algorithm.

Next, we quantify the average similarity of partitions to one another and show that this level of similarity is much greater than what is expected by chance. As in the main text, we use variation of information (VI) to quantify the similarity of two partitions to one another. We define the average similarity of each partition ensemble as the mean pairwise VI across all possible pairs of partitions. To show that the detected partitions are more similar to one another than expected by chance, we compare the observed mean pairwise VI of each partition ensemble against a null distribution generated by a permutation-based null model in which a node’s community assignment is swapped with that of another in the same partition with probability r . We vary the value of r from $r \approx 0.0017$ to $r = 1$ in 30 logarithmically-spaced steps and generate 100 null values at each step. We then perform one-tailed non-parametric t -tests that the observed mean pairwise VI

Figure 3: **Association matrices computed from partitions output by WSBM.** Each panel depicts a square, brain region \times brain region association matrix, whose elements indicate the fraction of all partitions in which two nodes were co-assigned to the same community. Brain areas are ordered according to a randomly selected partition. Sub-panels correspond to different numbers of communities, $K = 2, \dots, 10$.

is less than that of the null distribution and find that even when r is small, the observed value is statistically
 smaller than expected by chance ($p < 10^{-2}$; corrected for multiple comparisons with a false-discovery rate
 of 0.05) (Fig. 4). These results provide statistical evidence that the partitions detected using the WSBM
 are, in fact, more similar to one another than expected. These results extend and complement the visual
 evidence presented earlier.

Finally, it is worth noting that there are some reasons that we might expect partitions detected using the
 WSBM to be more variable than those detected using Q_{max} . Both algorithms are tasked with estimating
 nodes' community assignments. This problem is, of course, computationally intractable for all but the most
 trivial cases [1]. However, the problem is compounded for the WSBM, which must also estimate for every
 pair of communities a binary connection probability and the mean/variance of edges that fall between those
 communities. All else being equal, this means that the space of possible solutions is much larger for the
 WSBM than Q_{max} , leading to many near-optimal solutions on repeated runs.

We include these analyses in the Supplementary Material:

- Here we summarize additional analyses of the human connectome dataset to characterize the variance of solutions obtained using the WSBM. First, we demonstrate qualitatively and visually that partitions

Figure 4: **Mean pairwise variation of information (VI) of original and randomized partitions.** For a given number of communities, K , we estimated the mean pairwise VI, which serves as a measure of partition similarity. Lower values of VI imply greater similarity. The VI of the original partitions is shown as a red line. Using a parameterized permutation-based null model, we generate randomized partitions and compute null distributions for the mean pairwise VI. For all values of $K = 2, \dots, 10$, and for all parameter values, even small changes to community assignments result in statistically significant increases in mean pairwise VI. These observations support the hypothesis that the WSBM algorithm is converging to a set of solutions that are consistent and self-similar.

detected using the WSBM are similar to one another. Second, we use statistical methods to quantify the observed level of similarity and show that it is much greater than what would be expected under a permutation-based null model. These results demonstrate that the WSBM partitions are statistically reliable, supporting the use of the WSBM for community detection in network neuroscience.

In the main text we used the WSBM to partition brain networks into $K = 2, \dots, 10$ communities. Because the WSBM algorithm was non-deterministic, we ran it multiple times from different initial conditions (250 repetitions), generating partition ensembles at each value of K . Here, we provide visual evidence suggesting that the partitions comprising each ensemble are, in fact, similar to one another. Specifically, we compute for each partition ensemble its association matrix, $\mathbf{T} \in \mathbb{R}^{N \times N}$, whose element, T_{ij} , is equal to the fraction of partitions in which nodes, i and j , are assigned to the same community. If partitions were dissimilar to one another, the association matrix would exhibit no structure. In Figure 3 we show examples of association matrices generated from partitions obtained using the WSBM. Note that these matrices exhibit structure in the form of non-uniform community co-assignment, providing visual confirmation that partitions generated by the WSBM are relatively consistent across multiple runs of the algorithm.

Next, we quantify the average similarity of partitions to one another and show that this level of similarity is much greater than what is expected by chance. As in the main text, we use variation of information (VI) to quantify the similarity of two partitions to one another. We define the average similarity of each partition ensemble as the mean pairwise VI across all possible pairs of partitions. To show that the detected partitions are more similar to one another than expected by chance, we

compare the observed mean pairwise VI of each partition ensemble against a null distribution generated by a permutation-based null model in which a node’s community assignment is swapped with that of another in the same partition with probability r . We vary the value of r from $r \approx 0.0017$ to $r = 1$ in 30 logarithmically-spaced steps and generate 100 null values at each step. We then perform one-tailed non-parametric t -tests that the observed mean pairwise VI is less than that of the null distribution and find that even when r is small, the observed value is statistically smaller than expected by chance ($p < 10^{-2}$; corrected for multiple comparisons with a false-discovery rate of 0.05) (Fig. 4). These results provide statistical evidence that the partitions detected using the WSBM are, in fact, more similar to one another than expected. These results extend and complement the visual evidence presented earlier.

Finally, it is worth noting that there are some reasons that we might expect partitions detected using the WSBM to be more variable than those detected using Q_{max} . Both algorithms are tasked with estimating nodes’ community assignments. This problem is, of course, computationally intractable for all but the most trivial cases [1]. However, the problem is compounded for the WSBM, which must also estimate for every pair of communities a binary connection probability and the mean/variance of edges that fall between those communities. All else being equal, this means that the space of possible solutions is much larger for the WSBM than Q_{max} , leading to many near-optimal solutions on repeated runs.

We also call out these analyses and figures in the main text in the section **Weighted stochastic block-model**:

- We explore the convergence of the WSBM across multiple repetitions and the similarity of detected partitions in the **Supplementary Material** (Figs. S18, S19).

Comment 2

Page 3, in the newly added paragraph: “Higher order cognitive processes, for example, are thought to emerge through integration of information originating in different brain systems [44], which can only occur via the interaction of communities with one another.” I wouldn’t include the word “only” here, as the exact mechanism underlying the integration of information in the brain is currently unknown and there may be other possibilities than the one-to-one interaction between communities, as claimed here.

We agree with the reviewer and have made the suggested change.

Comment 3

Figure 3F: please state that the upper and lower limits of each box represent the 25th and 75th percentiles of each system’s assortativity in the figure caption.

We have followed the reviewers’s suggestion and now define the limits of the box in each plot.

- The limits of each box represent the interquartile range (25th and 75th percentiles).

Comment 4

Page 7, Functional relevance of the WSBM: “In the past when empirical estimates of FC could not be easily obtained, measures of similarity between brain regions” connectivity profiles (e.g., matching index) have been used as a stand-in [27, 28, 29]” Why would they not be easily obtained, as the FC estimates are simple correlations between different brain regions. Do the authors maybe mean “before the empirical estimates of FC have been discovered”?

The “matching index” and other metrics that quantified the structural overlap of connections have been, in
the past, used as stand-ins for FC or the functional relatedness of brain areas with respect to one another. The
usage of these metrics predates the now widespread practice of estimating functional connectivity empirically
from the correlation of activity time series. We now note this more clearly.

The passage now reads:

- • In the past before it was common to empirically estimate FC as the correlation of neural activity,
measures of similarity between brain regions’ connectivity profiles (e.g., matching index) were used as
a stand-in.

**Comment 4**

*Supplementary material, Page 9: “... and calculated the correlation of all system-level scores, obtaining*
*coefficients of $r=0.32$ and $r=32$ ($p<0.01$)”. I believe $r=32$ is a typo here.*

The reviewer is correct: this was a typographical error leaving out the decimal point. By coincidence, the
correlation coefficients were both $r = 0.32$.

**Comment 6**

*Figure S7: The abbreviations RC and nonRC used in the figure caption are not defined in the supplementary*
*material as rich club and non-rich club.*

We have updated the caption to indicate what RC and non-RC refer to.

- • The labels RC and non-RC used in panels C, G, K, O indicate nodes that were assigned to or not assigned
to putative rich clubs See **Rich club estimation** for more details.

**References**

[1] Santo Fortunato. Community detection in graphs. *Physics reports*, 486(3):75–174, 2010.

REVIEWERS' COMMENTS:

Reviewer #1 (Remarks to the Author):

The authors thoroughly responded to this most recent round of reviews and I have no further comments. When published, this article will make an important contribution to the literature.

Reviewer #3 (Remarks to the Author):

I thank the authors for addressing my raised concerns and attending my suggestions. I believe the newly added analysis nicely shows the convergence of the WSBM, which was my major question. I do not have any further comments or questions.

Dear Reviewers and Editor,

The referees raised no additional comments/questions in this last round of review (we include their final remarks, below). We thank them and the editor for their suggestions throughout the review process. As a result, the manuscript has been improved substantially.

Sincerely,
The authors

Reviewer #1 (Remarks to the Author):

The authors thoroughly responded to this most recent round of reviews and I have no further comments. When published, this article will make an important contribution to the literature.

Reviewer #3 (Remarks to the Author):

I thank the authors for addressing my raised concerns and attending my suggestions. I believe the newly added analysis nicely shows the convergence of the WSBM, which was my major question. I do not have any further comments or questions.